# Quantum measurements in fundamental physics: a user's manual

Jacob Beckey,[1,2,3] Daniel Carney,[1] and Giacomo Marocco[1]

[1]*Physics Division, Lawrence Berkeley National Laboratory, Berkeley, CA*
[2]*JILA, NIST and University of Colorado, Boulder, CO*
[3]*Department of Physics, University of Colorado, Boulder, CO*

We give a systematic theoretical treatment of linear quantum detectors used in modern high energy physics experiments, including dark matter cavity haloscopes, gravitational wave detectors, and impulsive mechanical sensors. We show how to derive the coupling of signals of interest to these devices, and how to calculate noise spectra, signal-to-noise ratios, and detection sensitivities. We emphasize the role of quantum vacuum and thermal noise in these systems. Finally, we review ways in which advanced quantum techniques—squeezing, non-demolition measurements, and entanglement—can be or currently are used to enhance these searches.

**CONTENTS**

jacob.beckey@colorado.edu, carney@lbl.gov, gmarocco@lbl.gov

# I.  INTRODUCTION

Detectors operate under the laws of quantum mechanics, which place non-trivial constraints on their capabilities [1, 2]. In recent years, sensors used in the search for fundamental targets like gravitational waves [3–6] and new fields beyond the Standard Model of particle physics [7, 8] have begun operating in the regime where these quantum mechanical limitations are increasingly relevant. Perhaps surprisingly, the properties of this fundamental quantum noise can be engineered, and even reduced [9–14].

In this review, we explain the sources of this quantum noise, how it appears both in principle and in practical experiments, and methods to circumvent it. Our primary goal is to remove mystery from this subject by showing how one can systematically calculate couplings of signals of interest to a given detector, how the detector's quantum mechanics affect its sensitivity, and how to use this information to predict the reach of an experiment.

To do this, we first introduce the general tools of what is called the input-output formalism, originally developed in quantum optics [15]. We then apply it to a wide variety of examples. This framework enables simple, microscopic models for a sensor coupled to signals, to its readout system, and to various baths which contribute to noise, including baths which are in their vacuum or more exotic quantum states. It also allows one to easily include other phenomenological contributions to the noise budget, for example, from measured sources of technical noise. This formalism is particularly powerful for linear detectors, i.e., those we can treat in linear response theory.

Since we are focusing on quantum noise effects, we should mention when these are actually important. In a real experiment, an enormous degree of classical engineering and design has to be done to get the system to the point where its dominant noise contributions come from quantum vacuum fluctuations. Although immensely challenging, this is now happening in a variety of practical experiments [7, 8, 13], and will continue to do so moving forward. Another crucial question is when it is better to "just" build a bigger, effectively classical device, rather than a quantum-limited one. This can only be analyzed on a case-by-case basis, but we emphasize that fundamental physics provides unique cases where a quantum detector is the only way forward: for example, devices requiring ultra-low energy thresholds, single-quantum detectors, or more generally devices searching for transients which need to integrate the signal as fast as possible in a fixed volume. Thus, we feel that now is a good time to lay out the basic foundations of the subject in a user-friendly manner.

The paper is structured according to the Table of Contents. For readers seeking more details on the topics presented here, we collate some helpful reviews: quantum noise and measurement in general [16–18]; gravitational wave detection with interferometers [19, 20]; searches for light axion and axion-like dark matter [21, 22]. Our conventions for units, etc. are given in Appendix A.

# II.  QUANTUM NOISE: BASICS

We are interested in the task of measuring the state of a system that is weakly coupled to some physics signal. The quantum mechanics of this measurement process will play a crucial role in determining the noise, and thus the possible sensitivity, of an experimental set-up. In this section, we give an introductory overview of how continuous measurements are modeled in quantum mechanics, as well as the basic signal processing needed to determine the sensitivity of a device to a given signal.

## A.  "Standard Quantum Limits"

Before diving into the details of noise spectra and sensitivities, we begin with a slightly more heuristic discussion which will help introduce some basic concepts in the quantum theory of measurement.

Consider trying to estimate the position $x$ of a fixed mirror by reflecting photons off the mirror; by fixed we mean the mirror is not dynamical, a condition we will relax momentarily. We will do this with an interferometer, as shown in Fig. 1. We use the labels $|0\rangle$ and $|1\rangle$ to label the two possible paths of a given photon at each step of the protocol. Light of wavelength $\lambda = 1/\omega$ will have wavefronts which separate by a relative phase $\phi = x/\lambda$ when they are recombined at the final beamsplitter. This gives a probability to be detected in the final $|0, 1\rangle$ ports

$$P(0) = |\langle 0|U|0\rangle|^2 = \sin^2 \frac{\phi}{2}$$
$$P(1) = |\langle 1|U|0\rangle|^2 = \cos^2 \frac{\phi}{2} \tag{1}$$

where $U$ is the total evolution $U = U_{\text{BS}}^\dagger U_\phi U_{\text{BS}}$.[1] Sending $N$ uncorrelated photons, we can make a histogram of hits $b = 0, 1$ in the two ports, and use it to infer the value of $\phi$ by setting $\sin^2 \phi/2 = \bar{b}$. With $\phi \ll 1$, i.e., for small displacements $x/\lambda \ll 1$, most outcomes are 0. Thus one can estimate that the variance var $b \approx \sin^2 \phi/2 \approx (\phi/2)^2$, and then propagation of error gives the standard deviation

$$\Delta\phi \approx \frac{\phi/(2\sqrt{N})}{\phi/2} = \frac{1}{\sqrt{N}} \tag{3}$$

_________

[1] This is essentially what we will refer to as a "homodyne" measurement throughout this paper: the light reflected off the object of interest is interfered with light which freely propagates from the initial laser, which provides a "local oscillator" of known phase [23]. The unitaries here are

$$U_{\text{BS}} = \frac{1}{\sqrt{2}} \begin{pmatrix} 1 & -1 \\ 1 & 1 \end{pmatrix}, \quad U_\phi = \begin{pmatrix} 1 & 0 \\ 0 & e^{i\phi} \end{pmatrix}. \tag{2}$$

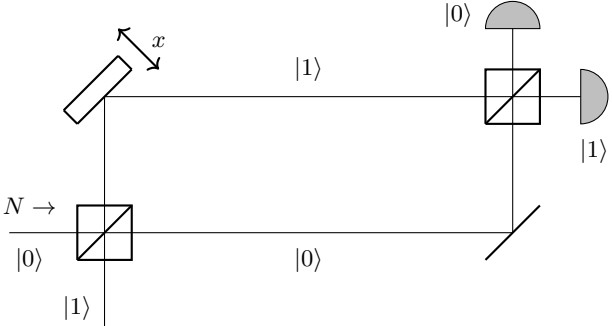

FIG. 1. **Prototypical interferometer for position measurements:** Light of wavelength $\lambda$ is incident on a beamsplitter. It then propagates freely along one path while interacting with the sensing element on the other (here, a mirror displaced by $x$ from the equal-path-length configuration). Finally, the two beams are recombined with an inverse beamsplitter. The light intensity (photon number) is counted in each output port. The change to the phase of the light from the interaction is denoted $\phi = x/\lambda$, which can be inferred from the measurement of the two final intensities.

or more appropriately

$$\Delta x \approx \frac{\lambda}{\sqrt{N}}. \qquad (4)$$

In words, a single photon can resolve the position with accuracy $\lambda$, and then $N$ uncorrelated photons can be used to average this error down like $1/\sqrt{N}$.

The limit (4), particularly the $1/\sqrt{N}$ scaling, is sometimes referred to as a "Standard Quantum Limit" (SQL), especially in the context of quantum metrology or discussions of the Fisher information; it is more appropriate to call this a shot noise limit [24–26]. However, this is a quantum limit in the following sense: one can show that non-trivial quantum states of the light, for example squeezed states, can enable scaling which goes faster than $1/\sqrt{N}$ (as fast as $1/N$). We discuss this in more detail, and in particular explain the precise sense in which such states are "non-classical", in Sec. V.

Now we contrast this shot noise limit (4) to a different kind of SQL, more applicable to the kinds of problems we will be focused on [2]. Consider again measuring the position $x$ of the mirror, but now at two times $t_1, t_2 = t_1 + \Delta t$. For example, we may want to know how far the slab moved in this time interval in order to determine if an external force acted on it. In the first time step, we measure the position $x$ with some uncertainty $\Delta x_1$, for example using the interferometry protocol of the previous paragraph. After the measurement, the mirror state must have

$$\Delta p_1 \geq \frac{2}{\Delta x_1}, \qquad (5)$$

by Heisenberg uncertainty. Now we wait for a time $\Delta t$, during which the oscillator evolves under the Schrödinger equation. Assuming $\Delta t$ is sufficiently small, this is approximately free evolution under $H \sim p^2/2m$, which means that the wavefunction spreads in position space:

$$\Delta x_2 = \Delta x_1 + \frac{\Delta p_1}{2m}\Delta t. \qquad (6)$$

We see a clear tradeoff: a better measurement $\Delta x_1 \to 0$ in the first step leads to a worse measurement $\Delta x_2 \to \infty$ in the second step. We can find an optimized solution by asking that $\Delta x$ is the same at each time step and minimized, by differentiating this equation and solving for $\Delta x$. One finds a simple result:

$$\Delta x_{\mathrm{SQL}} = \sqrt{\frac{\Delta t}{m}}. \qquad (7)$$

To compare to a famous example [3], suppose we are trying to look for gravitational waves at frequency $\sim 100$ Hz with detectors of mass $m \approx 40$ kg. Then $\Delta x_{\mathrm{SQL}} \approx 10^{-19}$ m. Much like the limit (4), this SQL (7) can also be circumvented. Here this would mean resolving changes in the state of the system over time with accuracy better than (7). In Sec. V, we give a number of ways to do this.

Two differences between the first type of limit (4) and the latter (7) should be emphasized. The first limit applies, as described here, to a measurement with $N$ probes at one fixed time. The second, on the other hand, refers to a differential measurement in time. Moreover, the first limit treats the signal – in our example, the position $x$ – as a fixed external parameter. In the limit (7), it is critical that the measurement at $t_1$ actually *changes the state of the measured system*, and this change propagates as an error in the second $t_2$ measurement.

As a consequence of these differences, the conclusions look superficially different: in particular, (4) suggests that we can measure the position variable to arbitrary accuracy at a given time, simply by sending $N \to \infty$ probes (e.g., by turning up a laser power). In contrast, (7) is *independent* of the number of probes—in fact, it reflects the idea that too many probes (e.g., too intense of a laser) will cause problems at $t_2$ by affecting a measurement at $t_1$ which is too accurate. To achieve (7), one is balancing two effects: reduction of the shot noise in the initial measurement and the consequent increase in "measurement back-action" in the second step. This actually implies a certain optimal number of probes (e.g., an optimum, finite laser power $P_{\mathrm{SQL}}$). We will say much more about this later, and see how this tradeoff works out mechanically in computations, particularly in Sec. IV.

In what follows, we will use the term SQL to refer to limits like (7), as statements about noise being limited by a balanced tradeoff between shot noise and quantum measurement back-action in time-dependent measurements. As a final remark, we offer a more heuristic definition of an SQL for the continuous measurement problems studied in this review. Consider continuously observing a harmonic oscillator at its resonant frequency, $\Delta t = 1/\omega$, so Eq. (7) reads $\Delta x_{\mathrm{SQL}} = \sqrt{1/m\omega}$. This is,

up to a factor of $\sqrt{2}$, just the width of the ground state wavefunction of the oscillator! Thus, observing at the SQL means that we can resolve the position of the oscillator with an accuracy set by its "quantum vacuum fluctuations". As a baseline intuitive picture, this is robust in all the examples that follow below. It is also a helpful way of understanding why these limits really denote the border between classical and quantum-limited measurements: anything better than the SQL means measurement below the scale of vacuum fluctuations.

## B. Warmup example: monitoring a pendulum

We now begin our excursion into precision quantum measurement theory with a simple example detector that demonstrates many of the quantum features of interest: a dielectric object suspended as a pendulum and monitored continuously with optical light. See Fig. 2. The essential working principle is that we can monitor the position of the dielectric object by looking at the phase shifts in laser photons scattering off it, much like the mirror of the previous section. The core idea, which will generalize to all the detectors studied in this review, is that we have a quantum mechanical readout system (here, the incident and scattered light) which monitors a particular quantum mechanical sensing element (here, the dielectric).

This model was chosen for two reasons. One is that it serves as an excellent toy model for a number of real detectors, for example the levitated dielectric force sensors of [27–29] and other mechanical sensors studied in Sec. IV. More importantly, however, it provides an example where we can give an explicit microscopic model of both the readout system and its coupling to the sensing element, and show how to systematically reduce this to a simple "quantum optics"-style description [30, 31]. We will use many lessons from this example to motivate the general input-output framework given in Sec. II C. Here we will sketch the important physics of this detector, but we give a complete and detailed derivation of all the results in Appendix B.

Consider a planar dielectric slab, of total mass $m$, pendulum frequency $\omega_m$, thickness $\ell$, and dielectric polarizability $\chi_e$, as shown in Fig. 2. The Hamiltonian for the center-of-mass $x$ of the slab is a simple harmonic oscillator

$$H_{\text{slab}} = \frac{p^2}{2m} + \frac{1}{2}m\omega_m^2 x^2,\qquad(8)$$

where we are assuming that the motion in the $y, z$ axes is negligible. The detailed nature of the trapping potential is not important; in practice it could be a literal suspension system as in Fig. 2, or an optical tweezing field [32], or a number of other variations.

To monitor the slab's position $x$, we shine a laser on the slab from the left, and we assume that we can measure photons which are either reflected or transmitted. The laser is aimed directly at the slab, so the whole system has

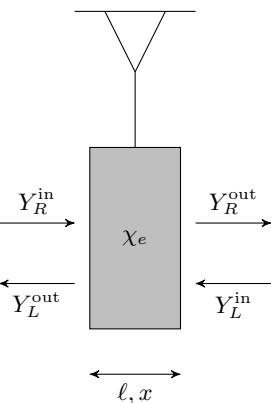

FIG. 2. **Dielectric slab.** Schematic diagram of a dielectric slab of mass $m$, width $\ell$, and electric polarizability $\chi_e$ suspended as a harmonic pendulum with frequency $\omega_m$, interacting with left- and right-moving electromagnetic waves. The coordinate $x$ denotes the displacement of the slab from its classical equilibrium position.

cylindrical symmetry around the beam, which will reduce the problem to a one-dimensional model. Assuming that the slab is a linear, homogeneous dielectric, it responds to the electric field by polarizing $\mathbf{P}(\mathbf{r}) = \chi_e\mathbf{E}(\mathbf{r})$, where $\chi_e = \epsilon_r - 1$ and $\epsilon_r$ is the dielectric constant. The potential describing this interaction is

$$V_{\text{int}}(x,t) = -\frac{1}{2}\chi_e \int_{\text{slab}} d^3\mathbf{r}\ |\mathbf{E}(\mathbf{r},t)|^2,\qquad(9)$$

where the dependence on the center of mass, $x$, will enter through the limits of integration.

The electromagnetic potential can be quantized as usual, by expanding into a complete basis of plane wave modes. It will be convenient to write the electric field $\mathbf{E} = \mathbf{E}_0 + \delta\mathbf{E}_L + \delta\mathbf{E}_R$, where the first term represents the laser drive, and the next two terms represent the left- and right-moving photon fluctuations around this drive. We can expand the interaction (9) around small displacements $x$ of the slab, include the laser drive, and reduce the problem to a simple interaction Hamiltonian of the form

$$V = V(0) + V_{RR} + 2V_{RL} + V_{LL},\qquad(10)$$

where $V(0)$ is independent of $x$, and the other terms are linear in $x$. Here, assuming the laser is reasonably strong so that $|\mathbf{E}_0| \gg |\delta\mathbf{E}_{L,R}|$, the $LL$ term is subdominant because it is quadratic in the fluctuations. We will also suppress the term $V_0$ proportional to $|\mathbf{E}_0|^2$, which generates overall renormalization of various constants but does not involve a dynamical coupling between the laser

fluctuations and the slab. The important terms are

$$V_{RR} = x \int_0^\infty dk \left[ f_k e^{i\omega_0 t} a_k^\dagger + f_k^* e^{-i\omega_0 t} a_k \right], \qquad (11)$$

$$V_{RL} = x \int_0^\infty dk \left[ g_k e^{i\omega_0 t} b_k^\dagger + g_k^* e^{-i\omega_0 t} b_k \right], \qquad (12)$$

$$V(0) = i \int dk \frac{1}{k - k_0} (f_k e^{i\omega_0 t} a_k^\dagger - f_k^* e^{-i\omega_0 t} a_k) + \quad (13)$$

$$i \int dk \frac{1}{k + k_0} (g_k e^{i\omega_0 t} b_k^\dagger - g_k^* e^{-i\omega_0 t} b_k) \qquad (14)$$

where the operators $a_k$ create right-moving modes with momentum $k > 0$ to the right and no transverse momentum, the $b_k$ create left-moving modes with momentum $k > 0$ to the left and no transverse momentum, and the couplings $f_k, g_k \sim \chi_e \sqrt{|\mathbf{E}_0|}$ are enhanced by the presence of the laser.

Note that the $V_{RR}$ term represents a laser photon transmitting through the slab while the $V_{RL}$ represents reflection. In both cases, we see that the photon will pick up some information (become slightly entangled with) the slab position $x$. Thus, when we measure the photon state far from the slab, we affect a (non-projective or "weak") measurement in the $x$ basis.

To describe the measurement process, we can use a language familiar from scattering theory. We define "in" and "out" fields, which are Heisenberg-picture fields that take initial- or final-state boundary conditions on the photons and propagate them to finite time. As shown in Appendix B, the difference between the output and input equations is given as

$$\begin{aligned} &a_k^{\text{out}}(t) - a_k^{\text{in}}(t) \\ &= i \int_{t_0}^{t_f} dt' f_k e^{-i\Delta_k(t-t')} \left( x(t') + i \frac{1}{k - k_0} \right). \end{aligned} \quad (15)$$

where $\Delta_k = \omega_k - \omega_0$ is the difference ("detuning") between the mode's frequency $\omega_k$ and the laser drive frequency $\omega_0$. We then find a scattering relation which relates the out fields to the in fields. In this particular system, these relations take the form

$$Y_R^{\text{out}} - Y_R^{\text{in}} = \sqrt{2} f x, \qquad (16)$$

$$Y_L^{\text{out}} - Y_L^{\text{in}} = \sqrt{2} g x. \qquad (17)$$

Here, we defined $Y = (a - a^\dagger)/i\sqrt{2}$, called an optical quadrature variable, which we will discuss in great detail starting in Sec. III. The coefficients $f, g$ are the microscopic $f_k, g_k$ evaluated at $k = k_0$, i.e., at the frequency of the laser, and we can take them to be real (see discussion around Eq. B32). Equations like (16) are called input-output relations in measurement theory; they show how information from system being probed ($x$) is encoded as a change in the scattered light ($Y^{\text{out}} - Y^{\text{in}}$).

The essential use of the input-output relations is to find the outgoing photon field, $Y^{\text{out}}$, in terms of the state of the slab, any force acting on the slab that we might want

to sense, and any noise. To do this, we need the equations of motion for the slab itself. These are, in the Heisenberg picture,

$$\begin{aligned} \dot{x} &= p/m \\ \dot{p} &= -m\omega_m^2 x + f X_L^{\text{in}} + g X_R^{\text{in}} - i(f^2 + g^2)x + F^{\text{sig}}. \end{aligned} \quad (18)$$

These equations, together with the input-output relations for the light can be solved in the frequency domain to yield

$$\begin{aligned} Y_L^{\text{out}} = Y_L^{\text{in}} + \sqrt{2} g \chi_m \big( &- \sqrt{2} f X_R^{\text{in}} - \sqrt{2} g X_L^{\text{in}} + \\ &F^{\text{in}} + F_0 2\pi \delta(\Omega) \big) \end{aligned} \quad (19)$$

$$\begin{aligned} Y_R^{\text{out}} = Y_R^{\text{in}} + \sqrt{2} f \chi_m \big( &- \sqrt{2} f X_R^{\text{in}} - \sqrt{2} g X_L^{\text{in}} + \\ &F^{\text{in}} + F_0 2\pi \delta(\Omega) \big), \end{aligned} \quad (20)$$

where $F^{\text{in}} = F^{\text{sig}} + F^{\text{noise}}$. Our result (20) encodes a great deal of information about the detection problem. First, it shows how a signal of interest which couples to the slab — in this case, some external force $F^{\text{sig}}$, for example from some dark matter interaction — is imprinted onto the light field and can thus be measured. Second, it shows that the slab will experience *noise*: the input laser fluctuations $Y_{L,R}^{\text{in}}$ are quantum fields, and thus subject to fluctuations governed by their quantum state.

In the language of open quantum systems, the photons forming the $Y^{\text{in}}$, $Y^{\text{out}}$ fields are "baths"—systems with large numbers of degrees of freedom which all couple to the sensor element. In general, these baths may or may not be accessible to the experimentalist, and may or may not be thermal. In this example, we can prepare the input states and measure the outgoing states. One can incorporate other types of baths in the exact same way. For instance, the slab is coupled to some support structure, which is at $T > 0$ and exchanging phonons with the slab. This leads to thermal noise on the slab, which could be incorporated as a noisy force $F^{\text{noise}}$; this phononic bath is generally not under our control and cannot be measured.

In the rest of this section, we show how to use input-output models like this to compute detailed noise spectra for a detector, and how to use these to determine our ability to resolve a signal like $F^{\text{sig}}$ above the noise floors set by the various baths coupling to the detector.

### C. Input-output formalism

First, we summarize the input-output formalism [15, 17] and put it in its general context, which we will use repeatedly in the rest of the paper. Most of the features of the example in the previous section can be generalized to a wide class of detectors.

The input-output formalism works whenever we have bilinear couplings of the baths (including the readout system) to the detector, as in Eq. (11). In this case, we have

some set of Heisenberg-picture input and output operators $\mathcal{O}_i^{\mathrm{in}}(t)$, $\mathcal{O}_j^{\mathrm{out}}(t)$, which are linearly related to each other. In the frequency domain, we write this as

$$\mathcal{O}_i^{\mathrm{out}}(\nu) = \sum_j \chi_{ij}(\nu)\mathcal{O}_j^{\mathrm{in}}(\nu), \qquad (21)$$

where the $\chi_{ij}$ are variously referred to as susceptibilities, response functions, transfer functions, Green's functions, etc. These depend only on the detector itself and not the signal, and can be calculated in standard linear response or perturbation theory as in the previous section; we give many examples in what follows. In time domain, Eq. (21) gives the output operators as convolutions of the input with the response functions.

A signal of interest is encoded as a particular input, which we will denote with an $F$ although it will not always be a literal force: $\mathcal{O}^{\mathrm{in}} = F^{\mathrm{in}} = F^{\mathrm{sig}} + F^{\mathrm{noise}}$. The second term here reflects the fact that we may also have noise in this variable. For example, if we are trying to measure a force as in the previous section, we might have to include both the signal force of interest as well as thermal fluctuations. In the simplest case, we can imagine that the detector reads out one specific output variable $\mathcal{O}_{\mathrm{det}}^{\mathrm{out}}$. The signal is encoded on the detector output[2], using Eq. (21), as

$$\mathcal{O}_{\mathrm{det}}^{\mathrm{out}}(\nu) = \chi_{\mathrm{det},F}(\nu)F(\nu) + \sum_{j \neq F} \chi_{\mathrm{det},\mathrm{j}}(\nu)\mathcal{O}_j^{\mathrm{in}}(\nu). \quad (22)$$

In our optomechanical force sensing example above, $\mathcal{O}_{\mathrm{det}}^{\mathrm{out}} = Y^{\mathrm{out}}$ is the output laser phase and $F^{\mathrm{in}}$ is the literal mechanical force acting on the dielectric mirror. In a gravitational wave detector based on moving mirrors, the same identifications hold, with $F^{\mathrm{in}}(\nu) \sim mL^2\nu^2 h(\nu)$ now representing the effective "force" caused by a passing gravitational wave with strain profile $h(\nu)$. In a typical axion cavity experiment, again one might have $\mathcal{O}_{\mathrm{det}}^{\mathrm{out}} = Y^{\mathrm{out}}$ when reading out the phase of microwave fluctuations down a transmission line exiting the cavity, while the input "force" that drives the cavity field $F^{\mathrm{sig}} \sim g_{a\gamma\gamma}a(\nu)$, where $g_{a\gamma\gamma}$ is the axion-photon coupling and $a(\nu)$ is the axion field. We give the detailed expressions in the appropriate sections below.

In Eq. (22), the $\mathcal{O}_j^{\mathrm{in}}$, including the signal, are noisy. This equation reflects the fact that the detector output has both any potential signal as well as noise from various baths coupled to the detector. Classically, the $\mathcal{O}_j^{\mathrm{in}}$ would be modelled as random variables; quantum mechanically, they are operators. In most cases, we will assume that the noises are well-modelled as having zero mean $\langle \mathcal{O}_j^{\mathrm{in}} \rangle = 0$ and stationary $\langle \mathcal{O}_j^{\mathrm{in}}(t)\mathcal{O}_j^{\mathrm{in}}(t') \rangle = \langle \mathcal{O}_j^{\mathrm{in}}(t-t')\mathcal{O}_j^{\mathrm{in}}(0) \rangle$.

This latter assumption, in particular, means that we can talk about the *noise power spectral density* (PSD) $S_{ij}^{\mathrm{in}}(\nu)$. The noise PSD of a pair of operators $\mathcal{O}_i, \mathcal{O}_j$ is a frequency-domain function defined by the autocorrelation function

$$\langle \mathcal{O}_i(t)\mathcal{O}_j(t') \rangle = \int_{-\infty}^{\infty} d\nu\, e^{i\nu(t-t')}S_{ij}(\nu). \qquad (23)$$

The brackets here represent quantum-mechanical expectation values taken in specific states; this can include thermal states, the vacuum, or more complex states like squeezed states. A key objective in quantum measurement is to engineer particular states which reduce the overall noise of a system. Examples of these input noise correlations were given above, such as the input thermal white noise fluctuations $S_{FF}^{\mathrm{in}} \sim 4mk_{\mathrm{B}}T$ on the mirror or the vacuum fluctuations $S_{XX}^{\mathrm{in}}$, $S_{YY}^{\mathrm{in}}$ in the laser amplitude and phase.

In practice, the most useful way to compute power spectral densities is by using a result called the Wiener-Khinchin theorem,

$$2\pi\, S_{ij}(\nu)\, \delta(\nu - \nu') = \langle \mathcal{O}_i(\nu)\mathcal{O}_j^{\dagger}(\nu') \rangle. \qquad (24)$$

We will use this repeatedly to obtain PSDs from the equations of motions of various detectors. We review this result and record our Fourier conventions and several useful facts related to PSDs in Appendix A.

Because of the linear relationships (21), we can calculate the noise PSD of any output variables in terms of the noise PSDs of the input variables:

$$S_{ij}^{\mathrm{out}}(\nu) = \sum_{i'j'} \chi_{ii'}(\nu)\chi_{jj'}^*(\nu)S_{i'j'}^{\mathrm{in}}(\nu). \qquad (25)$$

In general, these noise PSDs can include cross-correlations between the different variables. In a given system, one assumes a model for the input noise power—for example, thermal or vacuum fluctuations—and then Eq. (25) gives a prediction for the noise in the data coming out of the detector.

To estimate the signal $F^{\mathrm{sig}}$ itself, one often forms an estimator

$$\begin{aligned}
F_E(\nu) &= \frac{\mathcal{O}_{\mathrm{det}}^{\mathrm{out}}(\nu)}{\chi_{\mathrm{det},F}(\nu)} \\
&= F^{\mathrm{sig}}(\nu) + F^{\mathrm{noise}}(\nu) + \sum_{j \neq F} \frac{\chi_{\mathrm{det},\mathrm{j}}(\nu)}{\chi_{\mathrm{det},F}(\nu)}\mathcal{O}_j^{\mathrm{in}}(\nu).
\end{aligned} \qquad (26)$$

With our noise assumptions, this estimator is unbiased. The noise PSD of the estimator can be calculated from the input noise model using Eq. (25).

The essential question is whether one can see a given signal of interest [the first term of Eq. (26)] above the noise [the rest of the terms in Eq. (26)]. To make this precise, we need to formulate a notion of signal-to-noise ratios. This requires specifying exactly how the detector output is processed, as we detail in the next section.

---

[2] One often talks about a measurement "referred to the input", e.g., in our slab example, we can talk about the data in units of the output phase of the light (by plotting $\mathcal{O}_{\mathrm{det}}^{\mathrm{out}} = Y^{\mathrm{out}}$) or in units of the input force (by plotting $F_E = Y^{\mathrm{out}}/\chi^{YF}$).

## D. Signal versus noise

In Eq. (22), the detector is continuously measuring $\mathcal{O}_{\mathrm{det}}^{\mathrm{out}}(t)$ and records this as a classical data stream. In practice, one wants to do hypothesis testing for the presence of such signals in some finite time window. The hypothesis we want to test is if the observed data can be explained simply by random noise, or if we instead need to invoke an additional signal to explain it.

To understand the basic idea, consider trying to determine if a continuous, monochromatic force

$$F^{\mathrm{sig}}(t) = F_s(\cos \omega_s t + \phi) \qquad (27)$$

is acting on a sensor. We monitor the output of the detector for some time $T_{\mathrm{int}}$, and obtain a time series of data of the force estimator $F_E(t)$ from $0 \leq t \leq T_{\mathrm{int}}$. We then look through the Fourier transform of this data for a monochromatic line at $\omega_s$; we can resolve such a line with signal-to-noise ratio

$$\mathrm{SNR} = \frac{F_s}{\sqrt{S_{FF}(\omega_s)/T_{\mathrm{int}}}}, \qquad (28)$$

where $S_{FF}(\nu)$ is the noise PSD for $F_E(\nu)$. We will give a more precise justification of this formula shortly. However, the essence of this formula is that the signal is building up coherently for the whole time $T_{\mathrm{int}}$, while the noise is adding up in a Brownian fashion, leading to the overall $\sqrt{T_{\mathrm{int}}}$ scaling. Note also that this is why noise PSDs come with units of (quantity of interest)$^2$/Hz, so that the denominator here has the right units to make the SNR dimensionless.

More generally, we will focus on two basic kinds of data analysis, for which the signal-to-noise is treated somewhat differently. The first is searches for signals of particular "shapes" (i.e., specific forms in time or frequency domain), of which a monochromatic signal is perhaps the simplest. The other is searches for signals which are intrinsically noisy and are best described as generating excess noise, rather than a deterministic shape in the data.

### 1. Matched filtering searches

First, consider a search for a transient signal with a specific shape. To get some intuition, consider a simple test of an impulsive force acting at time $t_0$, where $F^{\mathrm{sig}}(t) = \Delta p \delta(t - t_0)$, say from a particle colliding with the dielectric slab of the first section. We could construct the observable

$$I(\tau) = \int_0^\tau dt F_E(t) \qquad (29)$$

from the data $F_E(t)$, which estimates the impulse delivered to the sensor. In each window of time $\tau$, this observable will take different values due to fluctuations

in the noise, regardless of the presence of a signal. The noise yields a variance given as

$$\begin{aligned} \langle \Delta I^2(\tau) \rangle &= \int_0^\tau dt dt' \, \langle F_E(t) F_E(t') \rangle \\ &= \frac{2}{\pi} \int_{-\infty}^\infty d\nu \frac{\sin^2(\nu \tau/2)}{\nu^2} S_{FF}(\nu), \end{aligned} \qquad (30)$$

To test whether a "bump" from the impulsive signal has occurred, one wants to see that the measured impulse is much larger than would typically be explainable by this noise. We quantify this by a signal-to-noise ratio (SNR)

$$\mathrm{SNR} = \frac{I(\tau)}{\sqrt{\langle \Delta I^2(\tau) \rangle}}, \qquad (31)$$

A signal-to-noise of order $\sim 1$ is the minimum to see a signal; usually one requires at least $\sim 3 - 5$. In other words, this quantity gives the number of standard deviations the data is from the mean, assuming Gaussian distributed noise. In particular, if there *is* a signal in the window $\tau$, then obtaining an SNR of order 1 requires a signal $\Delta p_{\mathrm{sig}} \gtrsim \sqrt{\langle \Delta I^2(\tau) \rangle}$.

For signals with particular shape, it turns out we can do better than a naive estimate like (31). Consider a signal which comes from some family of known shapes, one of which could be $F^{\mathrm{sig}}(t) = F_0(t - t_0)$. For example, this could be an impulsive force $F_0(t - t_0) \sim \delta(t - t_0)$, or a waveform $F_0(t - t_0) \sim \sin(\omega(t)t)$ of a typical binary merger event in a gravitational wave detector, where $\omega(t)$ represents the time-varying frequency "chirping". It is better in general to filter the data in a particular way, using a procedure known as "matched filtering" or "template matching", where we scan the data for this particular shape [33, 34].

Consider a linear observable of the form

$$\mathcal{O}_f(t) = \int dt' f(t - t') F_E(t'), \qquad (32)$$

where $f(t)$ is called a filter. For example, in our impulsive signal example above, $f(t)$ is just a box function of width $\tau$. One can compute the variance of the observable $\mathcal{O}_f$ in the same way as above, and find the filter $f$ which optimizes the resulting signal-to-noise ratio. It turns out (see Appendix B of [35] for a proof) that the optimum filter is given by

$$f_{\mathrm{opt}}(\nu) = \frac{F_0(\nu)}{S_{FF}(\nu)}, \qquad (33)$$

where $F_0(t)$ is the signal shape of interest, in which case the SNR is

$$\mathrm{SNR}_{\mathrm{opt}} = \sqrt{\int_{-\infty}^\infty d\nu \frac{|F_0(\nu)|^2}{S_{FF}(\nu)}}. \qquad (34)$$

Intuitively, what this filter does is to scan the data for the spectral shape of the signal, but weight it such that low-noise frequency bins (those where $S_{FF}$ is smallest) are weighted the highest.

### 2. Excess power searches

The other primary case of interest will be signals which are persistent and noisy. Examples include stochastic gravitational wave backgrounds [36, 37] and stochastic signals from dark matter candidates like light axions (Sec. III C). In these cases, rather than scanning the data $\mathcal{O}_{\text{out}}^{\text{det}}$ for a particular signal shape, we want to look for *excess noise power*. By excess, we mean excess compared to the expected noise in the device. Note that this means we assume we have a detailed model of the detector noise, or an accurate method of noise subtraction. We closely parallel the treatment of [38–42].

Consider monitoring an output variable, say the estimator for a signal of interest $F_E(t)$, for some total time $T_{\text{int}}$. Continuing to assume that our noise is stationary, we can take the observed data and convert it into a measured noise PSD:

$$S^{\text{meas}}(\nu) = \int_0^{T_{\text{int}}} dt \, e^{-i\nu t} \langle F_E(t) F_E(0) \rangle. \qquad (35)$$

The expectation value here can be obtained by appealing to ergodicity: we take all the pairs of data points $F_E(t + t'), F_E(t')$ separated by $t$ and then average over $t'$ to form the expectation value. Because our Fourier transform is "windowed", we do not get independent measurements of every frequency $\nu$, but rather independent measurements of the central value of the PSD in each frequency bin of bandwidth $\Delta\nu_{\text{bin}} = 1/2T_{\text{int}}$.

The value of $S^{\text{meas}}(\nu)$ at a given frequency is itself a random variable: in each given experimental run, one will measure different values. Assuming a good noise model, $S^{\text{meas}}(\nu)$ will have mean $\langle S^{\text{meas}}(\nu) \rangle_0 = S^{\text{exp}}(\nu)$ in the absence of a signal, where $\langle \cdot \rangle_0$ denotes averaging under the no-signal hypothesis. Assuming that our observable of interest is Gaussian distributed,[3] then so is $S^{\text{meas}}(\nu)$, with the standard error given $n$ measurements of the PSD [43]

$$(\Delta S^{\text{meas}})^2 = \frac{2}{n-1}(S^{\text{exp}})^2. \qquad (36)$$

Thus the excess power

$$S^{\text{excess}}(\nu) \equiv S^{\text{meas}}(\nu) - S^{\text{exp}}(\nu) \qquad (37)$$

is a Gaussian random variable with mean zero (under the null hypothesis) and variance $(S^{\text{exp}})^2/(n-1)$.

Consider a window of width $\Delta\nu_c \gg \Delta\nu_{\text{bin}}$. We construct the power excess $S^{\text{excess}}(\nu_i)$, where $i$ is the center of

---

[3] Note that we may alternatively write the measured PSD as $\text{Var}\left[ F_E(\nu) \right] = 2\pi T_{\text{int}} S^{\text{meas}}(\nu)$, which is the finite-time analog of the Wiener-Khinchin theorem of Eq. (24). This shows us that the measured PSD encodes the sample variance of the random variable $F_E(\nu)$; in other words, it is an unbiased estimator of the variance in $F_E(\nu)$, and is itself a Gaussian random variable, with variance given by Eq. (36).

the $\Delta\nu_c$ window. In this window, we have $n = \Delta\nu_c/\Delta\nu_{\text{bin}}$ independent measurements, and so the signal-to-noise ratio in this bin is, using (36) and (37),

$$\text{SNR}_i^2 = T_{\text{int}}\Delta\nu_c \left( \frac{S^{\text{excess}}(\nu_i)}{S^{\text{exp}}(\nu_i)} \right)^2, \qquad (38)$$

where we have assumed $n \gg 1$, i.e., $T_{\text{int}} \gg 1/\Delta\nu_c$.

Suppose we want to look for a potential signal manifesting as excess power with some expected bandwidth $\Delta\omega_s$—for example, an axion dark matter signal with $\Delta\omega_s \approx 10^{-6} m_a$ (see Sec. III C). Within the bandwidth $\Delta\omega_s$ of the purported signal, we sum the individual SNRs in quadrature to obtain the total SNR

$$\text{SNR}^2 = \sum_i \text{SNR}_i^2 \qquad (39)$$

$$= T_{\text{int}} \int_{\Delta\omega_s} d\nu \left( \frac{S^{\text{excess}}(\nu)}{S^{\text{exp}}(\nu)} \right)^2. \qquad (40)$$

In the second line, we took the formal limit $\Delta\nu_c \to 0$ while keeping $T_{\text{int}}\Delta\nu_c$ fixed.

We often encounter limits in which one of the signal PSD or detector PSD is very narrow compared to the other. In these cases, the expression for the SNR simplifies. If the signal has a bandwidth much smaller than the detector's bandwidth, then we may approximate $S_{\text{exp}}$ as taking the constant value $S_{\text{exp}}(\omega_s)$ over the entire support of the visibility. In this case, the optimal SNR is

$$\text{SNR} \approx \sqrt{2T_{\text{int}}\Delta\omega_s} \frac{\bar{S}^{\text{excess}}}{S^{\text{exp}}(\omega_s)}, \qquad (41)$$

where $\Delta\omega_s(\bar{S}^{\text{excess}})^2 \equiv \int d\nu (S^{\text{excess}})^2$. In a simple parametrization, one could model this as a bandwidth $\Delta\omega_s = \omega_s/Q_s$ where $\omega_s$ is the central frequency of the signal and $Q_s$ is its quality factor. If instead the detector has a much smaller bandwidth than the signal, then the signal PSD takes an approximately constant value over the full detector bandwidth. In this case, the SNR is

$$\text{SNR} \approx \sqrt{2T_{\text{int}}\Delta\omega_0} \frac{S^{\text{excess}}(\omega_0)}{\bar{S}^{\text{exp}}}, \qquad (42)$$

where $\Delta\omega_0(\bar{S}^{\text{exp}})^{-2} \equiv \int d\nu (S^{\text{exp}})^{-2}$ is the detector's bandwidth. In a simple example like a resonant cavity, $\Delta\omega_0 = \omega_0/Q_0$ where $\omega_0$ is the resonance frequency and $Q_0$ is the cavity quality factor.

### 3. Scan rates

For narrow signals and/or detectors, the SNRs we have derived depend strongly on the central (resonant) frequencies. Furthermore, the signal spectrum is usually set by some characteristic frequency, for instance the dark matter mass, which is an unknown parameter. In order to be sensitive to a wide range of signal frequencies, we would like to tune the resonant frequency in small steps.

The *scan rate* is the statistical measure determining how quickly we may carry out this tuning while still achieving a certain signal-to-noise ratio [44, 45]. We now derive an expression for the scan rate, closely following the logic of [46], although we consider more general signal PSDs.

Consider the optimal signal-to-noise ratio $\mathrm{SNR}(\omega_0, \omega_s)$ for a signal at frequency $\omega_s$ in a detector of resonant frequency $\omega_0$. If we tune $\omega_0$ in steps of $\Delta\omega$, then the integrated signal-to-noise ratio $\mathrm{SNR_I}$ is given by summing the individual SNRs in quadrature

$$\mathrm{SNR_I^2} = \frac{1}{\Delta\omega} \sum_n \mathrm{SNR}^2(\omega_0 + n\Delta\omega, \omega_s) \cdot \Delta\omega \qquad (43)$$

$$= \frac{T_{\mathrm{int}}}{\Delta\omega} \sum_n \int d\nu \left( \frac{S^{\mathrm{excess}}(\nu; \omega_s)}{S^{\mathrm{exp}}(\nu; \omega_0 + n\Delta\omega)} \right)^2 \cdot \Delta\omega, \qquad (44)$$

where we sum over an integer $n$ of such tunings and we have made explicit the dependence on the signal and detector frequency. We now take the limit where both the integration time and step size go to zero, $T_{\mathrm{int}}, \Delta\omega \to 0$, while keeping the ratio

$$\frac{\Delta\omega}{T_{\mathrm{int}}} = 2\pi\mathcal{R} \qquad (45)$$

fixed at the rate required to achieve some desired value for the integrated SNR, labelled $\mathcal{S}$; we insert the factor of $2\pi$ to follow standard convention, in which $\mathcal{R}$ has units of linear frequency (not angular frequency) per unit time. In this limit, we may express the scan rate as

$$\mathcal{R} = \frac{1}{\mathcal{S}^2} \int_{-\infty}^{\infty} \frac{d\omega}{2\pi} \int_{-\infty}^{\infty} d\nu \left( \frac{S^{\mathrm{excess}}(\nu; \omega_s)}{S^{\mathrm{exp}}(\nu; \omega_0 + \omega)} \right)^2, \qquad (46)$$

where we have extended the limits of integration to infinity by making the visibility a windowed function around the signal frequency.

This general expression may be simplified when the signal and detector PSDs satisfy certain properties. To this end, we make two observations. Firstly, we note that the signal PSD $S^{\mathrm{excess}}(\nu; \omega_s)$ appearing in the visibility's numerator often depends only on the difference between $\nu$ and $\omega_s$, i.e. changing the signal frequency only effects a translation of the PSD so that $S^{\mathrm{excess}}(\nu; \omega_s) = S^{\mathrm{excess}}(\nu - \omega_s)$. Secondly, the same is true for the detector PSD: $S^{\mathrm{exp}}(\nu; \omega_0) = S^{\mathrm{exp}}(\nu - \omega_0)$. Under these conditions, the scan rate becomes

$$\mathcal{R} = \frac{1}{\mathcal{S}^2} \int_{-\infty}^{\infty} \frac{d\omega}{2\pi} \int_{-\infty}^{\infty} d\nu \left[ \frac{S^{\mathrm{excess}}(\nu + \omega)}{S^{\mathrm{exp}}(\nu)} \right]^2, \qquad (47)$$

which is notably independent of the signal frequency, and so gives a measure of the intrinsic efficiency of the detector[4].

---

[4] This expression also shows that one may consider the scan rate as either an integral of the squared SNR over detector frequency or over signal frequency.

With our formula for the scan rate at hand, we may derive expressions for the scan rate in certain simplifying limits. For instance, consider the signal PSD to be peaked at a frequency $\omega_s$ with a bandwidth much narrower than the detector. An example of this case is a cavity with a quality factor less than that of dark matter. Here, the scan rate reduces to

$$\mathcal{R} = \frac{\Delta\omega_s}{\mathcal{S}^2} \int_{-\infty}^{\infty} \frac{d\nu}{2\pi} \left[ \frac{\bar{S}^{\mathrm{excess}}}{S^{\mathrm{exp}}(\nu)} \right]^2. \qquad (48)$$

## III. ELECTROMAGNETIC CAVITIES

An electromagnetic cavity supports a spectrum of normal modes. In this section, we consider the measurement of one particular mode onto which a physics signal is imprinted. We will see how the quantum mechanics of the cavity and readout systems impact the noise spectrum and signal sensitivity.

Inside a cavity, the electromagnetic potential can be expanded as

$$A_\mu(\mathbf{x}) = \sum_{\mathbf{p}, r} \epsilon_{r,\mu}^*(\mathbf{p}) u_{\mathbf{p}}^*(\mathbf{x}) a_{\mathbf{p},r} + \epsilon_{r,\mu}(\mathbf{p}) u_{\mathbf{p}}(\mathbf{x}) a_{\mathbf{p},r}^\dagger. \quad (49)$$

This is expressed in the Schrödinger picture. Here $\epsilon_r(\mathbf{p}), r = 1, 2$ are a complete set of polarization vectors, and the $u_{\mathbf{p}}(x)$ form a complete, discrete set of modes:

$$\int_{\mathrm{cav}} d^3\mathbf{x}\, u_{\mathbf{p}}(\mathbf{x}) u_{\mathbf{p}'}^*(\mathbf{x}) = \frac{\delta_{\mathbf{p}\mathbf{p}'}}{2\omega_{\mathbf{p}}}, \quad \epsilon_r^*(\mathbf{p}) \cdot \epsilon_{r'}(\mathbf{p}) = \delta_{rr'}. \quad (50)$$

The modes are normalized so that the energy density is

$$H_{\mathrm{cav}} = \frac{1}{2} \int_{\mathrm{cav}} d^3\mathbf{x} \left[ \mathbf{E}^2 + \mathbf{B}^2 \right] = \sum_{\mathbf{p}, r} \omega_{\mathbf{p}} a_{\mathbf{p},r}^\dagger a_{\mathbf{p},r} \qquad (51)$$

plus the usual infinite zero-point energy.

In a typical example, we will assume that we are reading out a specific cavity mode $a = a_{\mathbf{p}_0, r_0}$, although the generalization to a multi-mode cavity is straightforward. This can be the fundamental mode or some other convenient mode. To read out a single mode means that we can resolve frequencies at the level of the *free spectral range* of the cavity, which is the spacing between the modes. This single mode has the simple harmonic oscillator Hamiltonian,

$$H_{\mathrm{det}} = \omega_c a^\dagger a = \frac{\omega_c}{2}(X^2 + Y^2), \qquad (52)$$

with $\omega_c = \omega_{\mathbf{p}_0, r_0}$ denoting the relevant cavity frequency. We have defined what are known as the quadrature operators

$$X = \frac{a + a^\dagger}{\sqrt{2}}, \quad Y = -i\frac{a - a^\dagger}{\sqrt{2}}, \qquad (53)$$

which are dimensionless versions of the position and momentum of the oscillator. In particular, they are canonically conjugate $[X, Y] = i$.

Throughout this section, we assume that the measurement of the cavity proceeds via phase sensitive amplification of a single quadrature of the cavity. This implies that we add no noise in the amplification process [47].

## A. Input-output theory for a cavity mode

To use a cavity mode as a detector, we need to couple it to some readout system. This could be, for example, a simple electromagnetic transmission line which is inserted slightly into the cavity. The cavity mode interacts with the line like an antenna: the cavity field can drive currents in the wire, while conversely the currents in the wire can produce cavity field excitations. See Fig. 4. This can be handled with the input-output formalism as described above [17, 42].

The input-output treatment of this system closely parallels the treatment of Sec. II B. In general, we have a bath associated to the measurement system (the transmission line), as well as a bath associated to intrinsic internal losses to the cavity (e.g., moving charges in the walls which can emit and absorb radiation). Denoting the relevant input fields as $X_m^{\rm in}$, $X_\ell^{\rm in}$, respectively, and similarly for $Y$, these contribute to the cavity mode equations of motion as

$$\dot{X} = \omega_c Y - \frac{\kappa}{2} X + \sqrt{\kappa_m} X_m^{\rm in} + \sqrt{\kappa_\ell} X_\ell^{\rm in} + F_X^{\rm in}$$
$$\dot{Y} = -\omega_c X - \frac{\kappa}{2} Y + \sqrt{\kappa_m} Y_m^{\rm in} + \sqrt{\kappa_\ell} Y_\ell^{\rm in} + F_Y^{\rm in}, \quad (54)$$

where the total cavity decay rate is

$$\kappa = \kappa_m + \kappa_\ell. \quad (55)$$

The final terms $F_{X,Y}$ represent any external driving force, for example one caused by a background of axions, as discussed in the next section.

Physically, we assume that we have access to the measurement fields $X_m^{\rm in,out}$, but not the intrinsic loss fields $X_\ell^{\rm in,out}$. We can write equivalent equations of motion for $\dot{X}$ and $\dot{Y}$ in terms of the output fields, again following the logic of Sec. II B, and find input-output relations for the measurement port:

$$X_m^{\rm out} = X_m^{\rm in} - \sqrt{\kappa_m} X$$
$$Y_m^{\rm out} = Y_m^{\rm in} - \sqrt{\kappa_m} Y. \quad (56)$$

In the frame co-rotating with the cavity frequency $\omega_c$, we can easily solve (54) in the frequency domain to get $X(\nu), Y(\nu)$. Inserting these into (56), we get simple frequency domain relations:

$$X_m^{\rm out} = \chi_{mm} X_m^{\rm in} + \chi_{m\ell} X_\ell^{\rm in} + \chi_{mF} F_X^{\rm in}$$
$$Y_m^{\rm out} = \chi_{mm} Y_m^{\rm in} + \chi_{m\ell} Y_\ell^{\rm in} + \chi_{mF} F_Y^{\rm in}, \quad (57)$$

where the susceptibilities are

$$\chi_{mm}(\nu) = 1 - \kappa_m \chi_c(\nu)$$
$$\chi_{m\ell}(\nu) = -\sqrt{\kappa_m \kappa_\ell} \chi_c(\nu) \quad (58)$$
$$\chi_{mF}(\nu) = -\sqrt{\kappa_m} \chi_c(\nu),$$

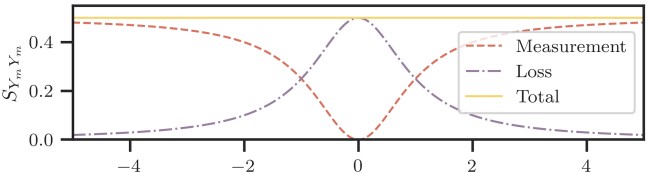

FIG. 3. **A typical cavity noise power spectrum:** Here we show how the noise from the measurement port ($S_{Y_m Y_m}^{\rm in}$) and loss port ($S_{Y_\ell Y_\ell}^{\rm in}$) add to form the full measured noise PSD of a homodyne cavity readout.

in terms of the basic cavity susceptibility

$$\chi_c(\nu) = \frac{1}{-i\nu + \kappa/2}. \quad (59)$$

Eq. (57) expresses the way in which a signal of interest—$F_{X,Y}^{\rm in}$—is imprinted on the field $X_m^{\rm out}, Y_m^{\rm out}$ exiting the cavity, which we can directly measure. Note that $|\chi_{mm}|^2 + |\chi_{m\ell}|^2 = 1$, a consequence of unitarity.

Suppose, for example, that we monitor the output phase quadrature $Y_m^{\rm out}$. Its noise spectral density can be computed directly from Eqs. (57) and (24),

$$S_{Y_m Y_m}^{\rm out} = |\chi_{mm}|^2 S_{Y_m Y_m}^{\rm in} + |\chi_{m\ell}|^2 S_{Y_\ell Y_\ell}^{\rm in}, \quad (60)$$

when there is no external force $F^{\rm in} = 0$. Here, the measured noise PSD is expressed in terms of the input noise PSDs, for which we need a model. For example, suppose that both the transmission line modes and cavity walls are in thermal equilibrium with some bath at temperature $T$. Then (see Appendix A 2)

$$S_{Y_m Y_m}^{\rm in}(\nu) = S_{Y_\ell Y_\ell}^{\rm in}(\nu) = n_{\rm th}(\nu) + \frac{1}{2}, \quad (61)$$

which is the sum of a thermal occupancy contribution $n_{\rm th}(\nu)$ and a frequency-independent constant $1/2$ arising from quantum vacuum fluctuations. The thermal contribution is given by the Bose-Einstein distribution

$$n_{\rm th}(\nu) = \frac{1}{e^{(\nu + \omega_c)/k_{\rm B} T} - 1}. \quad (62)$$

A typical dilution fridge can achieve temperatures $T_{\rm dil} \sim$ 10 mK. At frequencies $\nu \gg k_{\rm B} T_{\rm dil} \sim 1.3$ GHz, the thermal occupancy is near zero, and the PSD is dominated by the vacuum contribution. In this limit, the output noise power is simply

$$S_{Y_m Y_m}^{\rm out}(\nu) \xrightarrow{T \to 0} \frac{1}{2} |\chi_{mm}(\nu)|^2 + \frac{1}{2} |\chi_{m\ell}(\nu)|^2 = \frac{1}{2}, \quad (63)$$

or rather $\hbar/2$ in non-natural units. We plot these two noise contributions in Fig. 3. This is the limit in which the detector noise is entirely due to vacuum fluctuations, sometimes called the "Standard Quantum Limit" in this context, as discussed in Sec. II A. In Sec. V, we will discuss how more sophisticated input states, like squeezed vacuum states, affect this power spectrum.

If we wish to estimate the size of the force from its impact on the output phase quadrature, say, then from the general discussion of estimators around equation (26) and the particular form of (57), we see that the estimator of the $Y$-force $F_{Y,\mathrm{E}}$ is

$$F_{Y,\mathrm{E}} = \frac{Y_m^{\mathrm{out}}}{\chi_{mF}}, \qquad (64)$$

with associated noise given by

$$S_{FF} = \frac{S_{Y_m Y_m}^{\mathrm{out}}}{|\chi_{mF}|^2}. \qquad (65)$$

This will be the basic result we need to express sensitivities of various cavities to forces of interest, such as those arising from axion dark matter.

## B. Cavities as "particle" detectors

A prominent use of cavities in modern fundamental physics is in searches for new particles that interact directly with the electromagnetic field, including axions, dark photons, and a variety of other axion-like particles. These searches can proceed either by looking for an ambient source of these particles (for example, if they make up a component of the dark matter [48–56]), or by trying to create and then measure them directly within the laboratory [57–59].

Generally speaking, cavity modes with frequency $\omega_c$ are ideally suited for searches for new fields of mass $m \approx \omega_c$, in which case the field can be resonantly detected. In practice, this means that cavities are most useful for light fields; to date most activity has focused on radio-to-microwave cavities ($\omega_c \sim 100$ MHz $- 10$ GHz $\sim 10^{-6} - 10^{-4}$ eV). In this regime, the field excitations are long wavelength and do not really behave like particles, as discussed in App. F.

### 1. Axion-cavity coupling

Axions were originally proposed as a solution to the strong CP problem [60–63]. The relic density of axions was calculated soon after [64–66], suggesting that axions could make a viable dark matter candidate. The axion can thus solve two problems at once, so it is widely considered to be one of the best motivated dark matter candidates.

The axion is a real pseudoscalar field $a = a(\mathbf{x}, t)$, which couples to electromagnetism via

$$V = \frac{g_{a\gamma\gamma}}{8} \int d^3\mathbf{x}\, a\, F_{\mu\nu}\tilde{F}^{\mu\nu} = g_{a\gamma\gamma} \int d^3\mathbf{x}\, a\, \mathbf{E} \cdot \mathbf{B}, \quad (66)$$

where $g_{a\gamma\gamma}$ is a coupling constant with dimensions of 1/mass which parametrizes the strength of the interaction. This is the lowest-order coupling; there are additional higher order corrections [67, 68]. Here $\tilde{F}_{\mu\nu} =$

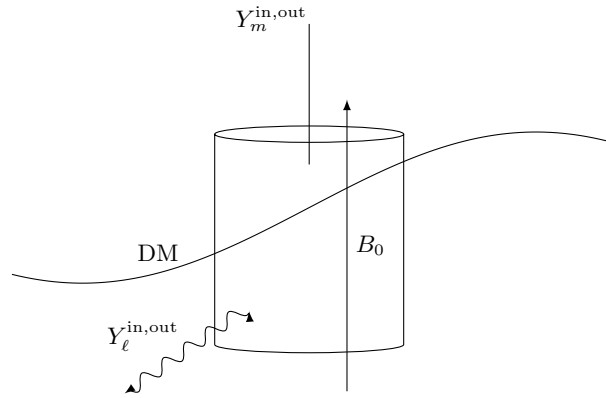

FIG. 4. **Axion cavity haloscope:** Many experimental searches for light axion and axion-like DM candidates utilize large microwave cavities, with tunable resonance frequencies, submerged in large static magnetic fields $B_0$. Axion-like DM, which has wavelength much longer than the cavity, would cause a displacement to the cavity quadratures which can be detected by measuring $Y_m^{\mathrm{out}}$. The cavity has intrinsic loss, modelled as a thermal bath with input/output fields $Y_\ell^{\mathrm{in,out}}$, and is readout with input/output fields $Y_m^{\mathrm{in,out}}$.

$\epsilon_{\mu\nu\rho\sigma}F^{\rho\sigma}$ is the "dual" field strength tensor and the electric and magnetic fields are, as usual, $E_i = F_{0i}, B_i = \epsilon_{ijk}F^{jk}$.

The coupling (66) allows the axion to convert into a pair of photons. To generate a strong coupling, many experiments use an external magnetic field $\mathbf{B}_0$, say along the $z$-axis $\mathbf{B}_0 = B_0\hat{\mathbf{z}}$, which leads to linear conversion between the axion and the electric field:

$$V = g_{a\gamma\gamma}B_0 \int d^3\mathbf{x}\, a\, \delta E_z. \qquad (67)$$

Here, $\delta\mathbf{E}$ represents small fluctuations in the electric field around the background $\mathbf{B}_0$. We are going to treat the axion as an external field, so even in the Schrodinger picture we expand it into time-dependent modes

$$a(\mathbf{x}, t) = \int d^3\mathbf{k}\, v_{\mathbf{k}}^*(\mathbf{x})e^{-i\omega_{\mathbf{k}}t}b_{\mathbf{k}} + v_{\mathbf{k}}(\mathbf{x})e^{i\omega_{\mathbf{k}}t}b_{\mathbf{k}}^\dagger. \quad (68)$$

Unlike the electromagnetic field inside the cavity, the axion is freely propagating without boundary conditions, and thus is expanded into a continuum of modes. The mode functions are

$$v_{\mathbf{k}}(\mathbf{x}) = \frac{e^{i\mathbf{k}\cdot\mathbf{x}}}{\sqrt{2\omega_{\mathbf{k}}(2\pi)^3}}, \quad \omega_{\mathbf{k}} = \sqrt{\mathbf{k}^2 + m_a^2} \qquad (69)$$

where $m_a$ is the axion mass. Here, as discussed above, we are going to mainly view the axion as a classical, time-dependent external field, so the $b, b^\dagger$ are just c-numbers which in general are drawn from a classical random probability distribution. Unless we are sensitive to effects from the axion vacuum fluctuations (which we are generally not), this approximation is extremely good [69].

Inserting the mode expansions (49) and (68) into (67), we can isolate the interaction Hamiltonian between a given cavity mode $a = a_{\mathbf{p}_0,r_0}$ and the entire axion field:

$$V_0(t) = F_Y(t)X + F_X(t)Y, \qquad (70)$$

where $X, Y$ are the cavity quadratures,

$$F_Y = g_{a\gamma\gamma}B_0 \int d^3\mathbf{k}\, \tilde{O}_\mathbf{k} e^{-i\omega_k t} b_\mathbf{k} + h.c., \qquad (71)$$

and we defined the "overlaps" of the axion and cavity mode functions

$$\tilde{O}_\mathbf{k} = \sqrt{2}\omega_{\mathbf{p}_0} \int_{\text{cav}} d^3\mathbf{x}\, v_\mathbf{k}^*(\mathbf{x})\text{Im}\left[\epsilon_{r_0,z}(\mathbf{p}_0)u_{\mathbf{p}_0,r_0}(\mathbf{x})\right]. \quad (72)$$

A similar expression holds for $F_Y$. Here, $\epsilon_{r_0,z}(\mathbf{p})$ is the component of the cavity mode polarization vector along the $z$ axis.

In the search for axion dark matter, a particularly nice simplification occurs because the axion's wavelength $\lambda \sim 10^3/\omega$ due to the dark matter velocity $v = 10^{-3}$ ($\approx 300$ km/s), so the axion's mode functions $v_\mathbf{k}(\mathbf{x})$ are approximately constant over the cavity volume. Thus we can approximate $a(\mathbf{x}, t) \approx a(0, t) =: a(t)$ by its value at the origin, and we can reduce (71) to

$$F_X(t) \approx g_{a\gamma\gamma}B_0 \sqrt{\frac{V\omega_{\mathbf{p}_0}}{2}} C \int d^3\mathbf{k}\, v_\mathbf{k}^*(0) e^{-i\omega_k t} b_\mathbf{k} + \text{h.c.}$$
$$= g_a\, C\, a(t). \qquad (73)$$

Here, $C$ is a dimensionless form factor parametrizing the overlap of the external magnetic field and the cavity mode,

$$C = \sqrt{\frac{2\omega_{\mathbf{p}_0}}{V}} \int_{\text{cav}} d^3\mathbf{x}\, \text{Im}[u_\mathbf{p}(\mathbf{x})\,\hat{\mathbf{z}} \cdot \epsilon_{r_0}(\mathbf{p}_0)], \qquad (74)$$

and

$$g_a := g_{a\gamma\gamma}B_0 \sqrt{\frac{V\omega_{\mathbf{p}_0}}{2}} \qquad (75)$$

is a dimensionless constant.

The axion signal is imprinted on the cavity detector output following the general input-output calculations given above. The axion "force" in Eq. (71) is taken as an input field $F_{X,Y} = F_{X,Y}^{\text{sig}}$. This is then imprinted on the cavity measurement port output fields $X_m^{\text{out}}, Y_m^{\text{out}}$ using the basic input-output relation Eq. (21), which in this case reduces to Eq. (57). One can form an estimator $F = F_E$ in the usual way, and talk about noise referred to the axion signal, or just work with the output phase quadratures $X_m^{\text{out}}, Y_m^{\text{out}}$ directly.

To calculate a signal-to-noise ratio and axion coupling sensitivity, we need to specify the detailed cavity parameters as well as the model for the axion signal $F^{\text{sig}}$. In Sec. III C we show how this works for cavity searches for axion dark matter. We first briefly show how to generalize this discussion of axions to similar fields, such as the dark photon.

### 2. Dark photon-cavity coupling

Another popular contender for a new degree of freedom is the so-called "dark photon". Unlike the axion, the existence of a dark photon would not alleviate the CP problem, but it is a viable dark matter candidate [70].

The dark photon terminology is sometimes used to mean a variety of things, but the canonical example is a vector field $A'_\mu = A'_\mu(\mathbf{x}, t)$ which couples to electromagnetism through a kinetic mixing effect [71]. This is encapsulated in an interaction Hamiltonian

$$V = \chi m_{A'}^2 \int d^3\mathbf{x}\, A^\mu A'_\mu, \qquad (76)$$

where $A_\mu$ is the usual photon of electromagnetism. Unlike the electromagnetic photon, the dark photon has non-zero mass $m_{A'}$. Thus there is a two-dimensional parameter space of these models, defined by the dimensionless mixing parameter $\chi \ll 1$ and mass $m_{A'}$. Note that the coupling to electromagnetism is proportional to the product $\chi m_{A'}^2$, so in the limit that the dark photon becomes massless $m_{A'} \to 0$, it decouples from the standard model.

The interaction (76) allows a photon to convert into a dark photon and vice versa; unlike the axion coupling (67), this does not require an external electromagnetic field. It also means that any electrically charged matter, say with charge $q$, will interact with the dark photon with an effective charged $\chi q \ll q$. Thus one can search for these in a variety of ways [72, 73]. In particular, one can use electromagnetic cavities as the detector, without the need for an external magnetic field.

Following the same steps as in Sec. III B 1, we can derive the response of a cavity mode to the dark photon field. The dark photon can be expanded into modes,

$$A'_\mu(\mathbf{x}, t) = \sum_{s=1,2,3} \int d^3\mathbf{k}\, \epsilon_{s,\mu}^*(\mathbf{k})v_\mathbf{k}^*(\mathbf{x})e^{-i\omega_\mathbf{k} t} b_{s,\mathbf{k}} + \text{h.c.} \qquad (77)$$

This is analogous to the axion case [Eq. (68)], except that we have to include polarization vectors $\epsilon_\mu^s$; there are three since the field is massive. The mode functions $v_\mathbf{k}(\mathbf{x})$ are again given by Eq. (69), with $m_a$ replaced by $m_{A'}$. Inserting this expansion and the cavity mode expansion (49) into (76), we can write the interaction $V_0(t)$ with the cavity mode of interest:

$$V_0(t) = F_Y(t)X + F_X(t)Y, \qquad (78)$$

Here, the external "force" on the cavity mode is

$$F_X = \sqrt{2}\chi m_{A'}^2 \sum_s \int d^3\mathbf{k} \int_{\text{cav}} d^3\mathbf{x}\, e^{-i\omega_k t} v_\mathbf{k}^*(\mathbf{x})$$
$$\times \epsilon_s^*(\mathbf{k}) \cdot \text{Re}\left[\epsilon_{r_0}(\mathbf{p_0})u_{\mathbf{p}_0,r_0}(\mathbf{x})\right] b_{s,\mathbf{k}} + \text{h.c.} \qquad (79)$$

and similarly for $F_Y$.

To get simple expressions, we first assume that the DM is randomly polarised, so that we may write the mode of

polarisation $s$ as $b_{s,\mathbf{k}} = b_{\mathbf{k}}/3$. This allows us to express the sum over the inner product of polarisation vectors as $\sum_s \epsilon_s^*(\mathbf{k}) \cdot \epsilon_{r_0}(\mathbf{p_0}) = \sum_s \epsilon_s^*(\mathbf{k}) \cdot \epsilon_{r_0}^*(\mathbf{p_0}) = 1$. Next, similar to axion DM, we can approximate the dark photon as homogeneous across the cavity. We can thus rewrite equation (80) as

$$
\begin{aligned}
F_X(t) &= \chi \sqrt{\frac{V}{6\omega_{\mathbf{p_0}}}} \, m_{A'}^2 C \int \frac{d^3\mathbf{k}}{\sqrt{2\omega_{\mathbf{k}}}} e^{-i\omega_{\mathbf{k}}t} b_{\mathbf{k}} + \text{h.c.} \\
&= g_{A'} C A'(t),
\end{aligned} \tag{80}
$$

where we have defined $A'(t) \equiv \sqrt{A'(0,t) \cdot A'(0,t)}$. The dimensionless form factor is now

$$
C = \sqrt{\frac{2\omega_{\mathbf{p_0}}}{V}} \int_{\text{cav}} d^3\mathbf{x} \, \text{Re}[u_{\mathbf{p}}(\mathbf{x})], \tag{81}
$$

and the dimensionless cavity-dark photon coupling constant is

$$
g_{A'} := \chi \sqrt{\frac{V}{6\omega_{\mathbf{p_0}}}} \, m_{A'}^2. \tag{82}
$$

### C. Dark matter searches with cavities

Searches for light axion and axion-like dark matter candidates with cavities are predicated on the hypothesis that the DM candidate makes up some appreciable fraction $f$ of the total dark matter density $\rho_{\text{DM}} = 0.3 \text{ GeV/cm}^3$, and that the mass of the field $m \lesssim 10 \text{ eV}$ [74]. In this regime, the basic picture is that we have a semi-coherent background of long-wavelength dark matter pervading the laboratory. In typical exclusion plots such as Fig. 6, the baseline assumption is that $f = 1$, i.e., the entirety of the dark matter is comprised of a single field. For $f < 1$, the bounds on the coupling become proportionally weaker.

Typically, the DM signal is modelled as a superposition of many waves, leading to a signal peaked around an (unknown) carrier frequency $\omega_s$ (where $\omega_s = m_a$ or $\omega_s = m_{A'}$ for the axion and dark photon, respectively), with random phase and linewidth $\gamma_s = \omega_s/Q_s$, where the quality factor $Q_s \approx 10^6$. For a review, see Appendix F. More precisely, this means that we have a noise power spectral density for the dark matter itself. The output noise PSD for, say, the measured phase quadrature is

$$
S_{Y_m Y_m}^{\text{out}} = \sum_{ij} \chi_{Y_m,i} \chi_{Y_m,j} S_{ij}, \tag{83}
$$

where the sum over $i,j$ includes all the input operators including the DM signal $F_{X,Y}$. In the simplest case where we assume uncorrelated input noise (i.e., no squeezing), we can write, in the frame co-rotating with the cavity mode,

$$
S_{Y_m Y_m}^{\text{out}} = |\chi_{mm}|^2 S_{Y_m Y_m}^{\text{in}} + |\chi_{m\ell}|^2 S_{Y_\ell Y_\ell}^{\text{in}} + |\chi_{mF}|^2 S_{F_Y F_Y}^{\text{in}}. \tag{84}
$$

The first two terms here are the input noise, while the last term represents the noisy dark matter signal. We have written the DM signal as an input force; this can be converted to an explicit dependence on the dark matter's PSD, using (73) or (80):

$$
S_{F_Y F_Y}^{\text{in}} = g_\chi^2 C^2 S_{\chi\chi}^{\text{sig}}. \tag{85}
$$

Here $\chi_{mF}$ is given in Eq. (58), and we give the expected distribution $S_{\chi\chi}$ for either $\chi = a$ (axion) or $\chi = A'$ (dark photon) dark matter in Eq. (F5). The couplings $g_\chi$ and form factors $C$ are given above.

Following the general discussion of Sec. II D 2, the observable we wish to estimate is the "force" PSD $S_{F_Y F_Y}^{\text{in}}$, or equivalently the dark matter PSD $S_{\chi\chi}^{\text{sig}}$. From Eqs. (84) and (85), we see that we can take the measured phase PSD and use it to form the estimator:

$$
S_{\chi\chi}^{\text{meas}} = \frac{S_{Y_m Y_m}^{\text{out}}}{g_\chi^2 C^2 |\chi_{mF}|^2}. \tag{86}
$$

This PSD is the noise PSD of the cavity "referred to the dark matter signal". The basic statistical question is how well we can measure the signal $S_{\chi\chi}^{\text{sig}}$ compared to the noise PSD $S_{\chi\chi}^{\text{meas}}$. In Fig. 5, we show the typical behavior of these power spectra, using the expressions for the susceptibilities and assuming input vacuum or thermal noise; improvements with the use of squeezed input fields are discussed in Sec. V.

Critically, we see that the signal PSD is typically well below the noise PSD, so one needs a long integration time compared to the signal frequency in order to measure the difference. Moreover, the best SNR is achieved in a very narrow frequency band. This means that to cover a sizable fraction of dark matter parameter space, one needs to continuously tune the cavity frequency in order to scan over many bands, much like a radio.

To give a more quantitative estimate of the scan rates, we can explicitly study two relevant limits, where either the signal or cavity is narrow compared to the other ($Q_s \gg Q_c$ or $Q_s \ll Q_c$, where $Q_c = \omega_c/\kappa$ is the cavity quality factor). At present, the former is a typical scenario for microwave cavity searches for axions, which require large external magnetic fields and use cavities $Q_c \sim 10^5$ [56, 77–80]. The latter is typical for searches for dark photons with radio-frequency cavities, which do not require magnetic fields, and have achieved much higher $Q_c \sim 10^{11}$ [76, 81–83]. We emphasize that these are present-day technical distinctions; there is no principled reason that an axion search could not be eventually performed in the $Q_s \ll Q_c$ limit.

**Narrow signal ($Q_s \gg Q_s$):** First, consider the axion search shown in the left of Fig. 5. We see that the axion PSD is very narrow compared to the width of the noise PSD for these cavity parameters. In this limit, we may make use of equation (41) to simplify the SNR as

$$
\text{SNR} = \sqrt{T_{\text{int}} \Delta\omega_a} \, \frac{\bar{S}_{aa}^{\text{in}}}{S_{aa}(\omega_a)}, \tag{87}
$$

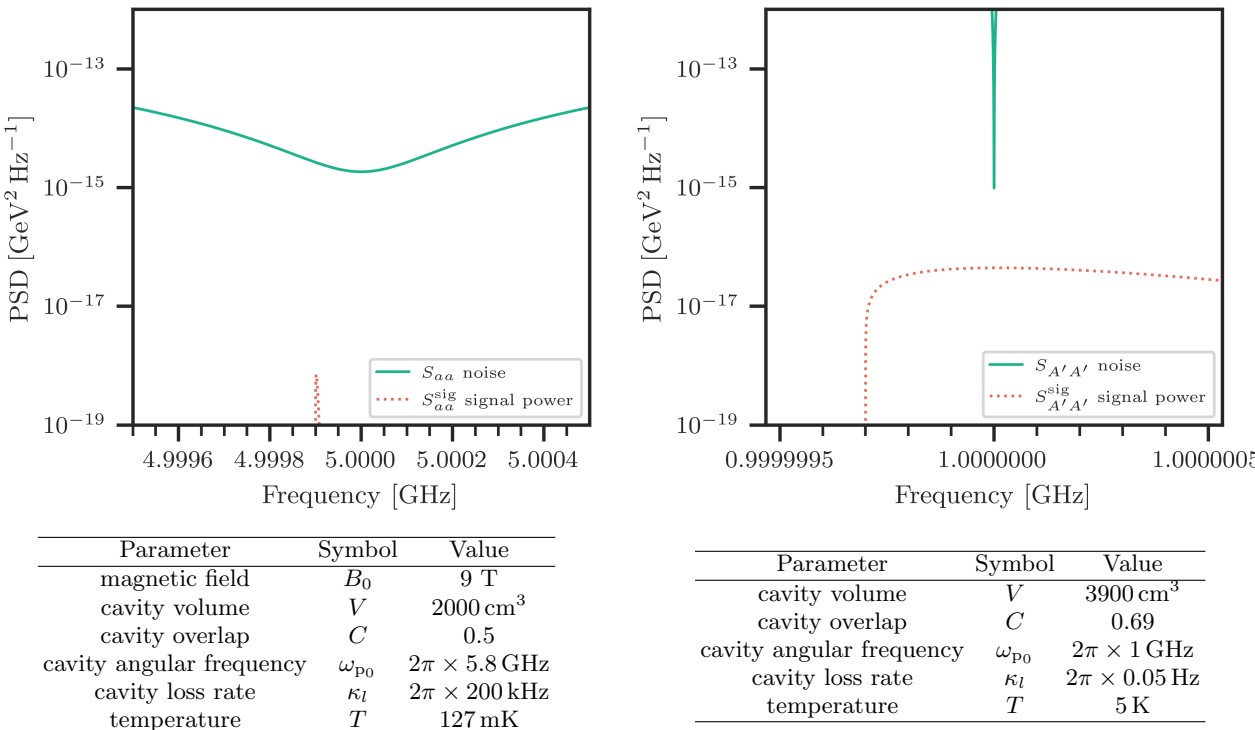

FIG. 5. **Prototypical dark matter PSDs and cavity noise PSDs.** Left: an axion search with a microwave cavity, where $Q_{\rm sig} \gg Q_{\rm det}$; parameters taken from [75]. Right: a dark photon search with a superconducting RF cavity, where $Q_{\rm sig} \ll Q_{\rm det}$; parameters taken from [76].

where $T_{\rm int}$ is the integration time of the experiment, $\omega_a \approx m_a$ is the angular frequency of the axion signal $\bar{S}_{aa}$ and $\Delta\omega_a$ is its width. We assume that $T_{\rm int}$ is much longer than the coherence time of the axion signal, so that we are probing the stochastic axion signal enough times that we may approximate $S_{aa}(\nu)$ by its constant expectation value. If the axion signal is well within a cavity linewidth of the resonant frequency $\omega_a - \omega_c \ll \kappa_l$, then by our expression (F8) for $\bar{S}_{aa}$ we have

$$
\begin{aligned}
{\rm SNR} &= \sqrt{\frac{T_{\rm int}}{\Delta\omega_a}} \frac{8 g_{a\gamma\gamma}^2 B_0^2 C V \rho_{\rm DM}}{9 m_a \kappa_l} \\
&\approx 20 \left(\frac{g_{a\gamma\gamma}}{10^{-14}\,{\rm GeV}^{-1}}\right)^2 \left(\frac{10\,\mu{\rm eV}}{m_a}\right)^{3/2},
\end{aligned}
\tag{88}
$$

where we again use the parameters of Fig. 5 as a benchmark, (86) for the signal PSD, and wrote this with explicit cavity parameters using the transfer functions (58) for $\kappa_m = 2\kappa_l$. We assume integration time $T_{\rm int} = 15$ min.

The SNR we have given quantifies the maximal sensitivity for a DM signal on-cavity-resonance. To quantify the broadband sensitivity of a cavity, we now look at the scan rate, as described in Sec. II D 3. The scan rate, in

the limit of a narrow signal, is given in Eq. (48) as

$$
\begin{aligned}
\mathcal{R} &= \frac{\Delta\omega_a}{\mathcal{S}^2} \int \frac{d\nu}{2\pi} \left(\frac{\bar{S}_{aa}^{\rm in}}{S_{aa}(\nu)}\right)^2 \\
&= \frac{3\kappa_l}{2 T_{\rm int}\mathcal{S}^2} {\rm SNR}^2(0) \\
&= \frac{32 (g_{a\gamma\gamma}^2 B_0^2 C V \rho_{\rm DM})^2}{27 \mathcal{S}^2 m_a^2 \Delta\omega_a \kappa_l}.
\end{aligned}
\tag{89}
$$

Note that the scan rate increases linearly with the inverse bandwidth $\Delta\omega$ (or, alternatively, the quality factor $Q \equiv \omega/\Delta\omega$) of both the cavity and the dark matter. If we wish to probe the couplings predicted by the KSVZ axion model [84, 85], for which [86]

$$
|g_{a\gamma\gamma}| \approx 3.9 \times 10^{-15}\,{\rm GeV}^{-1} \left(\frac{m_a}{10\,\mu{\rm eV}}\right),
\tag{90}
$$

the scan rate to reach a desired SNR of $\mathcal{S} = 5$ is

$$
\mathcal{R} \approx \frac{1.6\,{\rm GHz}}{\rm yr} \left(\frac{m_a}{10\,\mu{\rm eV}}\right),
\tag{91}
$$

where we have again used the experimental parameters in the table of Fig. 5.

**Narrow cavity** ($Q_s \ll Q_c$): Next, consider the dark photon search shown in the right of Fig. 5. We see that for such a high $Q_c$ cavity, we are in the regime where the DM PSD is approximately flat over the full cavity

bandwidth. In this case, we use the approximate expression for the SNR of Eq. (42) in terms of the integrated bandwidth $\bar{S}_{A'A'}$

$$\text{SNR} = \sqrt{T_{\text{int}}\Delta\omega_c}\frac{S^{\text{in}}_{A'A'}(\omega_{\mathbf{p}_0})}{\bar{S}_{A'A'}}, \tag{92}$$

where, for an optimally coupled cavity, $\Delta\omega_c = \kappa_m + \kappa_l = 3\kappa_m$. Numerically, this SNR is

$$\text{SNR} \approx 1 \times \left(\frac{10^{-16}}{\chi}\right)^2\left(\frac{m_{A'}}{1\,\mu\text{eV}}\right), \tag{93}$$

having assumed that $\omega_{\mathbf{p}_0} \approx m_{A'}$, and we use the thermal PSDs of Eq. (A29) with the parameters of Fig. 5 for $\kappa_m = 2\kappa_l$ and integration time $T_{\text{int}} = 30$ min.

As before, this SNR is valid for a fixed cavity frequency that lies close to the DM signal of interest. The scan rate, meanwhile, is

$$\begin{aligned}\mathcal{R} &= \frac{\Delta\omega_c}{\mathcal{S}^2}\int\frac{d\omega}{2\pi}\left(\frac{S^{\text{in}}_{A'A'}(\omega)}{\bar{S}_{A'A'}}\right)^2 \\ &= \frac{2(\chi^2 m_{A'}^2 C_0 V \rho_{\text{DM}})^2}{243\mathcal{S}^2\kappa_l\Delta\omega_a T^2} \\ &\approx \frac{0.1\text{ MHz}}{\text{yr}}\left(\frac{\chi}{10^{-16}}\right)^4\left(\frac{m_{A'}}{\mu\text{eV}}\right)^3\end{aligned} \tag{94}$$

making use of Eqs. (47) and (92). Notably, since the integrated bandwidth scales as $1/\Delta\omega_c$, the increase in scan rate with cavity quality factor persists to the $Q_c \gg Q_s$ regime.

### D. Cavities as gravitational wave detectors

While gravitational waves have now been detected using laser interferometers and now pulsar timing arrays [87], another method which has received considerable recent attention is the use of electromagnetic cavities [88–90].

A small metric perturbation $h_{\mu\nu}$ around a flat background spacetime $g_{\mu\nu} = \eta_{\mu\nu} + h_{\mu\nu}$ interacts with the electromagnetic field through the usual coupling

$$V = -\frac{1}{2}\int d^3\mathbf{x}\,h_{\mu\nu}T^{\mu\nu} \tag{95}$$

where the electromagnetic stress-energy tensor is

$$T_{\mu\nu} = F^{\mu\alpha}F^\nu_{\ \alpha} - \frac{1}{4}\eta^{\mu\nu}F_{\alpha\beta}F^{\alpha\beta} \tag{96}$$

in terms of the electromagnetic field strength tensor $F_{\mu\nu}$. The electric and magnetic fields are defined as usual $E_i = F_{0i}$ and $B_i = \epsilon_{ijk}F^{jk}$, much like the axion coupling (66). In particular, in the presence of an external magnetic field $\mathbf{B}_0$, this coupling leads to processes where incoming gravitational waves can be linearly converted

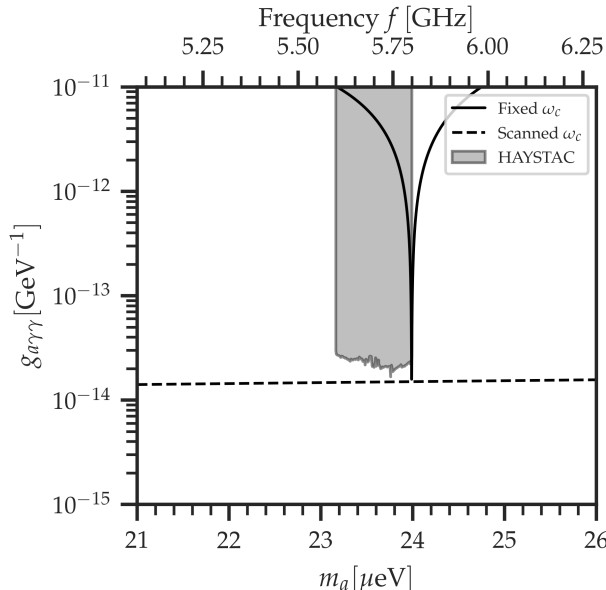

FIG. 6. **Axion dark matter search with a microwave cavity:** The discovery potential of a microwave cavity assuming quantum-limited noise spectra, compared to existing HAYSTAC limits [75]. We use the nominal parameters listed in Fig. 5, and use an SNR of 5 as our threshold. Fixed $\omega_c$ indicates the sensitivity at a single cavity frequency, while scanned $\omega_c$ assumes that we tune the cavity frequency with no other cavity parameters varying.

into electromagnetic waves [91]. For example, with a homogeneous, time-independent magnetic field $\mathbf{B}_0 = B_0\hat{\mathbf{z}}$, the coupling (95) reduces to

$$V = B_0\int_{\text{cav}}d^3\mathbf{x}\Big[(h_{xx} + h_{yy})\delta B_z - h_{xz}\delta B_x - h_{yz}\delta B_y\Big], \tag{97}$$

where $\delta\mathbf{B}$ is the perturbation of the magnetic field around the background $\mathbf{B}_0$. See, e.g., [69] for a detailed derivation.

The coupling (97) operates under the exact same conditions as the axion's interaction with the electromagnetic field described in Eq. (66), so existing axion haloscopes are automatically sensitive to gravitational waves. This mechanism can be used without a cavity, but just like the axion case, a cavity provides a way to make extremely sensitive measurements of the tiny electromagnetic fields generated by gravitational radiation. As a simple example, consider a cavity mode with wavevector $\mathbf{p}_0 = p_0\hat{\mathbf{x}}$ along the $x$-axis and polarization vector $\boldsymbol{\epsilon}_{r_0} = \hat{\mathbf{y}}$ along the $y$-axis. This mode contributes a magnetic field component only along the $z$-axis, given by

$$\delta B_z = -\partial_x\delta A_y = \omega_c\left[u^*_{\mathbf{p}_0}(\mathbf{x})a + u_{\mathbf{p}_0}(\mathbf{x})a^\dagger\right], \tag{98}$$

where, following our general cavity discussion, $a = a_{\mathbf{p}_0,r_0}$ is the cavity mode we are monitoring and $\omega_c = |p_0|$ is its frequency. Inserting this into Eq. (97), we find that

the cavity mode is driven by an external, time-dependent potential

$$V_0(t) = F_Y(t)X + F_X(t)Y, \qquad (99)$$

where

$$F_Y(t) = B_0\omega_c \int_{\text{cav}} d^3\mathbf{x}\ (h_{xx}(\mathbf{x}, t) + h_{yy}(\mathbf{x}, t))\operatorname{Re} u_{\mathbf{p}_0}(\mathbf{x})$$

$$F_X(t) = B_0\omega_c \int_{\text{cav}} d^3\mathbf{x}\ (h_{xx}(\mathbf{x}, t) + h_{yy}(\mathbf{x}, t))\operatorname{Im} u_{\mathbf{p}_0}(\mathbf{x}).$$
$$(100)$$

The similarity with the axion (70) and dark photon (78) should be clear.

The gravitational perturbations can be expanded into a mode basis, just like the axion and dark photon:

$$h_{ij}(\mathbf{x}, t) = \sum_{s=1,2} \int \frac{d^3\mathbf{k}}{M_{\text{pl}}} \epsilon_{ij}^s(\mathbf{k})v_{\mathbf{k}}^*(\mathbf{x})e^{-i\omega_k t}b_{\mathbf{k},s}^\dagger + h.c.$$
$$(101)$$

Here, $\epsilon_{ij}^s$ are a pair $(s = 1, 2)$ of polarization tensors. The factor of the Planck mass $M_{\text{pl}}$ follows from the Einstein-Hilbert action, rendering $h$ dimensionless (i.e., a strain). Unlike the axion or dark photon discussed above, however, gravitational waves are massless. Thus their wavelength $\lambda = 2\pi/\omega$, so the cavity is sensitive to gravitational waves of *the same size* as the cavity. This means that unlike the dark matter case we cannot approximate the wave as approximately constant across the detector as we did in Eq. (73). This is also different from the gravitational wave detectors based on mechanical systems like interferometers described in Sec. IV, where the GW wavelength $\lambda \gg L_{\text{det}}$.

To get an estimate for the detector sensitivity, consider a simple incoming plane wave. The detector is only sensitive to the $xx$ and $yy$ components of this wave; parametrize these as

$$h_{xx}(\mathbf{x}, t) = \overline{h}_{xx}e^{i\mathbf{k}\cdot\mathbf{x}-i\omega_k t}, \quad h_{yy}(\mathbf{x}, t) = \overline{h}_{yy}e^{i\mathbf{k}\cdot\mathbf{x}-i\omega_k t},$$
$$(102)$$

where the physical metric is the real part of this expression and the amplitudes $\overline{h}_{xx}, \overline{h}_{yy}$ are constants. In this state, the "force" on the detector's $Y$ quadrature is

$$F_Y(t) = g_0\overline{h}C_Y(\mathbf{k})e^{-i\omega_k t}, \qquad (103)$$

where $\overline{h} := \overline{h}_{xx} + \overline{h}_{yy}$ parametrizes the overall strain amplitude incident on the detector, $C_Y(\mathbf{k})$ is a dimensionless function [analogous to Eqs. (74), (81)],

$$C_Y(\mathbf{k}) = \sqrt{\frac{2\omega_c}{V_{\text{cav}}}} \int_{\text{cav}} d^3\mathbf{x}\ e^{i\mathbf{k}\cdot\mathbf{x}}\operatorname{Im} u_{\mathbf{p}_0}(\mathbf{x}). \qquad (104)$$

and the coupling rate is

$$g_0 = B_0\sqrt{V_{\text{cav}}\omega_c}$$
$$\approx 8 \times 10^{22}\ \text{Hz} \times \left(\frac{B_0}{1\ \text{T}}\right)\left(\frac{\omega_c}{1\ \text{GHz}}\right)^{1/2}\left(\frac{V_{\text{cav}}}{(10\ \text{cm})^3}\right)^{1/2}.$$
$$(105)$$

This large numerical value should be contrasted with the tiny strains $h < 10^{-20}$ produced by realistic gravitational wave sources. Analogous expressions hold for the force on the $X$ quadrature. Crucially, note that our dimensionless factor $C(\mathbf{k})$ here depends on the incoming GW wavelength and direction, and falls off rapidly when $|\mathbf{k} - \mathbf{p}_0| \gtrsim \kappa_c$, i.e., when the gravitational wave is non-resonant with the cavity.

Finally, we can compute a power spectral density using this analysis. The potential generated by the gravitational wave (99) leads to an input force in the Heisenberg equations for the cavity. In particular, this coupling drives the phase quadrature $Y$, as in Eq. (54). Thus, an incoming gravitational wave leaves an imprint on the phase quadrature of the cavity, which can be read out much like an axion signal. We construct the estimator [see Eq. (64)]

$$h_E(\nu, \mathbf{n}) = \frac{F_E(\nu)}{g_0 C_Y(\nu\mathbf{n})} = \frac{Y_m^{\text{out}}(\nu)}{g_0 C_Y(\nu\mathbf{n})\chi_{mF}(\nu)}, \qquad (106)$$

which gives us an estimate of $\overline{h}$ given an output stream of phase data $Y_m^{\text{out}}$ from the cavity's measurement port. We have written this as a function of frequency as well as the incoming angle of the gravitational wave, taken to be $\mathbf{k} = \nu\mathbf{n}$ where $\mathbf{n}$ is a unit vector. The resulting strain noise power spectral density is

$$S_{hh}(\nu, \mathbf{n}) = \frac{S_{Y_m Y_m}^{\text{out}}(\nu)}{g_0^2|C_Y(\nu\mathbf{n})|^2|\chi_{mF}(\nu)|^2}. \qquad (107)$$

The measurement port phase PSD is given explicitly in Eq. (60). This strain PSD has units of strain per frequency, as expected. We plot an example of the resulting strain sensitivity in Fig. 7.

## IV.  MECHANICAL SENSORS

Mechanical sensors consist of one or more mechanical degrees of freedom, for example the center of mass and higher-order phononic excitations of a solid body, coupled to and read out through the electromagnetic field. These systems can be constructed in a diverse array of scales: the mechanical degrees of freedom can range from the motional state of a single electron [92–94] or low-energy phononic modes of a liquid or crystal [95] to the center-of-mass coordinate of solid state objects ranging from nanoparticles [27, 28] up to the kilogram-scale mirrors of LIGO [96]. The electromagnetic readout can similarly range in frequency from microwave to optical [97].

The Hamiltonian for a mechanical element can be written as a simple mode sum

$$H_{\text{mech}} = \sum_n \omega_n d_n^\dagger d_n. \qquad (108)$$

Here, $n$ labels the motional modes of the system, in general including the center-of-mass (usually denoted $n = 0$)

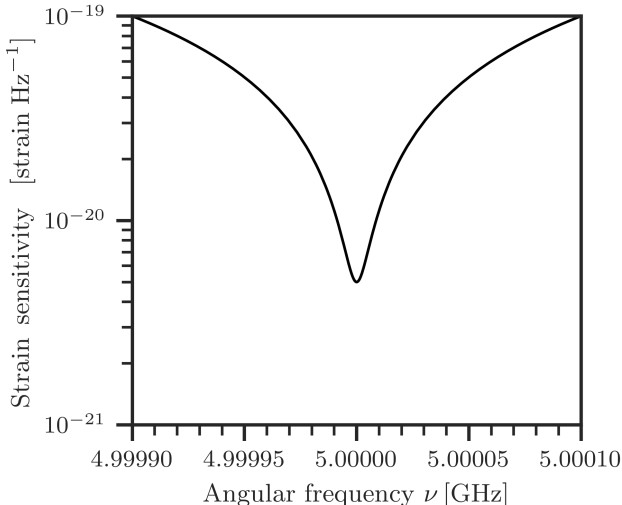

FIG. 7. **High-frequency gravitational wave detection with a microwave cavity:** An example strain PSD with a microwave cavity in an external magnetic field, from Eq. (107). We use the microwave cavity parameters given in the left of Fig. 5, and for simplicity assume gravitational waves incoming directly along the cavity axis $\mathbf{p}_0$.

and any higher-order modes. The creation and annihilation operators satisfy standard commutation relations.

To understand how these modes can be read out with the electromagnetic field, consider the canonical example of a mechanical force sensor: a cavity with a fixed, partially transparent mirror on one side, and a highly reflective mirror on the other end. See Fig. 8. This forms what is called a Fabry-Pérot cavity. Typically the reflective, movable mirror is modelled as a harmonic oscillator for the mode $n = 0$; for instance, it may be physically suspended as a pendulum. Alternatively, one could imagine for example a membrane clamped to a substrate, in which case the $n = 0$ mode is non-dynamical and one instead uses the higher-order phonon modes as a detector [98, 99]. Here we give a brief review of the basic physics of such a system; see Appendix E for more details.

The operating principle is that the cavity field, which can be expanded into modes as in Eq. (49), has mode frequencies which now depend on the dynamical length of the cavity. This leads to the so-called optomechanical coupling between the light and mirror. Let $L$ denote the equilibrium length of the cavity, and $x \ll L$ some small displacement of the movable mirror. Then we can expand the cavity mode frequencies

$$\omega_p(x) = \omega_p(0) \left[ 1 + \frac{x}{L} + \mathcal{O}\left( \left( \frac{x}{L} \right)^2 \right) \right], \qquad (109)$$

where $\omega_p(0)$ are the cavity mode frequencies at equilibrium. As a simple example, for a 1d cavity, we have integer-indexed frequencies $\omega_p(x) = \pi p/(L - x)$ so $\omega_p(0) = \pi p/L$.

In principle, one has to consider all of the mechanical

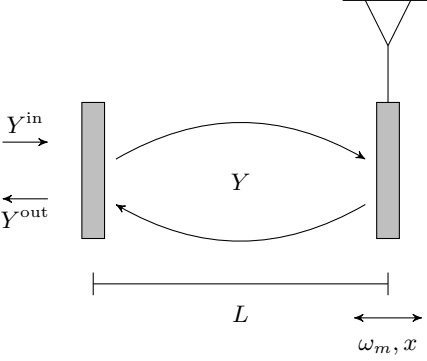

FIG. 8. **Fabry-Pérot cavity.** A common example of a mechanical force sensor is comprised of two parallel mirrors, one partially transmissive and one highly reflective and treated as a mechanical oscillator with mass $m$ and resonance frequency $\omega_m$. Here, we depict the leftmost mirror as fixed, and the rightmost suspended; even if both mirrors are moving, this description just amounts to identifying the mass $m$ as the reduced mass (Appendix E).

modes and all of the cavity modes, which all couple to each other through this effect. In simple situations, we can usually start by consider a single cavity mode, say $a = a_{p_0, r_0}$ with equilibrium frequency $\omega_c$, and a single mechanical mode, say $d_{n_0}$, with frequency $\omega_m$. It is conventional to define position and momentum variables for the mechanical mode $x_m = x_0(d + d^\dagger)$ and $p = p_0(d - d^\dagger)$; in the case of the center-of-mass mode these are literally the position and momentum of the mass, while for higher-order phonons they represent a mode amplitude and its conjugate. Here, $x_0 = 1/\sqrt{2m\omega_m}$ and $p_0 = \sqrt{m\omega_m/2}$ represent the ground-state wavefunction widths. Inserting Eq. (109) into the cavity mode Hamiltonian (52), we obtain the total Hamiltonian for the mirror and cavity mode:

$$H_{\text{bare}} = \frac{p^2}{2m} + \frac{1}{2}m\omega_m^2 x^2 + \omega_c a^\dagger a + g_0 x a^\dagger a, \qquad (110)$$

where the bare optomechanical coupling is[5]

$$g_0 = \frac{\omega_c}{L}. \qquad (111)$$

The notation "bare" refers to the fact that this Hamiltonian descibes the system in the absence of any drive laser.

Driving the system with a laser effectively shifts the cavity mode by a coherent state, $a \to \alpha + a$, where $|\alpha|^2 = n_{\text{cav}} = P_L/\omega_c\kappa$ is the average number of photons in the cavity given an input laser drive strength $P_L$ and cavity loss rate $\kappa$. After an appropriate renormalization of the

---

[5] Our convention here gives $g_0$ units of a rate squared. Another common convention is to scale by a factor of the mirror's ground-state width $x_0 = 1/\sqrt{2m\omega_m}$ to give $g_0$ units of a rate.

coupling strength and equilibrium position, this leads to a Hamiltonian

$$H_{\text{det}} = \frac{p^2}{2m} + \frac{1}{2}m\omega_m^2 x^2 + \omega_c a^\dagger a + \sqrt{2}gxX, \qquad (112)$$

where the "drive-enhanced" coupling is

$$g = |\alpha|g_0 \gg g_0, \qquad (113)$$

and again $X = (a + a^\dagger)/\sqrt{2}$ is the amplitude quadrature of the cavity mode [see Eq. (53)].

Eq. (112) defines a standard driven optomechanical cavity system. More sophisticated examples like Michelson interferometers, which involve a pair of such systems, are treated in the next sections. We also emphasize that beyond the center-of-mass mode considered here, the higher-order phononic modes of a solid can also be treated in an identical fashion, for example in a resonant bar (Weber-type) gravitational wave detector, or vibrational modes of liquid helium. These modes can be detected for example by coupling optical or microwave light through a Brilloun coupling [95, 100].

### A. Input-output theory for optomechanics

In our discussion of cavities in Sec. III, we assigned two baths to the cavity: one from the measurement port, and one from instrinsic losses such as blackbody radiation in the cavity walls. In cavity optomechanics, we have to add a third bath, associated to damping and losses in the mechanical system itself. Physically, this corresponds to things like gas in the chamber which is creating residual pressure on the mechanics, phonons transferring between the mechanical system and any physical support system like threads used to suspend it as a pendulum, and so forth. In this section, we will focus on cavities which are overcoupled, meaning that their loss $\kappa_m$ from the measurement port is much larger than the cavity's intrinsic loss $\kappa_m \gg \kappa_\ell$. This should be contrasted with some of the axion cavity examples in Sec. III C, where one often has nearly critically-coupled cavities $\kappa_m \approx \kappa_\ell$. The measurement port bath corresponds to fluctuations around the laser, so we have some ability to control and measure the state of this bath, while the mechanical bath and any intrinsic cavity loss baths are unobservable.

Following the same logic as in Sec. III, we use the optomechanical Hamiltonian (112) to compute the Heisenberg equations for the cavity and mechanical variables. Supplementing these with the appropriate bath terms, in the frame co-rotating with the laser (at frequency $\omega_L$) we obtain

$$\begin{aligned}
\dot{X} &= \Delta Y - \frac{\kappa}{2}X + \sqrt{\kappa}X^{\text{in}}, \\
\dot{Y} &= -\Delta X - \frac{\kappa}{2}Y + \sqrt{\kappa}Y^{\text{in}} - \sqrt{2}gx, \\
\dot{x} &= \frac{p}{m}, \\
\dot{p} &= -m\omega_m^2 x - \gamma p + F^{\text{in}} - \sqrt{2}gX
\end{aligned} \qquad (114)$$

Here, $\kappa$ is the loss rate of the cavity mode to the measurement port, while $\gamma$ is the mechanical damping rate.[6] Similarly, the input fields $X^{\text{in}}, Y^{\text{in}}$ represent the cavity bath while $F^{\text{in}}$ represents the random drive from the mechanical bath; $F^{\text{in}}$ will also contain most of the signals we are interested in. The parameter $\Delta = \omega_L - \omega_c$ is the detuning of the laser drive from the cavity frequency, and we will usually take $\Delta = 0$ in the following for simplicity, although many important phenomena require non-zero detuning (see, e.g., chapters 4-5 of Ref. [101]).

The cavity-mechanical interaction term $V = \sqrt{2}gxX$ causes the cavity field to be driven by the mechanics [the last term in the second equation of (114)], and conversely the mechanics to be driven by the cavity field [the last term in the fourth equation (114)]. Thus this interaction both imprints the mechanical state onto the cavity mode as well as causes a radiation pressure "back-action" force on the mechanics itself.

The input-ouput relations for the cavity measurement port are

$$\begin{aligned}
X^{\text{out}} &= X^{\text{in}} - \sqrt{\kappa}X, \\
Y^{\text{out}} &= Y^{\text{in}} - \sqrt{\kappa}Y.
\end{aligned} \qquad (115)$$

Following the same logic as in Sec. III, we can transform to the frequency domain to solve the equations of motion (114) for $X(\nu), Y(\nu)$ in terms of the input modes. Plugging these into (115), we obtain for example the output phase $Y_{\text{out}}$ as

$$Y^{\text{out}} = \chi_{YY}Y^{\text{in}} + \chi_{YX}X^{\text{in}} + \chi_{YF}F^{\text{in}} \qquad (116)$$

where the input-output transfer functions are

$$\begin{aligned}
\chi_{YY}(\nu) &= 1 + \kappa\chi_c(\nu) = e^{i\phi_c(\nu)} \\
\chi_{YX}(\nu) &= -2\kappa g^2\chi_c^2(\nu)\chi_m(\nu), \\
\chi_{YF}(\nu) &= -(2\kappa g^2)^{1/2}\chi_c(\nu)\chi_m(\nu).
\end{aligned} \qquad (117)$$

in terms of the cavity and mechanical susceptibilities

$$\begin{aligned}
\chi_c(\nu) &= \frac{1}{i\nu - \kappa/2}, \\
\chi_m(\nu) &= \frac{1}{m(\omega_m^2 - \nu^2 - i\gamma\nu)}.
\end{aligned} \qquad (118)$$

The expression $e^{i\phi_c(\nu)}$ for the $\chi_{YY}$ transfer function expresses the fact that the input phase quadrature gets a phase shift from the cavity, and follows from the expression for $\chi_c$. One can get a similar expression for the amplitude quadrature $X^{\text{out}}$, but we will mostly be considering detectors which measure the output phase.

—————

[6] Note that, following standard practice, we are modeling the mechanical bath as a pure damping effect, i.e., there is no bath term in the $\dot{x}$ equation of motion. This is in contrast to the cavity, where the bath acts symmetrically on the two quadratures $X, Y$. See Appendix E for some discussion on this point.

We can obtain noise spectral densities for the output in terms of the input noise, again following the logic outlined in the previous sections. The output phase has noise PSD

$$S_{YY}^{\text{out}} = |\chi_{YY}|^2 S_{YY}^{\text{in}} + |\chi_{YX}|^2 S_{XX}^{\text{in}} + |\chi_{YF}|^2 S_{FF}^{\text{in}} + \chi_{YX}\chi_{YY}^* S_{YX}^{\text{in}} + \chi_{YY}\chi_{YX}^* S_{XY}^{\text{in}}. \tag{119}$$

Here, we made the obvious physical assumption that the mechanical input noise (from phonons, gas, etc.) is uncorrelated with the input noise on the light (from laser fluctuations, etc.), so that $S_{XF}^{\text{in}} = S_{YF}^{\text{in}} = 0$; note however that this does not mean the total noise on the mirror is uncorrelated with the total noise in the light! The terms on the second line, however, represent possible correlations between the $X$ and $Y$ input noises. In many states of interest, particularly the vacuum, these terms vanish, but they can be non-zero (and in fact negative) in nontrivial states, particularly the squeezed vacuum. This point is discussed in detail in Sec. V.

We will also often be interested in signals which can be expressed as part of the force $F_{\text{in}}$ acting on the mechanical element. We can define an estimator $F_E$ using the readout $Y^{\text{out}}$, following the general discussion in Sec. II C:

$$F_E(\nu) = \frac{Y^{\text{out}}(\nu)}{\chi_{YF}(\nu)} \implies S_{FF} = \frac{S_{YY}^{\text{out}}}{|\chi_{YF}|^2}. \tag{120}$$

In gravitational wave detectors, one often parametrizes signals in terms of dimensionless strains; we discuss this in Sec. IV B 1.

Finally, to get some intuition and discuss the "Standard Quantum Limit" of these systems, we can give a typical example of the noise PSDs. Assume that the laser drive is a perfect coherent state, which means that the fluctuations around the laser ($X^{\text{in}}$, $Y^{\text{in}}$) are in the quantum vacuum state, with PSDs given by

$$S_{XX}^{\text{in}} = S_{YY}^{\text{in}} = \frac{1}{2}, \quad S_{XY}^{\text{in}} = 0. \tag{121}$$

as in Eq. (61). This should be a good approximation when the ambient temperature $T \ll \omega_L$, which holds for example in an optical system $\omega_L \gtrsim (1000 \text{ nm})^{-1} \sim 300$ THz $\sim 2300$ K even at room temperature. For the mechanical system however, which is typically a much lower frequency system, we have to deal with thermal noise. A typical parametrization of this noise is

$$S_{FF}^{\text{in}} = 4m\gamma k_{\text{B}}T, \tag{122}$$

which represents white noise with total power set by a combination of the mechanical damping rate $\gamma = \omega_m/Q_m$ and bath temperature $T$, where $Q_m$ is the mechanical quality factor.

In this system, the SQL can be understood as follows. In our example with input vacuum noise [Eq. (121)], the output noise for our force estimator has two terms

from the input light, one each from the input phase and amplitude fluctuations:

$$S_{FF}^{\text{quantum}} := \frac{|\chi_{YY}|^2}{|\chi_{YF}|^2}S_{YY}^{\text{in}} + \frac{|\chi_{YX}|^2}{|\chi_{YF}|^2}S_{XX}^{\text{in}}. \tag{123}$$

The notation "quantum" here is standard in the optomechanics literature and is what is often referred to as "quantum noise". The frequency-dependent coefficients here, in terms of the transfer functions given in Eq. (117), appear with different powers of the enhanced optomechanical coupling ($g^{-2}$ and $g^2$, respectively). Since $g = g_0|\alpha| \sim \sqrt{P_L}$ depends on the input laser power [see Eq. (113)], we can therefore tune the laser to optimize the sum of these noise contributions in various ways. In particular, one picks a particular reference frequency $\omega_*$, solves $\partial S_{FF}^{\text{quantum}}(\omega_*)/\partial g^2 = 0$ for $g$ to get a coupling $g_* = g_*(\omega_*)$ (or equivalently laser power), such that the total quantum noise at the frequency $\omega_*$ is minimized. This minimization means that the two contributions — one from the shot noise $S_{YY}^{\text{in}}$, the other from the backaction noise $S_{XX}^{\text{in}}$ — are equal at this frequency.

For example, one could choose the mechanical resonance frequency $\omega_* = \omega_m$. Minimizing $S_{FF}^{\text{quantum}}(\omega_*)$ with respect to $g$,

$$0 = \frac{\partial S^{\text{quantum}}(\omega_*)}{\partial g^2} \implies g_*^2 = \frac{1}{2\kappa|\chi_c(\omega_*)|^2|\chi_m(\omega_*)|} \tag{124}$$

and setting $\omega_* = \omega_m$, one finds, using the expressions (117), that the noise at resonance reduces to the simple expression

$$S_{FF}^{\text{quantum}}(\omega_m)\Big|_{\omega_*=\omega_m} = \frac{2}{|\chi_m(\omega_m)|} = 2m\gamma\omega_m. \tag{125}$$

This level of noise is often referred to as an SQL. Physically, it reflects the fact that the input phase noise ("shot noise") and input amplitude noise ("back-action", i.e., random radiation pressure) are tuned to be equal at this particular frequency. We can convert it to the form usually quoted, as a position sensitivity, using $x(\nu) = \chi_m(\nu)F(\nu)$,

$$S_{xx}^{\text{quantum}}(\omega_m)\Big|_{\omega_*=\omega_m} = \frac{1}{2m\gamma\omega_m}. \tag{126}$$

Since this is the SQL on resonance, we should consider integration over a bandwidth given by the mechanical linewidth $\Delta\nu \approx \gamma_m$ in an estimate of the variance [c.f. Eq. (30)],

$$\Delta x^2 \approx \int d\nu \frac{\sin^2(\nu\tau/2)}{\nu^2\tau^2}S_{xx}(\nu) \approx \frac{S_{xx}(\omega_m)\gamma_m}{\tau^2\omega_m^2} = \frac{1}{2m\omega_m}, \tag{127}$$

where we used an effective measurement time $\tau \sim 1/\omega_m$. This last expression reproduces the basic intuition that measurements at the SQL refer to measurements at the level of vacuum fluctuations—here, the ground-state wavefunction of the mechanical oscillator.

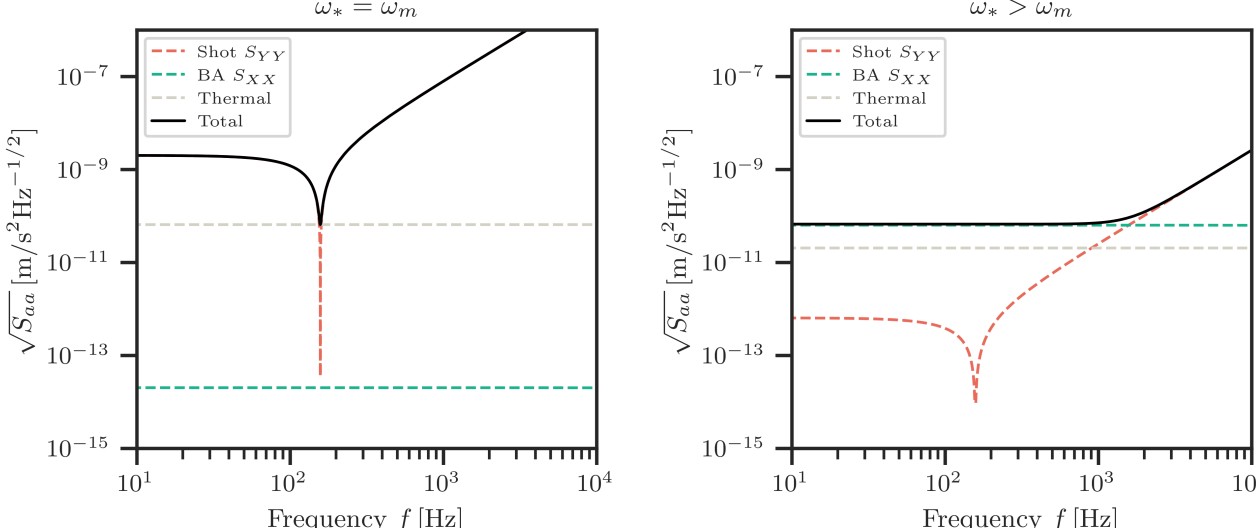

FIG. 9. **Mechanical force sensing.** Noise PSDs for a prototypical cavity optomechanical device, with different choices of the laser power [Eq. (124)]. Left: SQL on resonance $\omega_* = \omega_m$. Right: SQL above resonance $\omega_* \gg \omega_m$ ("free particle limit"). We plot this as acceleration sensitivity, which is obtained from the force PSD by $S_a = \sqrt{S_{FF}/m^2}$, thus coming with units of (acceleration)/$\sqrt{\text{Hz}}$. Device parameters $m = 1$ mg, $\omega_m/2\pi = 1000$ Hz, $Q_m = 10^5$, in a cavity of length $L = 10$ cm and $Q_c = 10^8$. In both plots we assume incoming vacuum noise for the laser and thermal noise on the mechanics at $T = 10$ mK. Improvements from squeezed light injection are shown in Fig. 16, in Sec. V A.

We give a much more general and detailed discussion in Appendix E 3, including the expression for the noise PSD at frequencies away from the resonance $\nu \neq \omega_m$. It is important to emphasize that, as we will see below, different detection problems (for example, gravitational waveform estimation at frequencies well above the mechanical resonance, or resonant detection of a specific monochromatic signal, as in some dark matter problems), the specific way one should choose the SQL frequency $\omega_*$ can vary. We analyze this in some detail in what follows.

### B. Mechanical gravitational wave detectors

Gravity couples to all forms of energy including rest mass, and can be detected in a number of ways. We have already discussed one method, based on conversion of gravitational power to electromagnetic power in a cavity, in Sec. III D. In this section we now discuss how mechanical sensors, like the simple Fabry-Pérot system discussed above or more sophisticated interferometer configurations, can be used to sense wave-like perturbations in the spacetime geometry.

#### 1. GW-mechanical coupling

The coupling of weak gravitational fields $h_{\mu\nu}$ to matter, including both massive objects and to light, is given by

$$V = \frac{1}{2} \int d^3\mathbf{x} \, h_{\mu\nu} T^{\mu\nu}. \qquad (128)$$

This is the same expression as in Eq. (95). In an optomechanical device, $T^{\mu\nu}$ contains contributions from both the light [as in Eq. (96)], as well as a contribution from the mechanical element

$$T_{\text{m}}^{\mu\nu}(x) = m \int d^4x \, \frac{dx^\mu}{d\tau} \frac{dx^\nu}{d\tau} \delta^4(x - x(\tau)), \qquad (129)$$

where $x^\mu(\tau)$ is the trajectory of the mechanical system in terms of its proper time $\tau$.

To make this coupling more explicit in terms of a specific detector architecture requires picking a gauge, i.e., a system of spacetime coordinates. There are two commonly used choices. The first, "transverse-traceless" (TT) gauge, gives a simple expression for $h_{\mu\nu}$ in terms of a plane wave expansion [see Eq. (101)]. In this gauge, the coupling to the mechanics becomes trivial to leading order in $h$, and the effect of the gravitational perturbations is to change the state of the light. The other choice, "Newtonian gauge" or the "proper detector frame", instead has the light unchanged by gravity but the mechanical element feels an effective force. For a more details on these gauges, see [102–105].

For our purposes, it is most straightforward to work in the proper detector frame, where we can directly use our force sensing results. In this case, the force felt by the mechanical system is given simply by

$$V(t) = \frac{m}{4} x^i x^j \ddot{h}_{ij}^{TT}(t), \qquad (130)$$

where, perhaps confusingly, we have written the gravitational perturbation itself in terms of its transverse-traceless expression (101). We have also assumed that we are interested in gravitational modes with wavelength $\lambda \sim 1/k \gg L$, where $L$ is the equilibrium length of the mechanical cavity; in this case, the spatial dependence of the $h$ modes drops out, and we can write simply

$$h_{ij}^{TT}(t) = \int d^3\mathbf{k} \frac{e^{-ikt}}{M_{\rm pl}\sqrt{2k}} \epsilon_{s,ij}^*(\mathbf{k}) b_{\mathbf{k},s}^\dagger + h.c. \qquad (131)$$

This generates a force on each axis of the mechanical element

$$F_{\mathrm{sig},i}^{\mathrm{in}}(t) = \frac{m}{2} x^j \ddot{h}_{ij}^{TT}(t). \qquad (132)$$

In a typical detector like LIGO, only one axis is monitored (per mirror in the Michelson interferometer), but in general one can make a multi-axis device as discussed in the beginning of this section. One usually sees this written as

$$F^{\mathrm{sig}}(\nu) = \frac{m}{2} L\nu^2 h(\nu), \qquad (133)$$

where $h(\nu)$ is the Fourier transform of

$$h(t) := S^{ij} h_{ij}^{\mathrm{TT}}(t), \qquad (134)$$

$S^{ij}$ is a dimensionless "antenna function" which picks out the components of the gravitational wave which couple to the detector, we moved to frequency space, expanded the mirror position $x = L + \delta x$ around the cavity's equilibrium length, and kept only the leading term. For the Fabry-Pérot detector of Sec. IV B 2 aligned along the $x$-axis we have $S_{ij} = \hat{x}_i \hat{x}_j$; for a Michelson interferometer like Sec. IV B 3, we have $S_{ij} = \hat{x}_i \hat{x}_j - \hat{y}_i \hat{y}_j$, where the $\hat{x}, \hat{y}$ are unit vectors.

### 2. Fabry-Pérot cavity

To begin, we consider a simple toy model for gravitational wave detection, using just a single Fabry-Pérot cavity. This system contains all the basic effects that we will need to understand more standard, Michelson-based interferometers such as LIGO. Although it is sometimes claimed that one needs a Michelson or other 2d configuration to observe GWs, this is incorrect: all we need is a ruler in one direction. The real reason for using a Michelson interferometer has to do with cancellation of certain laser effects, as we discuss in the next section.

The phase quadrature of the output field of the Fabry-Pérot cavity contains the GW signal we are interested in measuring. We are interested in sensing the "force" signal of Eq. (133). To infer the GW strain from this output quadrature, we construct its estimator

$$h_E(\nu) := \frac{2}{mL\nu^2} \chi_{YF}^{-1}(\nu) Y^{\mathrm{out}}(\nu). \qquad (135)$$

To determine our GW sensitivity, we can derive the strain noise PSD following the same logic of Sec. II C. Using Eq. (119) and Eq. (135), we have

$$S_{hh}(\nu) = \frac{1}{2} \left[h_{\mathrm{SQL}}^{FP}(\nu)\right]^2 \left(\frac{1}{|\chi_{YX}(\nu)|} + |\chi_{YX}(\nu)|\right), \qquad (136)$$

where we have assumed vacuum input fluctuations around the laser $S_{XX}^{\mathrm{in}} = S_{YY}^{\mathrm{in}} = 1/2$, and for simplicity assumed that the mechanical quality factor $Q_m$ is sufficiently high that we can neglect thermal noise.[7] The overall coefficient is

$$h_{\mathrm{SQL}}^{FP}(\nu) = \sqrt{\frac{8}{m^2 L^2 \nu^4} \frac{|\chi_{YX}(\nu)|}{|\chi_{YF}(\nu)|^2}}. \qquad (137)$$

We plot this noise PSD in Fig. 10.

A common simplification is to assume that the laser power is tuned to optimize noise well above the mechanical resonance frequency, i.e., $\omega_* \gg \omega_m$ in the language of Sec. IV A. For example, in LIGO, the laser is optimized around $\sim 100$Hz, and the resonance frequency of the suspended mirrors is $\sim 1$Hz. In this limit, we can treat the mirrors as "free masses", approximating the mechanical response function as $\chi_m(\nu) \approx -1/m\nu^2$. In this limit, the prefactor Eq. (137) simplifies to

$$h_{\mathrm{SQL}}^{FP} = \sqrt{\frac{8}{mL^2\nu^2}}. \qquad (138)$$

### 3. Michelson interferometer

We now turn to a more common architecture for gravitational wave detection: a laser impinging on a balanced beam-splitter that sends light into two Fabry-Pérot cavities at a right angle, forming a Michelson interferometer. See Fig. 10. The primary reason for using a Michelson interferometer rather than a simple Fabry-Pérot has nothing to do with the tensor nature of gravity, but rather to do with cancellation of a number of noise sources, particularly those in the laser system, as we will see explicitly below.

---

[7] In LIGO, the $Q_m$ for the fundamental pendulum mode is around $10^8$ [106], which justifies this assumption. In more general detectors, this is not necessarily true, and one should add the thermal noise term.

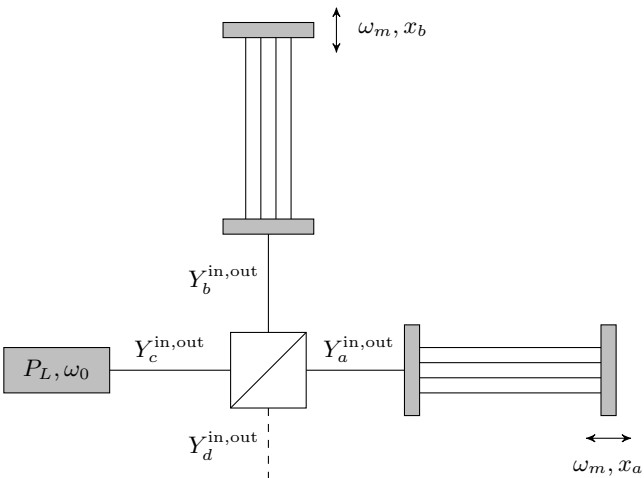

FIG. 10. **Michelson interferometer:** A laser with carrier frequency $\omega_0$ and power $P_L$ is sent into a balanced beamsplitter, sending two beams into two separate Fabry-Pérot cavities. The $Y_d^{\text{out}}$ line represents the field that is measured.

Denote the cavity modes in the two Fabry-Pérot arms by $a$ and $b$. We will assume that these cavities are physically identical, i.e., have the same response functions. The two cavities satisfy the usual input-output relations

$$Y_a^{\text{out}} = \chi_{YY} Y_a^{\text{in}} + \chi_{YX} X_a^{\text{in}} + \chi_{YF} F_a^{\text{in}}$$
$$Y_b^{\text{out}} = \chi_{YY} Y_b^{\text{in}} + \chi_{YX} X_b^{\text{in}} + \chi_{YF} F_b^{\text{in}}. \tag{139}$$

The input-output fields which we actually control, call them $c$ and $d$, are related to the $a, b$ input-output fields through a balanced beam-splitter:

$$\begin{pmatrix} a^{\text{in}} \\ b^{\text{in}} \end{pmatrix} = \frac{1}{\sqrt{2}} \begin{pmatrix} 1 & -1 \\ 1 & 1 \end{pmatrix} \begin{pmatrix} c^{\text{in}} \\ d^{\text{in}} \end{pmatrix}. \tag{140}$$

In the usual configuration, the input field $c^{\text{in}}$ has a laser drive, so here by $c^{\text{in}}$ we will mean fluctuations around this drive as in the previous sections. The field $d^{\text{in}}$ is the "dark port". In the simplest case, $d^{\text{in}}$ is a literal vacuum noise contribution; in more modern detectors, $d^{\text{in}}$ can also be squeezed (see Sec. V).

The basic idea of reading out the dark port is that if the two arms have the same length, the dark port is truly dark (up to vacuum noise); if the arm lengths are different, the two beams do not perfectly interfere at the beamsplitter, and light comes out of the dark port. Mathematically, consider monitoring the phase of light exiting the dark port:

$$Y^{\text{out}} := Y_d^{\text{out}} = \frac{1}{\sqrt{2}}(Y_b^{\text{out}} - Y_a^{\text{out}}). \tag{141}$$

From Eq. (139), we see that this phase output will pick up a term proportional to the signal

$$Y^{\text{out}} \supset \frac{\chi_{YF}}{\sqrt{2}}(F_a^{\text{in}} - F_b^{\text{in}}), \tag{142}$$

which in the case of a gravitational wave is given by (132).

To calculate the sensitivity of the interferometer, we need to find the noise PSD of $Y^{\text{out}}$. Using the usual Wiener-Khinchin result [Eq. (24)], this boils down to calculating the correlator $\langle Y^{\text{out}} Y^{\text{out}\dagger} \rangle$ in frequency space. Notice that, from Eq. (141), this could naively involve cross-correlators between the two arms. However, we have

$$\langle Y_a^{\text{in}} Y_b^{\text{in}\dagger} \rangle = \frac{1}{2} \langle (Y_c^{\text{in}} - Y_d^{\text{in}})(Y_c^{\text{in}\dagger} + Y_d^{\text{in}\dagger}) \rangle$$
$$= \frac{1}{2} \left[ \langle Y_c^{\text{in}} Y_c^{\text{in}\dagger} \rangle - \langle Y_d^{\text{in}} Y_d^{\text{in}\dagger} \rangle \right] \tag{143}$$
$$= 0,$$

where here we have assumed input vacuum noise, which in particular means that the $c, d$ modes are uncorrelated, and the contributions from the $c, d$ ports are equal. A similar computation shows that $\langle X_a^{\text{in}} X_b^{\text{in}\dagger} \rangle = 0$. Using this, we obtain

$$S_{YY}^{\text{out}} = \frac{1}{2} |\chi_{YY}|^2 + \frac{1}{2} |\chi_{YX}|^2, \tag{144}$$

where we have again assumed that we can neglect thermal noise terms (i.e. noise contributions from $F_{a,b}^{\text{in}}$).

Our result for the phase output noise Eq. (144) is identical to the result with a single Fabry-Pérot cavity, Eq. (119). There is however a difference, coming in the definition of "the strain" (134): a Fabry-Pérot configuation is sensitive to only metric perturbations along one axis ($h_{xx}$, for example), while the Michelson configuration is sensitive to the difference of two ($h_{xx} - h_{yy}$, for example). This follows directly from the definitions of the forces $F_{a,b}^{\text{in}}$ given in Eq. (132).

To derive the full sensitivity to strain, we can form the estimator

$$h_E(\nu) := \frac{2}{mL\nu^2} \chi_{YF}^{-1}(\nu) Y^{\text{out}}(\nu) \tag{145}$$

for the strain in the Michelson case, and compute its PSD using Eq. (144). One obtains

$$S_{hh} = \frac{h_{\text{SQL}}^2}{2} \left( |\chi_{YX}| + \frac{1}{|\chi_{YX}|} \right), \tag{146}$$

where we have again assumed vacuum inputs, and defined the standard quantum limit for the square root of the double-sided power spectral density of the gravitational-wave strain for a Michelson interferometer as

$$h_{\text{SQL}}(\nu) = \sqrt{\frac{4}{m^2 L^2 \nu^4} \frac{|\chi_{YX}(\nu)|}{|\chi_{YF}(\nu)|^2}}. \tag{147}$$

In Fig. 11, we plot both the single-arm and Michelson strain sensitivities.

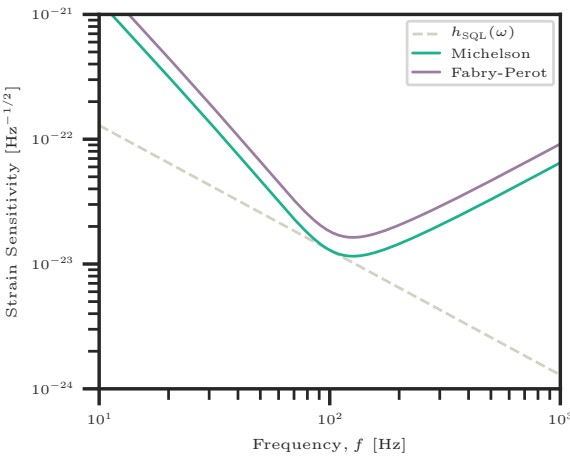

| Parameter | Symbol | Value |
|---|---|---|
| mirror mass | $m$ | 30 kg |
| cavity arm length | $L$ | 4 km |
| mechanical resonance | $\omega_m$ | $2\pi \times 1$ Hz |
| laser frequency | $\omega_L$ | $1.8 \times 10^{15}$ Hz |
| total cavity energy loss rate | $\kappa$ | $2\pi \times 200$ Hz |

FIG. 11. **Michelson vs. Fabry-Pérot:** Quantum noise contributions to the strain sensitivity $S_h := \sqrt{S_{hh}}$ of both a simple Fabry-Pérot [Eq. (136)] and a Michelson interferometer [Eq. (146)]. We assume input vacuum noise, and use the parameters listed in the table, which were taken from [104, 107]. Notice that the Michelson has an overall sensitivity that is a factor of 2 smaller than a single Fabry-Pérot cavity.

### C. Mechanical sensors as "particle" detectors

In addition to their use in gravitational wave detection, mechanical sensors also have a long history in particle physics. Perhaps most notable is the use of mechanical torsion balances to test the equivalence principle [108] and to search for "fifth forces", i.e., deviations from Newton's law of gravitation [109].

More recently, mechanical devices with a wide range of masses have been used or proposed for use as detectors of a variety of dark matter candidates [110] in both the particle-like ($m_{\rm DM} \gtrsim 1$ eV, [29, 93, 94, 111–114]) and wave-like ($m_{\rm DM} \lesssim 1$ eV, [115–120]) regimes, heavy sterile neutrinos produced in nuclear decays [121], and even standard model relic neutrinos [122–124]. Generally speaking, mechanical sensors can be sensitive to any field which couples to either nuclei or to electrons.

#### 1. Wave-like signals

Consider adding to the standard model a new bosonic field which couples to neutrons, protons, and/or to electrons. This field could be a scalar or pseudoscalar $\phi$ (including the axion or dark photon [125]), a vector or pseudovector $V_\mu$, or even a higher spin boson [115]. Sup-

pose this field couples to some specific quantum number that scales with the number of atoms in a mechanical element, say to total neutron, proton or electron number $N_{n,p,e}$. The essential idea is that if we are looking for excitations of this field with wavelength $\lambda = 1/k \gtrsim L$, where $L$ is the size of the mechanical element, then these waves act coherently on the whole mechanical system, driving its center of mass, total angular momentum, or some higher-order phononic modes supported across the bulk of the devices [22, 115–120].

As a concrete example, suppose we have a new massive vector field $A'_\mu$ of mass $m_{A'}$ that couples to neutrons,

$$V_{\rm int} = g \int d^3 \mathbf{x} \slashed{A}' \overline{n} n = g \sum_{i=1}^{N_n} \mathbf{E}' \cdot \mathbf{x}_i. \qquad (148)$$

Here the microscopic coupling involves the vector field $\slashed{A}' = \gamma^\mu A_\mu$ and neutron field $n$. In the second equality, we wrote this coupling in first quantization, where $\mathbf{x}_i$ are the positions of each neutron. In both cases, $g$ is a dimensionless coupling constant. The field $E'_i = \partial_0 A'_i - \partial_i A'_0$ reflects the idea that this coupling is just like the usual coupling of the photon to electrons (i.e., from the minimal coupling $\mathcal{L} = e \slashed{A} \overline{\psi} \psi$ in the Dirac Lagrangian).

The field $A'_\mu$ can be expanded into modes as in Eq. (77). Inserting this expansion into (148) shows that if we are searching modes whose wavelength $\lambda \sim 1/k$ is much larger than the size $L$ of our detector, $kL \ll 1$, then the spatial dependence of the modes drops out, and the new field acts on the center-of-mass $\mathbf{x}$ of the neutrons:

$$V_{\rm int} \approx g N_n \mathbf{E}' \cdot \mathbf{x}_n, \quad \mathbf{x}_n := \frac{1}{N_n} \sum_i \mathbf{x}_i. \qquad (149)$$

Assuming these neutrons are evenly distributed, this means we effectively have a force on the center-of-mass of the solid itself:

$$\mathbf{F}^{\rm in}_{\rm sig}(t) = g N_n \mathbf{E}'(t), \qquad (150)$$

where we can explicitly write the field at the location of the system in analogy with Eq. (133) by expanding into modes as in Eq. (131). We can therefore try to detect the presence of this force by reading out the mechanical element. Notice in particular that the force on the mechanics is proportional to the number of neutrons, i.e., to the total mass of the object. Thus this is essentially an acceleration signal.

As an example application, consider searches for these kinds of fields as dark matter candidates. This parallels the use of resonant cavities for axion and axion-like dark matter discussed in Sec. III C: the signal is a narrowband distribution of long-wavelength quanta at $\omega_s \approx m_\chi$, where $\chi$ is the dark matter candidate. To search for extremely weak couplings $g \ll 1$ we will want to use a detector with mechanical resonant frequency $\omega_m \approx m_{\rm chi}$.

We will continue to use the neutron-coupled vector (149) as a case study. Since dark matter has a velocity $v \sim 10^{-3}$, we can approximate $\mathbf{E}' \approx \partial_0 \mathbf{A}$; the spatial

derivative part is suppressed by $\omega_k/k = v \ll 1$. The power spectrum of the dark matter signal, expressed in force units, is thus

$$S_{FF}^{\rm sig}(\nu) = g^2 N_n^2 S_{EE}^{\rm sig}(\nu) = \nu^2 S_{A'A'}(\nu), \qquad (151)$$

where $S_{A'A'}(\nu)$ is the power spectrum of the DM vector field itself [see Eq. (F5)]. Just like the axion and dark photon examples of Sec. III C, this is a narrowband signal with (unknown) central frequency $\omega_s = m_{A'}$ and quality $Q_s \sim 10^6$. Here we will focus on a single-axis detector, and thus drop the boldface vector notation.

To perform a search for the excess noise power (151), we again follow the general discussion of Sec. II D 2. To estimate a sensitivity, we need to compare the signal (151) to the noise of the device $S_{FF}^{\rm meas}$, which here is given by Eqs. (119) and (120),

$$S_{FF}^{\rm meas} = \frac{S_{YY}^{\rm out}}{|\chi_{YF}|^2}, \qquad (152)$$

where again $\chi_{YF}$ is given in Eq. (117). We then compare the measured power to the expected power and see if there is a significant difference.

For simplicity, consider the limit where the DM signal is narrow compared to the detector; we show a typical example in Fig. 12. Then we can estimate the SNR for excess power in the bandwidth $\Delta\omega_s = m_{A'}/Q_{A'}$, using Eq. (41),

$$
\begin{aligned}
\mathrm{SNR} &= \sqrt{2T_{\rm int}\Delta\omega_s}\,\frac{\overline{S}^{\rm sig}}{S^{\rm meas}} \\
&= \sqrt{2T_{\rm int}m_{A'}/Q_{A'}}\,\frac{g^2 N^2 \nu^2 S_{A'A'}}{S_{FF}} \qquad (153) \\
&\sim \sqrt{T_{\rm int}}\,g^2 m.
\end{aligned}
$$

To get the last scaling relation, note that the numerator scales like $g^2 m^2$ while the denominator scales like $m$ with $m$ the detector mass. Thus, our sensitivity to the coupling scales like $g \sim 1/T_{\rm int}^{1/4} m^{1/2}$, characteristic of an incoherent acceleration signal.

Note that like the axion and dark photon cases discussed in Sec. III C, here we are assuming that $T_{\rm int} \gg T_{\rm coh} \approx m_{A'}/Q_{A'}$, i.e., we integrate for much longer than the time over which a given DM realization will behave more like a coherent wave. This is why we get a $T_{\rm int}^{1/4}$ scaling; we are adding up many such times. If instead we can construct the detector with low enough noise so that a single measurement with $T_{\rm int} \lesssim T_{\rm coh}$ is sufficient to see the signal, the SNR behaves much better, like $T_{\rm int}^{1/2}$, as in Eq. (28).

We show a typical pair of noise and DM signal PSDs for such a search in Fig. 12. Much like the cavity-based searches of Sec. III C, it appears that the best way to look for these signals is to perform a scan over mechanical resonant frequencies. This can be done by, for example, optical spring effects where the mechanical frequency is tunable with a laser [97, 126], and the same basic scan rate formalism of Sec. II D 3 applies.

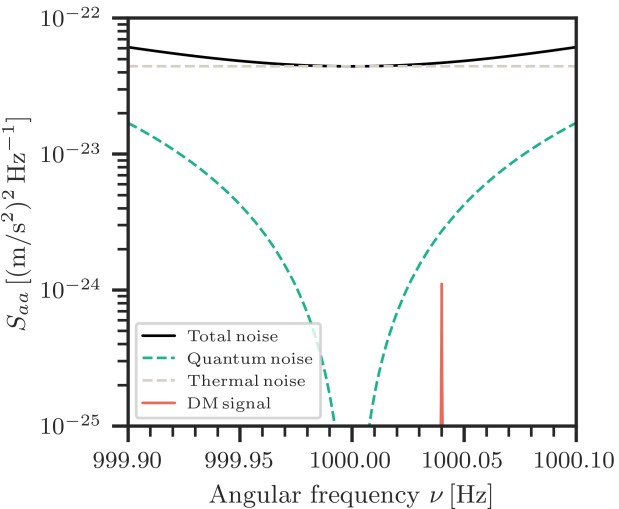

FIG. 12. **Ultralight dark matter search with a mechanical sensor:** Here we show the mechanical sensing analogue to the axion and dark photon plots of Fig. 5. The mechanical noise curves are using the same device parameters shown on the left of Fig. 9, except with $Q_m = 10^7$ for illustration. Note that the detector's noise PSD is effectively much wider than $\omega_m/Q_m$ (the width of the green curve), due to thermal noise broadening. The DM signal PSD is taken assuming a $B - L$ vector boson candidate with $g_{BL} = 10^{-25}$, at the border of current experimental bounds [115, 117, 119].

### 2. Particle-like signals

As a different application of mechanical sensing, we can consider looking for collisions of particles with the mechanical element. Suppose the collision rate is low enough such that in a total measurement time $T_{\rm int}$ (of order days, for example), the signal is almost always zero, while the noise is constantly acting. Thus the best search strategy is to try to resolve the individual collisions, which can be modeled as a sharp function of time

$$F^{\rm sig}(t) \approx \Delta p\,\delta(t), \qquad (154)$$

rather than to integrate for the full $T_{\rm int}$ and perform an excess noise power search. In particle physics slang, we are looking for discrete "bumps" in the output timeseries data.

Following the discussion in Sec. II D 1, the optimal signal processing strategy is with matched filtering. In the simplest case, with a precisely localized signal like (154), and given knowledge of the force noise PSD of our detector $S_{FF}(\nu)$, the optimum filter is simply $f_{\rm opt}(\nu) = 2\pi\Delta p/S_{FF}(\nu)$. The SNR for such an event — representing our ability to resolve a single discrete bump in the output data—is given by Eq. (34). To resolve a given hit at unit signal-to-noise ratio, this equation says that we must have a momentum kick at least as big as

$$\Delta p \geq \Delta p_{\rm thresh} = \left[\int_{-\infty}^{\infty} \frac{d\nu}{S_{FF}(\nu)}\right]^{-1/2}. \qquad (155)$$

This is exactly analogous to a standard energy threshold of a calorimeter, except that here we are reading out the force (or momentum) rather than an energy.[8] In particular, this means we can get directional information, especially with a device constructed to read out along more than one axis.

Using Eq. (155), one can derive a "standard quantum limit" benchmark for impulse sensing [35, 127],

$$\Delta p_{\text{SQL}} = \sqrt{m\omega_m} \approx 1 \text{ keV} \times \left(\frac{m_s}{1 \text{ fg}}\right)^{1/2} \left(\frac{2\pi \text{ kHz}}{\omega_m}\right)^{1/2}. \tag{156}$$

Notice that, converted into an energy threshold using $\Delta E \sim \Delta p^2/m$, this means that optically resolving momentum kicks like this can lead to incredibly small energy thresholds. In the ultimate limit where the mechanical motion is that of a trapped electron, one can see kicks of order $\Delta E \sim 1$ neV [93, 94].

Since we have freedom to shape the noise power $S_{FF}$ using different laser powers, there are multiple ways to achieve (156).

Conceptually, the simplest is to tune the laser so that the force noise SQL occurs on resonance $\omega_* = \omega_m$ [cf. Eq. (125)]. In this case, the noise is sharply minimized on the mechanical resonance $\omega_m$, with $S_{FF} \approx 2m\gamma\omega_s$ within a mechanical linewidth $\gamma$ (see Fig. 13). Integrating over this linewidth in (155) gives (156). For high-frequency resonators where the thermal noise does not swamp the quantum noise, this is a good strategy.

However, for low-frequency resonators, technical noise (particularly thermal noise) can render this strategy invalid, by washing out the sharp resonance feature. For example, consider Fig. 9: in the left figure, the noise on mechanical resonance is dominated by thermal noise, which washes out the sharp resonance. On the other hand, in the right figure, where the laser is tuned so that $\omega_* \gg \omega_m$, the thermal noise is everywhere subdominant. This is the limit in which the sensor is approximately a free particle, used in typical ground-based GW detectors (see Sec. IV B 2). Since our signal (154) is uniformly distributed over frequency space, we can work at these high frequencies and integrate for a shorter time $\tau \sim 1/\omega_*$. Expanding Eq. (E24) around $\nu = \omega_*$, one finds that $S_{FF}(\nu) \approx m\omega_*(1 + \nu^4/\omega_*^4)$. Inserting this into (155) and integrating over a bandwidth of order $\omega_*$, one obtains

$$\Delta p_{\text{thresh}} = \sqrt{m\omega_*}, \tag{157}$$

which is just the SQL (156) evaluated at $\omega_*$. The integration time required for this $\tau \sim 1/\omega_*$ is much shorter than the ringdown measurement $\tau \sim 1/\gamma_m = Q_m/\omega_m$, so much less additional technical noise builds up. The cost, however, is a much higher laser power.

---

[8] Note that readout of energy and momentum are not equivalent in quantum mechanics: for a harmonic oscillator, $[H, p] \neq 0$. This is analogous to the distinction between linear (quadrature) readout and quadratic (intensity) readout when measuring photons.

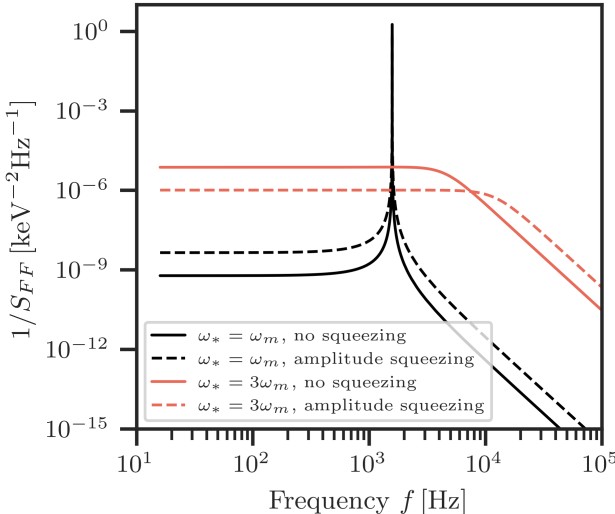

FIG. 13. **Impulse threshold of a mechanical sensor:** here we show how different laser power choices affect the momentum threshold of a mechanical detector, as in Eq. (155). This is shown for a detector of mass $m = 1$ fg (i.e., a $\sim 100$ nm radius solid sphere) trapped at 100 kHz, as in [27, 28, 114, 121].

Much like other SQLs, Eq. (156) is not fundamental. We discuss ways to obtain even lower thresholds by engineering the quantum noise in Sec. V.

## V. QUANTUM NOISE: BEYOND THE SQL

We have seen in the previous sections that measuring observables with non-zero variance in the vacuum state leads to noise at the level of the Standard Quantum Limit, in the sense of Eq. (7). In this section, we will give some examples of how this noise may be reduced.

Before we begin, we emphasize that reduction of quantum noise below the SQL is only important if one is already at the point where classical, technical noises are subdominant to the SQL. In particular, this is a moving target: once one starts reducing the quantum noise, it may well be that the experiment again becomes technically limited. For example, with sufficient noise reduction, Advanced LIGO will become limited again by mirror coating losses [4]. Moreover, all of the techniques discussed below require the use of fragile quantum states and are thus limited by loss or error mechanisms. Put together, these issues reflect the need for simultaneous improvements both in quantum measurement protocols as well as the classical engineering which implements them.

In this section, we will largely focus on the technique which is most widely implemented in practice: the use of squeezed light [23, 128]. We will then briefly outline some other, currently less established techniques: quantum non-demolition measurements and their use in back-action evasion measurements [9], and the use of multiple

entangled probes [25].

## A. Squeezed light

The first method of quantum noise reduction we will treat is the use of (single-mode) squeezing in the readout system, i.e., in the optical or microwave fields sent into and read out of the detectors. Since the first observation of squeezed light in 1985 [129], great strides have been made in the experimental generation and control of these states [130] and they have already been used in practice in both axion searches [7] and in ground-based gravitational wave detectors [12, 13, 131].

The basic idea of squeezing is simple. Consider a detector made of a simple harmonic oscillator, for example a cavity mode as in Sec. III. Its ground state, or any coherent state, has a Gaussian wavefunction with noise equally weighted between the two quadratures $X$ and $Y$

$$\Delta X = \Delta Y = \frac{1}{\sqrt{2}}, \qquad (158)$$

which saturates the Heisenberg uncertainty relation $\Delta X \Delta Y = 1/2$. There are other Gaussian states which also saturate Heisenberg uncertainty, but with unequal weights of the quadratures, for example one can find a family of Gaussian states with

$$\Delta X = \frac{e^{2r}}{\sqrt{2}}, \quad \Delta Y = \frac{e^{-2r}}{\sqrt{2}}, \qquad (159)$$

where $r \geq 0$ is real. These are the (phase) "squeezed vacuum states", defined in detail below. Squeezed states of light are distinctly non-classical, in a precise sense: they produce observable signatures which, unlike coherent states, are inconsistent with any classical model of radiation [23, 69, 132–134].

The simplest use of these states in detection problems occurs when we can imprint the whole signal on the squeezed variable ($Y$, in this example). It then becomes possible to make more precise measurements of the signal, since the detector is less noisy in this variable. In particular, keeping with our definition of the SQL as sensitivity at the level of vacuum fluctuations, this means one can achieve sensitivities *better* than the SQL. In practice, as we will see, this is not always possible. However, there are also more sophisticated uses for these states, including lowering the power requirements on lasers and increasing the rate over which one can scan for narrowband signals.

Squeezed states of a single mode can be efficiently described using the unitary operator

$$S_1(\xi) = \exp\left(\frac{1}{2}(\xi^* a^2 - \xi a^{\dagger 2})\right), \qquad (160)$$

where the mode has creation operator $a^\dagger$, and $\xi = re^{i\theta}$, is a complex parameter. It is common to refer to $r$ as "the" squeezing parameter and $\theta$ as the squeezing angle, which determines the quadrature that will be squeezed.

We will consider two main classes of squeezed readout states: the squeezed vacuum, and squeezed coherent states. Given a coherent state defined by $a |\alpha\rangle = \alpha |\alpha\rangle$, the squeezed coherent states are denoted

$$|\alpha, \xi\rangle = S_1(\xi) |\alpha\rangle. \qquad (161)$$

When acting on the input fields $X^{\text{in}}, Y^{\text{in}}$, both parameters $r, \theta$ can generically be frequency-dependent. The squeezed vacuum has noise PSDs are (see Appendix A 2 for the calculations)

$$\begin{aligned}
S_{XX}^{\text{in}} &= \frac{1}{2}[\cosh 2r - \cos\theta \sinh 2r], \\
S_{XY}^{\text{in}} &= S_{YX}^{\text{in}} = -\frac{1}{2}[\sinh 2r \sin\theta], \qquad (162) \\
S_{YY}^{\text{in}} &= \frac{1}{2}[\cosh 2r + \cos\theta \sinh 2r].
\end{aligned}$$

The limit $r \to 0$ recovers the vacuum noise PSDs [Eq. (A22)]. Notice in particular that, unlike the vacuum, this state has cross-correlations $S_{XY} \neq 0$ between the two quadratures. Moreover, $S_{XY}$ can be negative, unlike $S_{XX}$ and $S_{YY}$. If we instead are sending in a squeezed coherent state—for example, a laser drive $\alpha$ sent through a squeezing apparatus—then the same vacuum PSDs apply to the fluctuations around this drive.

A thorough discussion of the physical generation of squeezed states is beyond the scope of this review. However, the basic idea is relatively straightforward: we want to realize (160) as $U = e^{-iHt}$, where $H$ is some Hamiltonian. While this is accomplished differently for microwave [10, 135] and optical frequencies [129], the essential technique [136] is to couple the electromagnetic mode to a medium with a non-linear interaction, e.g.,

$$H = \omega a^\dagger a + \omega_p b^\dagger b + i\chi^{(2)}(a^2 b^\dagger - a^{\dagger 2} b), \qquad (163)$$

where $\omega_p$ is the frequency of a third mode. The subscript $p$ is because one "pumps" this mode by injecting a strong drive $b \to \beta e^{-i\omega_p t}$, where $\beta$ is the drive amplitude. Transforming into the interaction picture, and choosing $\omega_p = 2\omega$, we obtain a time-independent interaction Hamiltonian

$$H_I = i(\eta^* a^2 - \eta a^{\dagger 2}), \qquad (164)$$

which yields the evolution operator

$$U_I(t) = \exp\left(-iH_I t\right) = \exp\left(\eta^* t a^2 - \eta t a^{\dagger 2}\right), \qquad (165)$$

where $\eta = \chi^{(2)}\beta$. This is just (160), with $\xi = 2\eta t$. Physically, this mechanism (called parametric down conversion) amounts to the conversion of pump photons into pairs of the desired mode excitations.

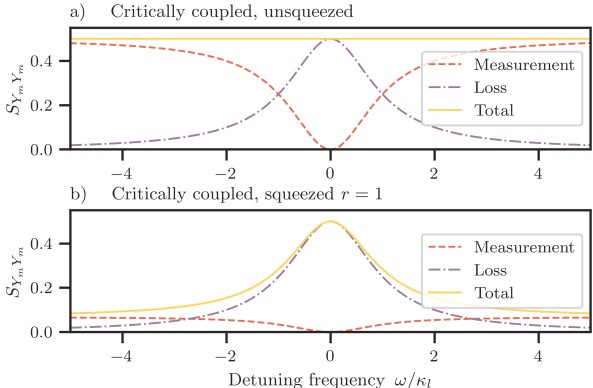

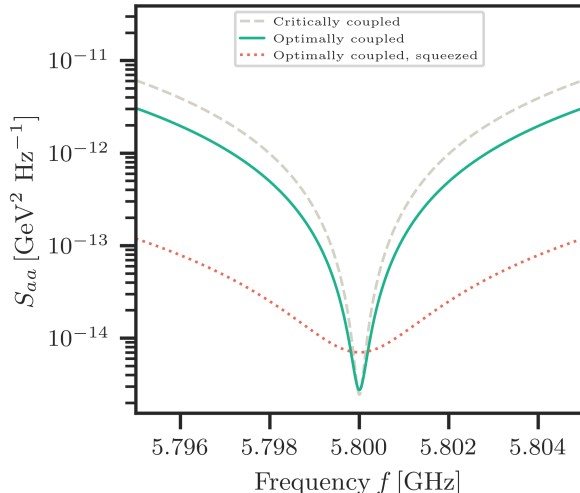

FIG. 14. **Squeezed vacuum in a cavity haloscope:** Here we show the contributions to the output noise of the measurement port's phase quadrature, as given by Eq. (166). The two scenarios are $a$) a critically coupled $\kappa_l = \kappa_m$, zero-temperature cavity with no squeezing; and $b$) a critically coupled, zero-temperature cavity with phase squeezing. Here the frequencies are given in the rotating frame, i.e., are zeroed at the cavity resonance.

### 1. Squeezed vacuum in a cavity haloscope

Consider phase squeezing the vacuum going into the measurement port, such that the output phase quadrature of the measurement port $Y^{\text{out}}$ has noise

$$
\begin{aligned}
S_{mm}^{\text{SMS}} &= |\chi_{mm}|^2 S_{mm}^{\text{in}} + |\chi_{ml}|^2 S_{ll}^{\text{in}} \\
&= \left(\frac{1}{2} + n_{\text{th}}\right)\left(e^{-2r}|\chi_{mm}|^2 + |\chi_{ml}|^2\right)
\end{aligned}
\tag{166}
$$

where we have used the expression for $S_{YY}^{\text{in}}$ in Eq. (162) with $\theta = \pi$. Squeezing the vacuum in this way reduces the noise contribution from the measurement port by a multiplicative factor. As can be seen in Fig. 14, this creates noise reduction across a band of frequencies, although the noise PSD at the cavity resonance is unchanged. As discussed in Sec. III C, searching for narrowband dark matter signals with cavities requires scanning over a range of cavity frequencies, thus this broadening of the noise PSD allows a larger range of axion masses to be probed in a fixed time [46], as has been demonstrated in the HAYSTAC experiment [77, 137].

Note that the cavity susceptibilities $\chi_{mm}$ and $\chi_{ll}$ depend on both the loss rate $\kappa_l$ as well as the measurement rate $\kappa_m$. The measurement rate is an experimental parameter that may be optimised over. If we wish to maximise the on-resonance sensitivity, differentiating Eq. (166) with respect to $\kappa_m$, using Eq. (58), gives the minimal on-resonance noise for $\kappa_m^* = \kappa_l$. Such a choice of measurement rate is called *critical coupling*. We plot the PSD for a critically coupled cavity in Fig. 14, for both the squeezed and non-squeezed case. Since $\chi_{mm}(0) = 0$ for a critically coupled cavity, the measurement port decouples on-resonance, and so the peak sensitivity is un-

FIG. 15. The noise on the axion estimator $S_{aa}$ for a critically and optimally coupled cavities, in both the squeezed ($r = 1, \theta = \pi$) and unsqueezed cases.

altered by squeezing. Instead, we see that squeezing decreases the noise away from resonance, thus increasing the bandwidth of the cavity.

If, instead of maximising the on-resonance sensitivity, we want to maximise the scan rate of the cavity, then Eq. (48) shows us that we need to maximise the bandwidth

$$
B = \int d\nu \left(\frac{1}{S_{FF}^{\text{SMS}}(\nu)}\right)^2 = \int d\nu \left(\frac{|\chi_{mF}|^2}{S_{xx}^{\text{SMS}}(\nu)}\right)^2.
\tag{167}
$$

In this case, we may directly calculate the optimal coupling $\kappa_m^*$ in the same manner, by solving $dB/d\kappa_m = 0$. One finds that

$$
\kappa_m^* = \frac{-1 + 2e^{2r} + \sqrt{4e^{4r} - 4e^{2r} + 9}}{2}\kappa_l.
\tag{168}
$$

If $r \geq 0$, it is in fact better to be overcoupled (meaning $\kappa_m > \kappa_l$), with $\kappa_m^* \to 2\kappa_l$ in the no-squeezing, $r = 0$ limit, while in the large-squeezing limit we optimally have $\kappa_m^* \to 2e^{2r}\kappa_l$. We plot the axion noise PSD for such optimally coupled cavities in both the squeezed and unsqueezed case in Fig. 15. We see that the act of squeezing and overcoupling slightly reduces the on-resonance sensitivity, with the benefit that the bandwidth is greatly enhanced.

Using an optimally coupled measurement port with $\kappa_m = 2e^{2r}\kappa_l$, the bandwidth of Eq. (167) scales as

$$
B = \frac{27}{(12 - 4e^{-2r} + e^{-4r})^{3/2}}e^{2r}B_0,
\tag{169}
$$

where $B_0$ is the bandwidth without squeezing. By comparison with Eq. (89) for the scan rate of a low-$Q$ cavity, which is linear in the bandwidth, we see two ways of accelerating such an axion haloscope: increase the squeezing

amplitude $r$, or decrease the intrinsic loss rate $\kappa_l$ of the cavity. If instead the cavity bandwidth is narrow compared to the signal bandwidth, then the SNR of Eq. (42), as well as the scan rate Eq. (94), are proportional to $B$, and so both increase with squeezing as $e^{2r}$ at large $r$.

### 2. Squeezed coherent light in a mechanical force sensor

Consider our model for an optomechanical force sensor based on a single Fabry-Perot cavity, as in Sec. IV A. The output force noise due to the light (i.e., the shot noise and back-action), with a coherent laser drive, was given in Eq. (123). Suppose now that we used a squeezed light source instead. This changes the input noise PSDs from the vacuum results to those of Eq. (162), and in particular generates a cross-term $S_{YX}^{\mathrm{in}}$. Using the general expression for the output phase noise (119) and converting it to a force noise with the appropriate transfer function, we find a new cross-correlation term in the force PSD:a

$$
\begin{aligned}
S_{FF}^{\mathrm{quantum}} =& \frac{|\chi_{YY}|^2}{|\chi_{YF}|^2} S_{YY}^{\mathrm{in}} + \frac{|\chi_{YX}|^2}{|\chi_{YF}|^2} S_{XX}^{\mathrm{in}} \\
& + \frac{2\mathrm{Re}\,\chi_{YX}\chi_{YY}^*}{|\chi_{YF}|^2} S_{YX}^{\mathrm{in}}.
\end{aligned}
\tag{170}
$$

As emphasized above, the last cross-correlator term is not necessarily positive, unlike those in the first line. In other words, it can be used to reduce noise.

A simple application of this idea, however, shows that this can be a bit subtle. Consider first tuning the input laser power so that the noise at some reference frequency $\omega_*$ is at the SQL, as in Sec. IV A. Notice that the third term in Eq. (170) is independent of the optomechanical coupling strength $g$, so it does not affect the optimal value $g_*$. However, the two autocorrelators $S_{YY}^{\mathrm{in}}, S_{XX}^{\mathrm{in}}$ are also modified in the squeezed state and no longer equal, so $g_*$ does depend on the squeezing parameters. A simple calculation gives

$$
g_*^2 = \frac{\sqrt{S_{YY}^{\mathrm{in}}(\omega_*)/S_{XX}^{\mathrm{in}}(\omega_*)}}{2\kappa|\chi_c(\omega_*)|^2|\chi_m(\omega_*)|}.
\tag{171}
$$

Plugging this back in gives the quantum noise, at frequency $\omega_*$,

$$
\begin{aligned}
S_{FF}^{\mathrm{quantum}}(\omega_*) =& \\
\frac{\sqrt{S_{YY}^{\mathrm{in}}(\omega_*)S_{XX}^{\mathrm{in}}(\omega_*)}}{|\chi_m(\omega_*)|} &+ m(\omega_m^2 - \omega_*^2)S_{YX}^{\mathrm{in}}(\omega_*).
\end{aligned}
\tag{172}
$$

To obtain the last line, we used the explicit transfer functions (117). This expression is only at the SQL frequency $\omega_*$; we give the full PSD in Appendix E 3.

Consider first choosing $\omega_* = \omega_m$, i.e., optimizing the shot noise-back-action tradeoff on the mechanical resonance frequency, as in the left plot of Fig. 16. We see that the cross-correlator in Eq. (172) will not contribute to the output force noise on resonance. Moreover, as one

can verify using Eq. (162), the term in the square root has a minimum value of $1/4$, at which point it saturates Heisenberg uncertainty in the $X, Y$ quadratures. Thus, comparing to Eq. (125), we see that this noise floor is never lower than the un-squeezed SQL, at least on the mechanical resonance frequency. However, the PSD off resonance can be improved by using phase-squeezed input light. This is consistent with the fact that, in this example, the PSD is dominated by shot noise everywhere away from the mechanical resonance (see Fig. 9).

In the high-frequency, "free mass" regime, we can choose instead $\omega_* \gg \omega_m$. See the right plot of Fig. 16. In this example, the noise above resonance is dominated by quantum noise rather than thermal noise (again see Fig. 9), so we can affect a broadband reduction of the noise by injecting squeezed light. We will see the same basic behavior in the case of a Michelson interferometer for gravitational wave detection in the next section.

In addition to these improvements to noise power, the use of squeezed light has an important "technical" advantage: it can reduce the amount of laser power required to reach a given noise level. This is true even in the case where squeezing cannot reduce the PSD itself below the SQL. For example, with $\omega_* = \omega_m$, the laser power to obtain (172) is $P_L \sim g_*^2 \sim e^{-2r}g_*^2|_{\mathrm{no\ squeezing}}$. This is important in systems where too much incident power can cause problems like melting of the target surface [96] (or even the entire target [138]).

### 3. Squeezed vacuum in a Michelson interferometer; Advanced LIGO

In a Michelson interferometer, there are two input ports ($c$ and $d$ in Fig. 10), and we can control the input states to both of these. It turns out that a very useful reduction in noise can be achieved by sending in the usual coherent state (laser) into one port ($c$) but sending squeezed vacuum into the dark port ($d$), the port which is actually measured [2]. This technique has now been implemented in Advanced LIGO [13].

Similar to the Fabry-Pérot case, the use of squeezed light in the dark port changes the measured noise power spectrum, and can in fact lower it (compared to vacuum input) over a certain frequency band. We emphasize first that here the laser power is tuned so that the shot noise-back-action equality ("SQL") is achieved at a frequency $\omega_* \gg \omega_m$ well above the frequency of the mirror's pendulum motion, so it is possible to reduce the noise below this SQL level. The detailed way in which the PSD changes depends on the details of the squeezed state, in particular its frequency dependence, as we now explain.

We follow the same notation as in Sec. IV B 3. In Eq. (144), we gave the noise power for the measured variable, the phase output $Y^{\mathrm{out}}$ coming out of the dark port, assuming that the fluctuations into both the $c, d$ ports were the vacuum. More generally, assuming arbitrary (but uncorrelated) states are sent into $c, d$, one finds by

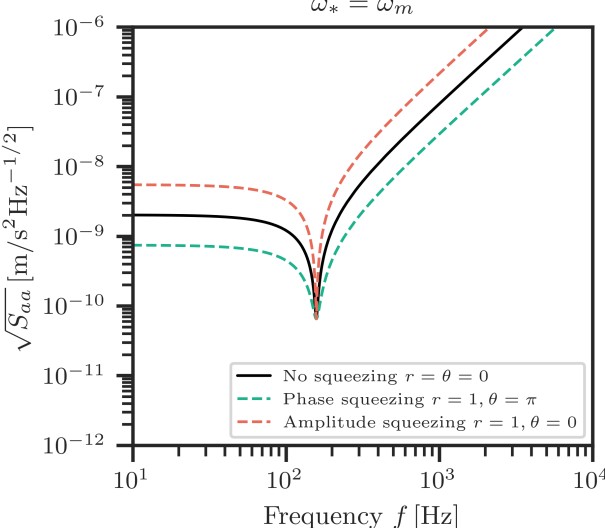 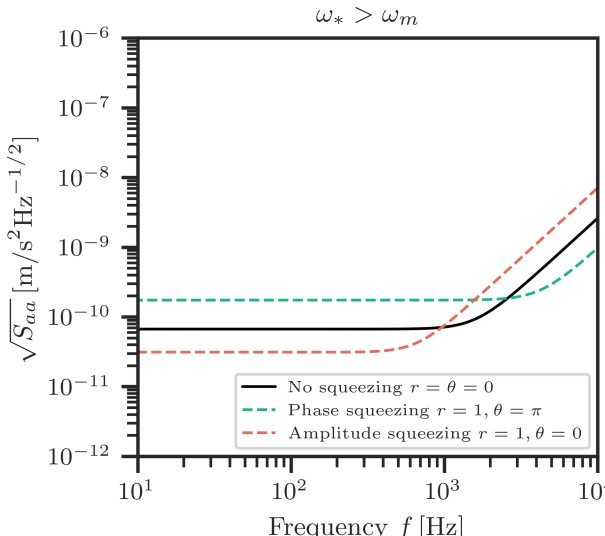

FIG. 16. **Mechanical force sensing with squeezed light.** Noise PSDs for a prototypical cavity optomechanical device, with different choices of the laser power [Eq. (171)], and assuming squeezed input light [Eq. (170)]. Left: SQL on resonance $\omega_* = \omega_m$. Right: SQL above resonance $\omega_* \gg \omega_m$ ("free particle limit"). We plot this as acceleration sensitivity, which is obtained from the force PSD by $S_a = \sqrt{S_{FF}/m^2}$, thus coming with units of (acceleration)$/\sqrt{\text{Hz}}$. Same device parameters as in Fig. 9. The explicit PSD is given in Eq. (E25).

a straightforward calculation analogous to the one presented in Sec. IV B 3 that the output noise is given by

$$
\begin{aligned}
S_{YY}^{\text{out}} = {}& \frac{1}{2} |\chi_{YY}|^2 \left[ S_{YY}^{\text{in},c} + S_{YY}^{\text{in},d} - \left( S_{YY}^{\text{in},c} - S_{YY}^{\text{in},d} \right) \right] \\
& + \frac{1}{2} |\chi_{YX}|^2 \left[ S_{XX}^{\text{in},c} + S_{XX}^{\text{in},d} - \left( S_{XX}^{\text{in},c} - S_{XX}^{\text{in},d} \right) \right] \\
& + \text{Re } \chi_{YY}\chi_{YX}^* \left[ S_{YX}^{\text{in},c} + S_{YX}^{\text{in},d} - \left( S_{YX}^{\text{in},c} - S_{YX}^{\text{in},d} \right) \right].
\end{aligned}
\tag{173}
$$

As a sanity check, notice that if both $c, d$ ports have input vacuum, the terms in parentheses are all zero and this reduces to the simple result given in Eq. (144). More generally, however, we see that the beamsplitter can create correlations between the arms and these do not necessarily cancel.

Now we can specialize to the case with vacuum input to $c$ and squeezed vacuum input to $d$. There, the terms in parentheses are no longer zero since the two noise PSDs come with different gain factors. However, we see that all the $c$ terms cancel with each other, and we are left with

$$
S_{YY}^{\text{out}} = |\chi_{YY}|^2 S_{YY}^{\text{in},d} + |\chi_{YX}|^2 S_{XX}^{\text{in},d} + 2\text{Re } \chi_{YY}\chi_{YX}^* S_{YX}^{\text{in},d}.
\tag{174}
$$

This is identical to the result for a simple input coherent squeezed state injected into a single arm Fabry-Pérot, as in Eq. (172). In this expression, the explicit forms of the input PSD are given in Eq. (162). The strain PSD can be computed as usual by defining the strain estimator (145) and dividing through by the appropriate response

functions,

$$
S_{hh}(\nu) = \frac{1}{m^2 L^2 \nu^4} \frac{1}{|\chi_{YF}(\nu)|^2} S_{YY}^{\text{out}}(\nu),
\tag{175}
$$

where we remind the reader that the various transfer functions are collected in Eq. (117).

In Fig. 17, we show how the strain PSD (175) depends on the choice of squeezing parameter $r$ and squeezing angle $\theta$. In this figure, the laser is tuned so that the SQL is achieved at $\omega_* = 100$ Hz, well above the mechanical resonance. Compared to the noise curve with vacuum input (labeled "SQL"), we see that pure phase squeezed light $\theta \equiv 0$ reduces the noise above $\omega_*$ at the cost of *increasing* the noise below $\omega_*$. Pure amplitude squeezing $\theta = \pi$ has the opposite behavior. This reflects the fact that the overall noise is dominated by shot noise (input phase fluctuations) above $\omega_*$ but dominated by back-action noise (input amplitude fluctuations) below $\omega_*$. To get a truly broadband reduction in noise, we see that we need to somehow engineer an input state where $r = r(\nu), \theta = \theta(\nu)$, a *frequency-dependent squeezed state* [107, 139].

To understand the type of frequency dependence we need in more detail, it is convenient to re-write the strain PSD somewhat more explicitly. Let

$$
\mathcal{K} = \mathcal{K}(\nu) = |\chi_{YX}(\nu)|.
\tag{176}
$$

For measurement frequencies $\nu \gg \omega_m$, we can approximate all the transfer functions by the free particle limit

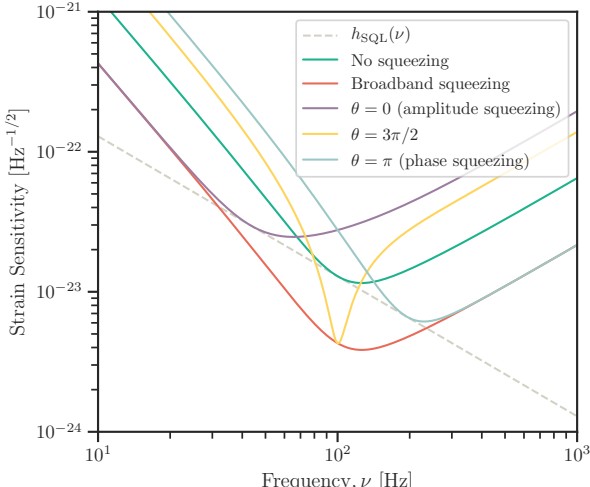

FIG. 17. **Squeezed vacuum in a Michelson interferometer:** By injecting squeezed vacuum into the dark port of an interferometer, the standard quantum limit can be surpassed. This figure demonstrates the strain sensitivities obtained with no squeezing (green curve), fixed squeezing angles (purple, yellow, and teal curves obtained by setting $\theta$ to the appropriate value in Eq. (177)), and broadband squeezing (red curve). Same detector parameters as in Fig. 11.

$\omega_m \to 0$, and the strain PSD (175) reduces to

$$S_{hh} = \frac{h_{\mathrm{SQL}}^2}{2}\left(\mathcal{K} + \frac{1}{\mathcal{K}}\right)\left[\cosh 2r - \cos\left(\theta + \Phi\right)\sinh 2r\right],$$

(177)

where $\theta = \theta(\nu), r = r(\nu)$ are frequency dependent, $h_{SQL}(\nu)$ was given in Eq. (147), and

$$\Phi := 2\arctan\frac{1}{\mathcal{K}}.$$

(178)

We give the detailed steps in Appendix E 4. From this result we see that in the free-mass limit, the best case would be to engineer a frequency dependent squeezing angle that satisfies

$$\theta(\nu) = -\Phi(\nu),$$

(179)

so that the strain PSD becomes simply

$$S_{hh}(\omega) = \frac{h_{\mathrm{SQL}}^2}{2}\left[\mathcal{K} + \frac{1}{\mathcal{K}}\right]e^{-2r}.$$

(180)

This is shown as the red curve labeled "broadband" in Fig. 17.

A detailed discussion of the generation of frequency-dependent squeezed vacuum is beyond the scope of this review. However, we note that proof-of-principle experiments that utilize external filter cavities to achieve frequency-dependent (broadband) squeezed light was demonstrated in [14], and frequency-dependent squeezed light has now been used in Advanced LIGO [131].

## B. Quantum non-demolition; back-action evasion

In Sec. II A, we gave a classical argument for the SQL (7), which gives the precision with which we can perform repeated position measurements in time. Given that this argument is based only on the Schrödinger equation and Heisenberg uncertainty, it seems unlikely that we could improve the noise bound [2].

However, consider the reason for measuring the change in position $x(t_2) - x(t_1)$: we want to determine the magnitude of a force acting on the object. For small times we can approximate the action of the force as $F \approx m(x(t_2) - x(t_1))/\Delta t^2$, which from (7) translates to an SQL on force measurements

$$\Delta F_{\mathrm{SQL}} = \sqrt{\frac{m}{\Delta t^3}}.$$

(181)

How could we do better? One of many ways is to note that here we are inferring the force by measuring the position at two different times. The problem we ran into is that the uncertainty in $x$ is not preserved under time evolution, as in Eq. (6). But what if we measured instead the *momentum* variable, $p$? Since $[H, p] = 0$ during free evolution, we also have $[H, p^2] = 0$, i.e., the variance of the momentum is conserved. Thus if we make a very accurate measurement of $p_1 = p(t_1)$, then even if $\Delta p_1 \to 0$, the measurement in the second time step will be just as accurate $\Delta p_2 = \Delta p_1$. Since the force changes $p \to p + F\Delta t$, we see that we can measure the force *arbitrarily well* $\Delta F \to 0$, clearly beating the SQL [16].

The above technique is referred to as either a quantum non-demolition measurement (QND), a back-action evading (BAE) measurement, or both [9]. Historically, the QND term referred to measurement of a degree of freedom that commutes with the sensor Hamiltonian—in our free particle example, $[H, p] = 0$. The BAE term refers to the fact that in this measurement, we have eliminated the measurement back-action—the measurement in the first step does not increase the uncertainty of the measurement in the second step.

In this short section, we give a bit more detail by highlighting two examples of this general idea which have seen some use in practice. The first is based on dispersive readout of a qubit, a technique widely used in microwave-based quantum computing architectures [140, 141] and recently applied to a dark matter search [8]. We then give more details of the momentum measurement discussed above, which in the gravitational wave community has been called a speedmeter.

### 1. Dispersive qubit measurements

Consider a qubit with Pauli matrices $\sigma_{x,y,z}$ placed in a cavity with mode $a$, and coupled via

$$H = \omega_c a^\dagger a + \omega_q \sigma_z + g_0 a^\dagger a \sigma_z.$$

(182)

This is called a "dispersive coupling". Note that in the interaction term, the relevant qubit and cavity operators commute with their respective free Hamiltonians. Thus, one can actually use this system as either a QND measurement of the cavity photon number $a^\dagger a$ by reading out the qubit, or a QND measurement of the qubit $\sigma_z$ state by reading out the cavity mode. In a dark matter search with cavities, the goal is to perform QND measurement of the photon number in a science cavity [8]; to readout the qubit, one can further couple a second cavity to the qubit, and use an input-output line connected to this "readout" cavity to perform the measurement on the qubit state, which then affects a measurement of the "science" cavity's photon number.

Let us then examine the measurement of the qubit state in more detail. Coupling the cavity to an input-output system like a transmission line, we can use input-output theory to see how the signal down the line encodes the state of the qubit. Including the input fields, driving the input line $a^{\rm in} \to \alpha_0 + a^{\rm in}$, and moving to the frame co-rotating with the drive (see Appendix E), the Hamiltonian becomes essentially $H_{\rm driven} = \omega_q \sigma_z + g X \sigma_z$, plus the bath terms. Here $g = g_0|\alpha_0|$ is the drive-enhanced coupling, which we can in principle make arbitrarily large with a large drive. In the usual input-output approximations, the equations of motion become

$$
\begin{aligned}
\dot{X} &= -\frac{\kappa}{2} + \sqrt{\kappa} X^{\rm in} \\
\dot{Y} &= -\frac{\kappa}{2} + \sqrt{\kappa} Y^{\rm in} + g\sigma_z \\
\dot{\sigma}_z &= 0.
\end{aligned}
\tag{183}
$$

We are ignoring any intrinsic noise on the qubit itself for simplicity.

We can use these equations of motion to solve for the output fields, as usual. The key point is that a measurement of the output homodyne signal, say,

$$
Y^{\rm out} = e^{i\phi_c} Y^{\rm in} + g\chi_c \sigma_z, \tag{184}
$$

encodes the qubit state ($\sigma_z$) and includes noise from the input phase $Y^{\rm in}$ but not the conjugate $X^{\rm in}$. In particular, the signal of interest is proportional to $g$, which we can make large with a strong drive, while the noise $S_{YY}^{\rm in}$ is fixed. Thus, we can read out the qubit state with arbitrarily high signal-to-noise ratio. This should be contrasted with, for example, the optomechanics PSDs [Eqs. (119), (120)], where noise from both $Y^{\rm in}$ (shot noise) as well as $X^{\rm in}$ (back-action noise) enter. For this reason, this measurement can be viewed as back-action evading.

### 2. Speedmeters

Next, consider a cavity optomechanical system like that studied in Sec. IV. Following the heuristic discussion at the beginning of this section, we could try to make a non-demolition version of this system, instead of

the usual optomechanical coupling $V = g_0 x a^\dagger a$ between the mechanics and cavity mode, we would like to engineer a coupling $V' = g_0' p a^\dagger a$. We will say a bit more about how one might attempt to do this at the end of this section, but for now suppose that we can literally get a momentum coupling of this form [16, 142].

The equations of motion for the system, given in Eq. (114) for the standard position-coupled case, now become

$$
\begin{aligned}
\dot{X} &= -\frac{\kappa}{2} X - \sqrt{\kappa} X^{\rm in}, \\
\dot{Y} &= -\frac{\kappa}{2} Y - \sqrt{\kappa} Y^{\rm in} - g'p, \\
\dot{x} &= \frac{p}{m} + g' X, \\
\dot{p} &= -m\omega_m^2 x - \gamma p + F^{\rm in}.
\end{aligned}
\tag{185}
$$

Here we have driven the cavity as usual, so $g' = g_0'|\alpha_0|$ with $\alpha_0$ the coherent laser drive, assumed zero laser-cavity detuning $\Delta = 0$, and have written these in the frame corotating with the cavity.

While these equations look superficially similar to those of the usual position coupling (114), they have very different properties. Solving these equations in the frequency domain and witing the usual input-output relations, the output phase is given by

$$
Y^{\rm out} = \chi_{YY}' Y^{\rm in} + \chi_{YX}' X^{\rm in} + \chi_{YF}' F^{\rm in}, \tag{186}
$$

where now

$$
\begin{aligned}
\chi_{YY}' &= e^{i\phi} \\
\chi_{YX}' &= g'^2 \kappa m^2 \omega_m^2 \chi_c^2 \chi_m \\
\chi_{YF}' &= i g' \sqrt{\kappa} m\nu \chi_c \chi_m.
\end{aligned}
\tag{187}
$$

Forming the force estimator $F_E = Y^{\rm out}/\chi_{YF}'$ as usual, we can compute the noise PSD

$$
\begin{aligned}
S_{FF} &= S_{FF}^{\rm in} + \frac{1}{g'^2 \kappa m^2 \nu^2 |\chi_c|^2 |\chi_m|^2} S_{YY}^{\rm in} \\
&\quad + g'^2 \kappa m^2 \frac{\omega_m^4}{\nu^2} |\chi_c|^2 S_{XX}^{\rm in}.
\end{aligned}
\tag{188}
$$

We have highlighted the $\omega_m^4/\nu^2$ dependence in the last term explicitly; it is the key ingredient to the back-action evasion of this system.

Recall that in our discussion of the optomechanical force sensing SQL in Sec. IV, the SQL at frequency $\nu = \omega_*$ was achieved by setting the laser power, or equivalently the drive-enhanced coupling $g$, so that the shot noise ($\sim S_{YY}/g^2$) and back-action ($\sim g^2 S_{XX}$) terms were equal at some frequency $\omega_*$. In this momentum-coupled system, on the other hand, we see that at frequencies $\omega_m/\nu \ll 1$ *the back-action noise term drops out*, at least to second order in $\omega_m/\nu \ll 1$. Thus the back-action is highly suppressed, and we can turn up the laser power to send $g'$ as high as we can manage, which in turn reduces the shot noise. Note that $\omega_m/\nu \ll 1$ is precisely the limit in which the mass responds like a free

mass and we can neglect its harmonic motion, so this is a detailed realization of the discussion of the QND momentum measurement suggested at the beginning of this section.

Realizing this type of back-action evasion is a subject of active research. In practice, there are a few key difficulties. One is that, for terrestrial gravitational wave detectors, the target signal is usually in the $\nu \sim 100 - 1000$ Hz regime, while $\omega_m \sim$ few Hz, so the back-action evasion is limited even in principle.

More importantly, understanding how to actually engineer a momentum coupling has proven to be tricky [16]. At present, the best-understood proposal is to have the input laser interact with the mechanical element not once but twice, with opposite phase and a short time delay, which enables a differential position measurement and thus an approximate momentum measurement [20, 35, 142, 143]. This, much like squeezed light, is limited by optical losses while the photons are traversing the setup. Substantial room exists for new protocols and experiments in this direction.

### C. Entanglement

The theory of quantum entanglement and its experimental quantification is a rich area of active research [144, 145]. In addition to the fundamental implications entanglement has for our understanding of reality [146], it has been shown in many settings that entanglement is a resource that allows quantum sensors to surpass various Standard Quantum Limits. Here we aim to outline a few simple ideas along these lines.

Before we begin, note that we have already implicitly discussed the use of entangled states! In Sec. V A, we saw that single-mode squeezed light, which is not entangled, can enable advantages in measurement sensitivities. However, when a squeezed vacuum is injected into the dark port of a Michelson interferometer, the state which comes out of the initial beamsplitter—that is, the state which is injected into the two Fabry-Pérot arms, see Fig. 10—is itself an entangled state of the two beams. (We give an explicit proof of this below). Thus there is not really a sharp distinction between "using entanglement" and "using squeezing". In fact, at least at present, this squeezed state injection is the best way in which entanglement can be used to enhance a practical experiment.

In what follows, we highlight a few other ways one can conceivably leverage entanglement. We begin with a review of a protocol commonly invoked in quantum metrology, involving entangled states of input light sent into a single device. We then briefly outline a more exotic idea, involving the injection of entangled states into multiple sensing devices and thus entangling the sensing elements themselves.

#### 1. Heisenberg scaling and the shot noise limit

Let us first review how entangled probe states are most commonly described as a resource [24, 25], and how they can be used to beat the simplest kind of SQL. Consider the argument we gave for position estimation of a mirror in an interferometer we gave in Eq. (4), which is sometimes called an SQL but more properly should be called a shot noise limit. This argument was based on sending $N$ uncorrelated photons into the interferometer of Fig. 1, i.e., producing the state

$$|\psi_{\text{in}}\rangle = \left( \frac{|0\rangle + |1\rangle}{\sqrt{2}} \right) \otimes \left( \frac{|0\rangle + |1\rangle}{\sqrt{2}} \right) \otimes \cdots \qquad (189)$$

after the initial beamsplitter. This represents $N$ unentangled photons. With this state, the shot noise limit worked out to an error $\Delta x_{\text{SN}} = \lambda/\sqrt{N}$.

Now, suppose instead that we have a very different kind of beamsplitter,[9] which instead prepares the state

$$|\psi_{\text{in}}\rangle = \frac{|000\cdots\rangle + |111\cdots\rangle}{\sqrt{2}}. \qquad (190)$$

We can repeat the analysis of Sec. II A. As this state traverses the interferometer it picks up a relative phase on the $|111\cdots\rangle$ term,

$$|\psi_{\text{out}}\rangle = \frac{|000\cdots\rangle + e^{iN\phi}|111\cdots\rangle}{\sqrt{2}}. \qquad (191)$$

We can then make a measurement to determine the probability that this state, after acting with the final beamsplitter (which inverts the original, "magic" beamsplitter), returns to its initial state, in analogy with Eq. (1). This measurement can similarly be used to infer the value of $\phi = x/\lambda$, and one finds instead that the error is

$$\Delta x = \frac{\lambda}{N}, \qquad (192)$$

which is parametrically superior to the naive shot noise limit (4), where error scales like $1/\sqrt{N}$.

Surpassing $1/\sqrt{N}$ error scaling would have many benefits. It also provably requires quantum resources. One does not need states which are exotic as Schrödinger's cat-like state in (190). For example, Caves has shown that sending in a squeezed state with $N$ average photons gives $\delta x \sim 1/N^{3/4}$ [2]. Nevertheless, these protocols do not offer a clear advantage compared to the time-dependent measurement SQL (7) used in this review; in that language, they only affect a better measurement at a single fixed time.

--------

[9] Dowling referred to this as a "magic beamsplitter" [147–149]. Some suggestions for constructing a gate like this include [150, 151] in optics and [152] in microwaves. In qubit systems, this same state is more commonly called the GHZ state, especially in the context of three qubits [153, 154].

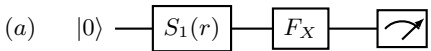

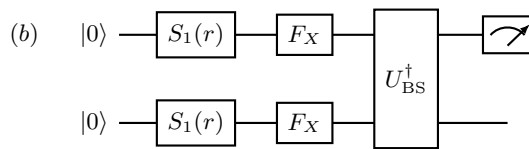

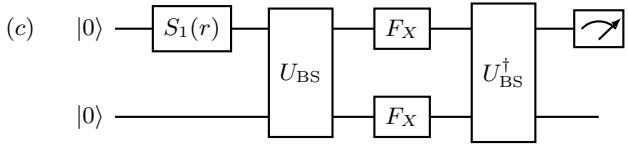

FIG. 18. **Sensing with squeezing and/or entangling operations:** Input vacuum is sent through a sequence of single-mode squeezing and/or beamsplitting operations, interacts with one or more cavities, and is then recombined and read out. (a): One cavity, one squeezing operation. (b) Two cavities, each input squeezed, no entanglement between cavities. (c) Two cavities, one input squeezed and then sent through a beamsplitter; this produces an entangled input state to two cavities, which is then recombined and read out.

### 2. Distributed quantum sensing

In the previous example, we looked at the use of entangled states of the input fields being sent into a single sensing device. In this section, we turn to an alternative idea, involving sending entangled input states into *multiple* devices, sometimes referred to as "distributed quantum sensing" [155–158].

To illustrate the basic idea, we will consider a very simple toy model: two cavities used to sense a "force" $F_X$ in one quadrature. We will assume that this force is identical on both cavities; for example, this could occur with two nearby cavities of size $L$ probing a dark matter signal with wavelength $\lambda \gg L$. We will also assume that the cavity is lossless $\kappa_m \gg \kappa_\ell$ (see Sec. III). Note that these are both major simplifications compared to the realistic wave-like dark matter searches considered in Sec. (III C), in which the cavity loss plays a crucial role ($\kappa_m \approx \kappa_\ell$) and in which the force is distributed across both quadratures with a random phase, not a single deterministic quadrature.

Consider the first protocol (a) in Fig. 18. The output amplitude $X^{\mathrm{out}}$ can be solved in the frequency domain to give (see Sec. III)

$$X^{\mathrm{out}} = \chi_{mm}X_m^{\mathrm{in}} + \chi_{mF}F_X^{\mathrm{in}}, \tag{193}$$

where $F_X^{\mathrm{in}} = F_X^{\mathrm{sig}} = F_0\cos(\omega_0 t)$ is a weak, monochromatic, classical force that displaces the amplitude quadrature of our electromagnetic field. We will assume this force is purely deterministic with no noise, so our

ability to resolve it above the noise from the measurement port $X^{\mathrm{in}}$ is given by the simple expression Eq. (28). Using the same methods of Sec. III, we form the estimator

$$F_E = \frac{X^{\mathrm{out}}}{\chi_{mF}} = F^{\mathrm{sig}} + \frac{\chi_{mm}}{\chi_{mF}}X^{\mathrm{in}}, \tag{194}$$

from which we can compute the force PSD

$$S_{FF} := S_{F_X F_X} = \frac{|\chi_{mm}|^2}{|\chi_{mF}|^2}S_{XX}^{\mathrm{in}} = S_{FF}^{1,\mathrm{vac}}e^{-2r}, \tag{195}$$

where $S_{FF}^{1,\mathrm{vac}} = 1/2$ is the PSD of a single cavity with pure vacuum input, and we assume that we are squeezing the input $X$ quadrature [Eq. (162)]. Thus, the signal-to-noise ratio is

$$\mathrm{SNR}_{(a)} = \frac{F_0}{\sqrt{e^{-2r}S_{FF}^{1,\mathrm{vac}}(\omega_0)/T_{\mathrm{int}}}}. \tag{196}$$

We see that the force can be read out with arbitrarily high SNR, in the limit of large squeezing. Of course, in a real cavity, this effect is cut off by a loss port, which adds a constant ($r$-independent) term to the noise PSD in the denominator.

Now consider the second protocol (b) of Fig. 18. Here the beamsplitter acts to recombine the output fields, so that we can measure the joint output quadrature

$$Q^{\mathrm{out}} = \frac{X_1^{\mathrm{out}} + X_2^{\mathrm{out}}}{\sqrt{2}}. \tag{197}$$

The estimator we should use for the force thus becomes

$$F_E = \frac{\sqrt{2}}{2}Q^{\mathrm{out}} = F^{\mathrm{sig}} + \frac{1}{2}\frac{\chi_{mm}}{\chi_{mF}}\left[X_{m,1}^{\mathrm{in}} + X_{m,2}^{\mathrm{in}}\right]. \tag{198}$$

The key point is that the two input noise terms are random and uncorrelated, and therefore add noise in quadrature. Mathematically, this can be seen in the PSD resulting from this $F_E$, which is

$$S_{FF} = \frac{1}{4}\frac{|\chi_{mm}|^2}{|\chi_{mF}|^2}\left[S_{FF,1} + S_{FF,2}\right] = \frac{1}{2}S_{FF}^{1,\mathrm{vac}} \tag{199}$$

where we used $\langle X_{m,1}^{\mathrm{in}}X_{m,2}^{\mathrm{in},\dagger}\rangle = 0$. We see that the noise PSD is reduced (with respect to the signal) by a factor of 2, and so the SNR for this protocol is

$$\mathrm{SNR}_{(b)} = \frac{F_0}{\sqrt{e^{-2r}S_{FF}^{1,\mathrm{vac}}(\omega_0)/T_{\mathrm{int}}}} = \sqrt{2}\mathrm{SNR}_{(a)}. \tag{200}$$

In general, with $N$ detectors exposed to a coherent force, one can achieve $\mathrm{SNR} = \sqrt{N}\mathrm{SNR}_{(a)}$, which arises solely from classical correlations. At this point we have still not used entanglement to do anything.

Finally, to see how entanglement can be useful, consider the third protcol (c) of Fig. 18. Here, a single-mode squeezed state is split on a balanced beamsplitter. A squeezed vacuum interfering with a vacuum state

produces an entangled state [159–161], so here we have an entangled input field incident on the pair of cavities. Concretely, we have the input fields $\tilde{X}$ incident on the cavities, where

$$\tilde{X}^{\text{in}}_{m,1} = \left[S_1 X^{\text{in}}_{m,1} - X^{\text{in}}_{m,2}\right]/\sqrt{2}$$
$$\tilde{X}^{\text{in}}_{m,2} = \left[S_1 X^{\text{in}}_{m,1} + X^{\text{in}}_{m,2}\right]/\sqrt{2}. \tag{201}$$

Again measuring the output $Q^{\text{out}}$ given in Eq. (197), we find now that

$$Q^{\text{out}} = \frac{1}{\sqrt{2}}\left[\frac{2}{\sqrt{2}}\chi_{mm}S_1 X^{\text{in}}_{m,1} + 2\chi_{mF}F^{\text{in}}\right]. \tag{202}$$

Notice that any dependence on $X_2$ has been removed by our beamsplitting operations. Our estimator for the force is thus

$$F_E = F^{\text{sig}} + \frac{1}{\sqrt{2}}\frac{\chi_{mm}}{\chi_{mF}}X^{\text{in}}_{m,1}, \tag{203}$$

which gives the PSD

$$S_{FF} = \frac{1}{2}S^{1,\text{vac}}_{FF}e^{-2r} \implies \text{SNR}_{(c)} = \text{SNR}_{(b)}. \tag{204}$$

Note that both the PSD and SNR are identical to the result from protocol (b), given in Eq. (200)!

The conclusion here would naively be that there is no advantage in the entangling protocol (c) compared to (b), but this is not quite correct. It is true that one can achieve the same noise PSDs using either (b) or (c). However, there are "technical" advantages to protocol (c). One is that creating the squeezed states is not free—one has to have a non-linear element and a pumping laser, so reducing the number of squeezers from 2 (or $N$, with $N$ detectors) to 1 can be a substantial advantage. Another related advantage is that the power incident on the detectors is lower in (c) than in (b). The number of photons in the squeezed vacuum is not zero but rather $\langle n \rangle = \sinh^2(2r) \approx e^{2r}$. For realistic detectors, particularly mechanical systems, where too much incident power can cause substantial problems (e.g., melting the device surface), this can be a substantial advantage.

## VI. OUTLOOK

The search for new physics, which necessarily involves observation in previously unreachable regimes, has always led to a never-ending need for more precision in measurement. In the past decade, the fundamental quantum mechanical noise of real systems—even those in seemingly quiet states like the vacuum—has come to play an increasing role in the physics of these ultra-sensitive detectors.

Since all devices operate under the laws of quantum mechanics, it thus seems inevitable that quantum effects in measurement will continue to play an increasing role in fundamental physics. We have attempted here to lay out some of the basic tenets of quantum measurement relevant to current or near-future devices, and hope that this will prove a useful tool in the quest for subtle hints of physics beyond our current understanding.

## ACKNOWLEDGEMENTS

We gratefully acknowledge discussions with Rana Adhikari, Tehya Andersen, Ben Brubaker, Aaron Chou, Aashish Clerk, Animesh Datta, James Gardner, Sohitri Ghosh, Evan Hall, Konrad Lehnert, Zhen Liu, Lee McCuller, Klaus Mølmer, Elizabeth Ruddy, Alp Sipahigil, Jacob Taylor, Mankei Tsang, and Xueyue (Sherry) Zhang. We particularly thank Roni Harnik for suggesting that we write an overview like this, and apologize to him for producing it more than a year after the Snowmass process ended.

We thank the Kavli Institute for Theoretical Physics (supported in part by the National Science Foundation under Grants No. NSF PHY-1748958 and PHY-2309135) for hospitality while part of this work was completed. Our work at LBL is supported by the U.S. DOE, Office of High Energy Physics, under Contract No. DEAC02-05CH11231, the Quantum Information Science Enabled Discovery (QuantISED) for High Energy Physics grant KA2401032, the Quantum Horizons: QIS Research and Innovation for Nuclear Science Award DE-SC0023672, and an Office of Science Graduate Student Research (SCGSR) fellowship (administered by the Oak Ridge Institute for Science and Education for the DOE under contract number DE-SC0014664).

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

## Appendix A: Conventions

In this review, we always use natural units $\hbar = c = 1$, although we leave factors of $k_{\mathrm{B}}$ explicit in order to reduce confusion between $T$'s used for times and $T$'s used for temperatures. We quote numerical values in a mix of SI and natural units without apology, choosing instead to use the unit most common in the relevant field. In field theory expressions, single-particle momentum eigenstates are normalized as $\langle \mathbf{p} | \mathbf{p}' \rangle = \delta^3(\mathbf{p} - \mathbf{p}')$, to facilitate comparison with non-relativistic answers. Our metric conventions are those of civilized people: $(-+++)$. We denote angular frequencies by Greek variables $\omega, \nu$ and linear frequency by Latin variables $f = \omega/2\pi$.

In the rest of this appendix, we spell out our conventions for Fourier transforms and power spectral densities. We also record the PSDs for input fields in common states (vacuum, thermal, and squeezed).

### 1. Fourier transforms

In this paper, we take the convention of defining the Fourier transform of an operator $\mathcal{O}(t)$ as

$$\mathcal{O}(\nu) \equiv \mathcal{F}\{\mathcal{O}(t)\} = \int_{-\infty}^{\infty} dt \, e^{i\nu t} \mathcal{O}(t) \qquad (A1)$$

and its inverse is

$$\mathcal{O}(t) \equiv \mathcal{F}^{-1}\{\mathcal{O}(\nu)\} = \int_{-\infty}^{\infty} \frac{d\nu}{2\pi} e^{-i\nu t} \mathcal{O}(\nu). \qquad (A2)$$

With this notation, there is an ambiguity regarding the interplay of the Fourier transform and conjugation. We define $\mathcal{O}^{\dagger}(\nu)$ to be the Hermitian conjugate of the Fourier transform of $\mathcal{O}(t)$ (as opposed to the Fourier transform of the Hermitian conjugate). That is,

$$\mathcal{O}^{\dagger}(\nu) \equiv [\mathcal{F}\{\mathcal{O}(t)\}]^{\dagger}, \qquad (A3)$$

$$\mathcal{O}^{\dagger}(t) \equiv [\mathcal{F}^{-1}\{\mathcal{O}(\nu)\}]^{\dagger}. \qquad (A4)$$

Note that with this convention, we have

$$\mathcal{F}\{\mathcal{O}^{\dagger}(t)\} = \mathcal{O}^{\dagger}(-\nu), \qquad (A5)$$

and

$$\mathcal{F}^{-1}\{\mathcal{O}^{\dagger}(\nu)\} = \mathcal{O}^{\dagger}(-t). \qquad (A6)$$

A common example is the Fourier transform of the position quadrature

$$X(t) = \frac{a(t) + a^{\dagger}(t)}{\sqrt{2}}, \qquad (A7)$$

which becomes

$$X(\nu) = \frac{a(\nu) + a^{\dagger}(-\nu)}{\sqrt{2}}. \qquad (A8)$$

### 2. Quantum power spectral densities

The double-sided power spectral density of two operators $\mathcal{O}_1(t), \mathcal{O}_2(t)$ is defined as

$$S_{\mathcal{O}_1 \mathcal{O}_2}(\nu) = \lim_{T \to \infty} \frac{1}{T} \langle \mathcal{O}_{1,T}(\nu) \mathcal{O}_{2,T}^{\dagger}(\nu) \rangle, \qquad (A9)$$

where $\mathcal{O}_{i,T}(\nu)$ is the windowed Fourier transform defined as

$$\mathcal{O}_{i,T}(\nu) = \int_{-T/2}^{T/2} \mathcal{O}_i(t) e^{-i\nu t} dt, \qquad (A10)$$

and again note that in our conventions, we have

$$\mathcal{O}_{i,T}^{\dagger}(\nu) = \int_{-T/2}^{T/2} \mathcal{O}^{\dagger}(t) e^{i\nu t} dt. \qquad (A11)$$

We will see that the autocorrelation functions defined via

$$C(t, t') = \langle \mathcal{O}(t) \mathcal{O}(t') \rangle, \qquad (A12)$$

are closely related to the power spectral density.

We will often restrict our attention to *stationary* random processes, whose correlation functions are time-translation invariant, meaning that the correlator depends only on the difference between the two times:

$$C(t, t') \equiv C(t - t'). \qquad (A13)$$

The *Wiener-Khinchin theorem* provides a formal connection between the two-point correlator of a stationary process and its PSD. Specifically, the theorem states that for a stationary random process the PSD is the Fourier transform of the two-point correlator,

$$S_{\mathcal{O}_1 \mathcal{O}_2}(\nu) = \int_{-\infty}^{\infty} dt e^{-i\nu t} \langle \mathcal{O}_1(t) \mathcal{O}_2(0) \rangle. \qquad (A14)$$

In particular, this implies a very useful relationship:

$$2\pi \, S_{\mathcal{O}_1 \mathcal{O}_2}(\nu) \, \delta(\nu - \nu') = \langle \mathcal{O}_1(\nu) \mathcal{O}_2^{\dagger}(\nu') \rangle. \qquad (A15)$$

We will use this equation repeatedly throughout this paper, for example, to take expressions for output field $\mathcal{O}^{\mathrm{out}}(\nu)$ and use them to compute output noise power spectra.

The variance in a filtered observable is given by

$$\mathrm{Var}\, \mathcal{O}(t) = \frac{1}{2} \int dt_1 \, dt_2 \, f(t - t_1) f(t - t_2)$$
$$\cdot \left( \langle \{O(t_1), O(t_2)\} \rangle + \langle [O(t_1), O(t_2)] \rangle \right) \qquad (A16)$$

Looking at the piece involving the commutator, we find

$$\mathrm{Var}_{\mathrm{AS}}\, \mathcal{O}(t) \equiv \frac{1}{2} \int dt_1 \, dt_2 f(t - t_1) f(t - t_2) \langle [O(t_1), O(t_2)] \rangle$$
$$= -\frac{1}{2} \int dt_1 \, dt_2 f(t - t_1) f(t - t_2) \langle [O(t_2), O(t_1)] \rangle$$
$$= -\mathrm{Var}_{\mathrm{AS}}\, \mathcal{O}(t), \qquad (A17)$$

where in the second line we have used the antisymmetry of the commutator, and the third line follows since the rest of the integrand is symmetric under exchange of $t_1$ and $t_2$. This shows that the contribution of the antisymmetric part of the correlator to the variance vanishes.

### a. Vacuum

In this section, we calculate the PSDs for the quadrature operators $X^{\text{in}} = (a^{\text{in}} + a^{\text{in}\dagger})/\sqrt{2}$ and $Y^{\text{in}} = -i(a^{\text{in}} - a^{\text{in}\dagger})/\sqrt{2}$. We start with the discretized expression

$$a^{\text{in}}(t) = \frac{1}{\sqrt{2\pi\rho}} \sum_k e^{-i\omega_k(t-t_0)} a_k(t_0), \tag{A18}$$

where $\rho$ is the density of states and we assume the modes are in a vacuum state. Given the above definition and the fact that $[a_k(t_0), a_{k'}^\dagger(t_0)] = \delta_{kk'}$, the commutators satisfy

$$[a^{\text{in}}(t), a^{\text{in}\dagger}(t')] = \frac{1}{2\pi\rho} \sum_k e^{-i\omega_k(t-t')} \to \delta(t-t'), \tag{A19}$$

the last arrow indicating the continuum limit.

Our goal is now to calculate the noise PSD

$$S_{XX}^{\text{in}}(\nu) = \int dt\, e^{i\nu t} \langle x^{\text{in}}(t) x^{\text{in}}(0) \rangle. \tag{A20}$$

In the vacuum, which satisfies $a^{\text{in}}|0\rangle$, we have

$$\langle x^{\text{in}}(t) x^{\text{in}}(0) \rangle = \frac{1}{2} \langle \delta(t) \rangle, \tag{A21}$$

and so the PSD is simply

$$S_{XX}^{\text{in}} \equiv \frac{1}{2}, \tag{A22}$$

independent of $\nu$. The same calculation produces $S_{YY}^{\text{in}} = 1/2$ in vacuum.

### b. Thermal states

We now wish to evaluate PSDs in the case that the bath is in the thermal state. From the previous calculation, it should be clear that we can just work with a single oscillator.

The thermal state is

$$\rho = Z^{-1} \sum_n e^{-\beta E_n} |n\rangle\langle n|, \tag{A23}$$

where $Z$ is the partition function, $\beta = 1/k_{\text{B}}T$ is the inverse temperature, $E_n = n\omega_0$ is the energy of the $n$th state (up to a constant), and $|n\rangle$ is the $n$th Fock state. In general, the thermal expectation value of an operator $\mathcal{O}$ is given by

$$\langle \mathcal{O} \rangle_{\text{th}} = \text{Tr}[\rho\mathcal{O}]. \tag{A24}$$

We are interested in the quadratic correlators of the $X$ and $Y$ quadratures. First, note that

$$\langle a\,a \rangle_{\text{th}} = \langle a^\dagger a^\dagger \rangle_{\text{th}} = 0. \tag{A25}$$

The non-zero expectation values are

$$\begin{aligned} \langle a^\dagger a \rangle_{\text{th}} &= \frac{1}{Z} \sum_n n e^{-\beta E_n} \\ &= -\frac{1}{Z} \frac{\partial}{\partial \beta\omega_0} \sum_n \left(e^{-\beta\omega_0}\right)^n \\ &= \frac{1}{e^{\beta\omega_0} - 1}. \end{aligned} \tag{A26}$$

Using this result, and noting that $aa^\dagger = [a, a^\dagger] + a^\dagger a$, we have

$$\langle aa^\dagger \rangle_{\text{th}} = \frac{1}{1 - e^{-\beta\omega_0}}. \tag{A27}$$

Finally, we want the two-point correlators of the $X$ and $Y$ quadratures. The cross-correlations are

$$\langle XY \rangle_{\text{th}} = \frac{1}{2} \langle aa^\dagger + a^\dagger a \rangle_{\text{th}} = - \langle YX \rangle_{\text{th}}, \tag{A28}$$

from which we see that the symmetrized $XY$ correlators vanish. Thus obtain

$$\langle XX \rangle_{\text{th}} = \langle YY \rangle_{\text{th}} = \frac{1}{2} \coth \frac{\beta\omega_0}{2}, \tag{A29}$$

as reported in Eq. (61).

### c. Single-mode squeezed vacuum

To compute the quadrature PSDs for SMS states, we can start by looking at how the quadrature operators evolve under the SMS interaction Hamiltonian given in Eq. (164)

$$H_I = i\chi^{(2)}(\beta^* a^2 - \beta a^{\dagger 2}), \tag{A30}$$

where we recall that $\beta = |\beta|e^{i\theta}$ describes the laser pump that one controls in the lab. As we will see in detail, the strength of the pump, $|\beta|$, will be used to control the squeezing strength, and the phase $\theta$ will be used to control which quadrature is squeezed. We now make the time dependence explicit and solve the equation of motion for the annihilation operator in the interaction picture

$$\dot{a}(t) = i[H_I, a(t)], \tag{A31}$$

$$= -2\eta a^\dagger(t), \tag{A32}$$

where we have defined $\eta \equiv \chi^{(2)}\beta$. Taking the conjugate of this expression gives $\dot{a}^\dagger(t) = -2\eta^* a(t)$. Together these imply

$$\ddot{a}(t) = 4|\eta|^2 a(t). \tag{A33}$$

Defining $s \equiv 2|\eta|$, we can write this differential equation as

$$\ddot{a}(t) + (is)^2 a(t) = 0, \qquad (A34)$$

which is the simple harmonic oscillator differential equation with solution

$$a(t) = A \cos{(ist)} + B \sin{(ist)}, \qquad (A35)$$

where we note $\cos{ist} = \cosh{st}$ and $\sin{ist} = i \sinh{st}$. The initial conditions imply

$$A = a(0), \qquad (A36)$$

$$B = -i \frac{\beta}{|\beta|} a^\dagger(0) = i e^{i\theta} a^\dagger(0). \qquad (A37)$$

Noting that the squeezing parameter is given as $r = st = 2\chi^{(2)}|\beta|t$, we have

$$a(t) = a(0) \cosh r - a^\dagger(0) e^{i\theta} \sinh r, \qquad (A38)$$

and dropping the time dependence, we have

$$S_1^\dagger(r,\theta) a S_1(r,\theta) = a \cosh r - a^\dagger e^{i\theta} \sinh r. \qquad (A39)$$

By linearity, it follows that the quadratures transform as

$$S_1^\dagger(r,\theta) X S_1(r,\theta) = X(\cosh r - \sinh r \cos\theta) \qquad (A40)$$
$$- Y \sinh r \sin\theta, \qquad (A41)$$
$$S_1^\dagger(r,\theta) Y S_1(r,\theta) = Y(\cosh r + \sinh r \cos\theta) \qquad (A42)$$
$$- X \sinh r \sin\theta, \qquad (A43)$$

For example, amplitude squeezing occurs when $\theta = 0$, so the effect of the squeezing operator on the quadratures is simply the map $X \to X e^{-r}$, while the phase quadrature becomes $Y \to Y e^r$. Generally, a squeezed state exhibits reduced variance along a quadrature specified by the squeezing angle $\theta$ (with increased variance along the orthogonal quadrature).

Using the above results, we can easily compute the noise PSDs with squeezed input fields. For example,

$$2\pi S_{XX}^{SMS}(\nu)\, \delta(\nu - \nu') = \langle\xi| X(\nu) X(\nu') |\xi\rangle, \qquad (A44)$$

by inserting the identity in the form $\mathbb{I} = S_1(\xi) S_1(\xi)^\dagger$, and substituting Eq. (A40). This yields

$$S_{XX}^{SMS}(\nu) = (\cosh r - \sinh r \cos\theta)^2 S_{XX}(\nu) \qquad (A45)$$
$$+ (\sinh r \sin\theta)^2 S_{YY}(\nu), \qquad (A46)$$

where we have used $S_{XY}(\nu) = S_{YX}(\nu) = 0$ in the initial (vacuum) state. The remaining PSDs can be computed in a similar fashion. The result is reported in Eq. (162).

## Appendix B: Detailed dielectric slab calculations

In this appendix, we give a complete and detailed treatment of the dielectric sensor described in Sec. II B. We

chose the slab geometry because its planar symmetry allows for the use of simple plane wave states. Our treatment here is novel, but we have drawn on beautiful related results in the much harder problem where the slab is replaced by a sphere [30, 31]. Our general derivation of the input-output takes much inspiration from the classic reference [17].

Consider a planar slab, of total mass $m$, pendulum frequency $\omega_m$, dielectric polarizability $\chi_e$, and width $\ell$. See Fig. 2. The Hamiltonian for the center-of-mass $x$ of the slab is a simple harmonic oscillator

$$H_{\text{slab}} = \frac{p^2}{2m} + \frac{1}{2} m \omega_m^2 x^2, \qquad (B1)$$

where we are assuming that the motion in the $y, z$ axes is negligible. The detailed nature of the trapping potential is not important; in practice it could be a literal suspension system as in Fig. 2, or an optical tweezing field [32], or a number of other variations. Assuming that the slab is a linear, homogeneous dielectric, it responds to the electric field by polarizing $\mathbf{P}(\mathbf{r}) = \chi_e \mathbf{E}(\mathbf{r})$. The potential describing this interaction is

$$V_{\text{int}}(x,t) = -\frac{1}{2} \chi_e \int_{\text{slab}} d^3\mathbf{r}\, |\mathbf{E}(\mathbf{r},t)|^2, \qquad (B2)$$

where the dependence on the center of mass, $x$, will enter through the limits of integration. In experiments, both the back-scattered and forward-scattered light may contain useful information, thus we would like to be able to separately monitor these modes. To enable this, we decompose the electric field into left- and right-moving modes as

$$E(\mathbf{r}) = E_R(\mathbf{r}) + E_L(\mathbf{r}),$$
$$E_R(\mathbf{r}) = \frac{i}{(2\pi)^3} \int d^3\mathbf{k} \sqrt{\omega_{\mathbf{k}}} \left[ e^{-i(kr_x + \mathbf{k}_\perp \cdot \mathbf{r}_\perp)} a_{k,\mathbf{k}_\perp} - \text{h.c.} \right],$$
$$E_L(\mathbf{r}) = \frac{i}{(2\pi)^3} \int d^3\mathbf{k} \sqrt{\omega_{\mathbf{k}}} \left[ e^{i(kr_x - \mathbf{k}_\perp \cdot \mathbf{r}_\perp)} b_{k,\mathbf{k}_\perp} - \text{h.c.} \right].$$
$$(B3)$$

where $\omega_{\mathbf{k}} = \sqrt{k^2 + \mathbf{k}_\perp^2}$, $k$ ($\mathbf{k}_\perp$) are parallel (perpendicular) to the slab's normal vector, and the creation and annihilation operators satisfy $\left[ a_{\mathbf{k}}, a_{\mathbf{k}'}^\dagger \right] = \delta\left( \mathbf{k} - \mathbf{k}' \right)$. Finally, note that we ignore the polarization of light because we will expand around a strong background field that we take to be linearly polarized.

Next we determine how these modes interact with the slab. For small displacements ($x \ll \lambda$), we may expand Eq. (B2) as

$$V(x) = V(0) + x\partial_x V(0) + O(x^2). \qquad (B4)$$

The non-trivial interaction between the slab position $x$ and the light is encoded in the term proportional to $\partial_x V(0)$. In the integral (B2), the slab position $x$ appears

only in the boundary conditions, and we can evaluate

$$x\partial_x V(0)$$
$$= -\frac{\chi_{\mathrm{e}} x}{2} \int d^2\mathbf{x}_\perp |E(\ell/2, \mathbf{x}_\perp)|^2 - |E(-\ell/2, \mathbf{x}_\perp)|^2. \tag{B5}$$

The next step is to linearize this interaction around a strong background drive that we assume is provided by a laser to the left of the slab. The right-moving mode, then, may be decomposed as a coherent beam plus vacuum fluctuations resulting in terms in the interaction potential of the form

$$V = V_{RR} + 2V_{RL} + V_{LL}, \tag{B6}$$

where the R(L) subscripts indicate the right (left) direction of propagation. The expansions are rather messy, but we will simplify shortly with several approximations. Concretely, the RL cross term (evaluated at $+\ell/2$) is given as

$$V_{RL} \supset -\frac{\chi_{\mathrm{e}}}{2} x \int d^2\mathbf{x}_\perp E_R(\ell/2, \mathbf{x}_\perp) E_L(\ell/2, \mathbf{x}_\perp)$$

$$= \chi_{\mathrm{e}} \frac{x}{2(2\pi)^6} \int d^2\mathbf{x}_\perp \int d^2\mathbf{k}'_\perp d^2\mathbf{k}_\perp \int_0^\infty dk dk' \sqrt{\omega_\mathbf{k}} \sqrt{\omega_{\mathbf{k}'}}$$

$$\times \left[ e^{-i(k\ell/2+\mathbf{k}_\perp \cdot \mathbf{x}_\perp)} a_{k,\mathbf{k}_\perp} - e^{i(k\ell/2+\mathbf{k}_\perp \cdot \mathbf{x}_\perp)} a^\dagger_{k,\mathbf{k}_\perp} \right]$$

$$\times \left[ e^{i(k'\ell/2-\mathbf{k}'_\perp \cdot \mathbf{x}_\perp)} b_{k',\mathbf{k}'_\perp} - e^{-i(k'\ell/2-\mathbf{k}'_\perp \cdot \mathbf{x}_\perp)} b^\dagger_{k',\mathbf{k}'_\perp} \right]$$

$$= \frac{\chi_{\mathrm{e}} x}{2(2\pi)^4} \int d^2\mathbf{k}_\perp \int_0^\infty dk dk' \sqrt{\omega_{k\mathbf{k}_\perp}} \sqrt{\omega_{k',-\mathbf{k}_\perp}}$$

$$\times \left[ e^{-i(k-k')\ell/2} a_{k,\mathbf{k}_\perp} b_{k',-\mathbf{k}_\perp} - e^{-i(k+k')\ell/2} a_{k,\mathbf{k}_\perp} b^\dagger_{k',\mathbf{k}_\perp} \right.$$

$$\left. -e^{i(k+k')\ell/2} a^\dagger_{k,\mathbf{k}_\perp} b_{k',\mathbf{k}_\perp} + e^{i(k-k')\ell/2} a^\dagger_{k,\mathbf{k}_\perp} b^\dagger_{k',-\mathbf{k}_\perp} \right], \tag{B7}$$

where in the final equality we have used the approximation that the slab is infinitely large which is justified when the beam spot is much smaller than the extent of the slab. The full expression is then given as

$$V_{RL} = \frac{\chi_{\mathrm{e}}}{2(2\pi)^4} \int d^2\mathbf{k}_\perp \int_0^\infty dk dk' \sqrt{\omega_{k\mathbf{k}_\perp}} \sqrt{\omega_{k',-\mathbf{k}_\perp}} \tag{B8}$$

$$\left[ e^{-i(k-k')\ell/2} a_{k,\mathbf{k}_\perp} b_{k',-\mathbf{k}_\perp} - e^{-i(k+k')\ell/2} a_{k,\mathbf{k}_\perp} b^\dagger_{k',\mathbf{k}_\perp} \right.$$

$$\left. - e^{i(k+k')\ell/2} a^\dagger_{k,\mathbf{k}_\perp} b_{k',\mathbf{k}_\perp} + e^{i(k-k')\ell/2} a^\dagger_{k,\mathbf{k}_\perp} b^\dagger_{k',-\mathbf{k}_\perp} \right]$$

$$- (l \to (-l)).$$

To simplify this further, we make the rotating wave approximation. In the Schrödinger picture, we have

$$V_{RL} = -i \frac{\chi_{\mathrm{e}}}{(2\pi)^4}$$

$$\times \int d^2\mathbf{k}_\perp \int_0^\infty dk dk' \sqrt{\omega_{k\mathbf{k}_\perp}} \sqrt{\omega_{k',-\mathbf{k}_\perp}} \tag{B9}$$

$$\times \sin[(k+k')\ell/2] \left( a_{k,\mathbf{k}_\perp} b^\dagger_{k',\mathbf{k}_\perp} - a^\dagger_{k,\mathbf{k}_\perp} b_{k',\mathbf{k}_\perp} \right),$$

where we have dropped so-called "counter-rotating" terms. That is, if we used this interaction to compute transition matrix elements in time-dependent perturbation theory, the terms of the form $ab$ will come with phases likes $e^{i(k+k')t}$, whereas terms like $a^\dagger b$ will come with $e^{i(k-k')t}$. Integrating over long times, the first type of behavior will average to give a vanishing contribution (recall that $k, k' > 0$) whereas the second type will not always average out. We thus drop the counter-rotating terms $ab, a^\dagger b^\dagger$. Finally, note that the $LR$ contribution is identical (leading to an overall factor of 2) and the $V_{RR}, V_{LL}$ terms can be derived in a similar fashion.

Finally, we introduce a laser drive. In the above the field has been totally general. With a laser, the right-moving modes become a coherent state peaked sharply around a specific mode $k = k_0, \mathbf{k}_\perp = 0$. We can introduce this by displacing the mode

$$a_{k,\mathbf{k}_\perp} \to (2\pi)^3 \alpha_0 e^{i\omega_0 t} \delta(k - k_0) \delta^2(\mathbf{k}_\perp) + a_{k,\mathbf{k}_\perp}. \tag{B10}$$

where now $a$ has the interpretation of a fluctuation around the drive $\alpha_0$. Here, $\alpha_0 = |\alpha_0| e^{i\phi}$ is in general complex (i.e., the laser can have arbitrary initial phase $e^{i\phi}$). We note that $\alpha_0$ can be determined directly from experimental parameters as follows. Given the intensity $I$ and frequency $\omega$ of light, we have $|\alpha_0|^2 = \frac{I}{\omega}$. Then, we have $a_k \equiv \sqrt{\frac{1}{A}} a_{k,\mathbf{k}_\perp=0}$, $b_k \equiv \sqrt{\frac{1}{A}} b_{k,\mathbf{k}_\perp=0}$, in terms of the laser cross-sectional area $A$. This definition results in $a_k, b_k$ having dimensions of $(\mathrm{mass})^{-1/2}$, ensures they obey one-dimensional commutation relations $\left[ a_k, a^\dagger_{k'} \right] = \delta(k - k')$, and implies their free evolution is governed by the Hamiltonian

$$H_{\mathrm{EM}} = \int_0^\infty dk \omega_k \left[ a^\dagger_k a_k + b^\dagger_k b_k \right], \quad \omega_k := \omega_{k,\mathbf{k}_\perp=0}. \tag{B11}$$

Inserting Eq. (B10) back into the interaction considerably simplifies things, and in particularly makes the interaction *linear* in the fluctuations, thus a bilinear coupling between the mechanics $x$ and the light:

$$V_{RL} = x \int_0^\infty dk \left[ g_k e^{i\omega_0 t} b^\dagger_k + g^*_k e^{-i\omega_0 t} b_k \right], \tag{B12}$$

where we also defined the couplings

$$f_k = -i \frac{1}{2\pi} \alpha_0 \chi_{\mathrm{e}} \sqrt{A\omega_k \omega_0} \sin[(k_0 - k)\ell/2] \tag{B13}$$

$$g_k = -i \frac{1}{2\pi} \alpha_0 \chi_{\mathrm{e}} \sqrt{A\omega_k \omega_0} \sin[(k_0 + k)\ell/2]. \tag{B14}$$

We see that the problem has been reduced to one purely along the $x$-axis, which we anticipated by symmetry. We also see that that the coupling between the mirror position $x$ and a given photon is now enhanced by a factor $\alpha_0$ (square-root of the laser power) compared to the value without the laser.

By the same kind of calculation, one can get similar expressions for the $V_{LL}$, and $V_{RR}$. The latter will contain

terms up to quadratic in $\alpha_0$. The quadratic term leads to a shift in the equilibrium position of the slab, and the pure vacuum terms (i.e. those not enhanced by the drive) will be sub-dominant and thus omitted. By similar arguments we neglect the $V_{LL}$ term. The remaining non-negligible term is given as

$$V_{RR} = x \int_0^\infty dk \left[ f_k e^{i\omega_0 t} a_k^\dagger + f_k^* e^{-i\omega_0 t} a_k \right] \quad (B15)$$

The physics of these couplings are simple: the $RL$ terms describe reflection of a mode off the dielectric slab, while the $RR$ terms describe transmission. Finally, we need to handle the $V(0)$ term in Eq. B4

$$V(0) = -\frac{1}{2}\chi_e \int_{\text{slab}} d^3\mathbf{r} |\mathbf{E}(\mathbf{x_0}, t)|^2. \quad (B16)$$

If we denote the coherent laser light as $E_{\text{coh}}$, the terms that will be enhanced by the laser (and thus non-negligible) will be given as

$$V(0) =$$
$$-\frac{1}{2}\chi_e \int_{\text{slab}} d^3r \left[ |E_{\text{coh}}|^2 + 2\mathbf{Re}(E_{\text{coh}} E_R^*) + 2\mathbf{Re}(E_{\text{coh}} E_L^*) \right]. \quad (B17)$$

While the first term is constant and will not affect the dynamics, the second term is given as

$$\chi_e \int d^2 x_\perp dx \mathcal{E}_{k_0} \mathcal{E}_k (\alpha_0 e^{i\omega_0 t} e^{-ikx} - \alpha_0^* e^{-i\omega_0 t} e^{ikx})$$
$$\int dk d^2 k_\perp (e^{-i(kx + k_\perp r_\perp)} a_{k,k_\perp} - e^{-i(kx_0 + k_\perp r_\perp)} a_{k,k_\perp})$$
$$\approx \chi_e \int dk dx_0 \mathcal{E}_{k_0} \mathcal{E}_k \alpha_0 (e^{i\omega_0 t} a_k^\dagger e^{i(k-k_0)x} + \text{h.c.}) \quad (B18)$$

under the rotating wave approximation. Note that the third term can be handled similarly yields an identical form. Integrating over the linear extent of the slab yields

$$V(0) = i \int dk \frac{1}{k - k_0} (f_k e^{i\omega_0 t} a_k^\dagger - f_k^* e^{-i\omega_0 t} a_k) \quad (B19)$$
$$+ i \int dk \frac{1}{k + k_0} (g_k e^{i\omega_0 t} b_k^\dagger - g_k^* e^{-i\omega_0 t} b_k) \quad (B20)$$

We now pause and take stock of what we have thus far. In particular, we have derived an interaction Hamiltonian of the form

$$V(x) = V_{RR}(x) + 2V_{RL}(x) + V(0), \quad (B21)$$

with

$$V_{RR} = x \int_0^\infty dk \left[ f_k e^{i\omega_0 t} a_k^\dagger + f_k^* e^{-i\omega_0 t} a_k \right], \quad (B22)$$

$$V_{RL} = x \int_0^\infty dk \left[ g_k e^{i\omega_0 t} b_k^\dagger + g_k^* e^{-i\omega_0 t} b_k \right], \quad (B23)$$

$$V(0) = i \int dk \frac{1}{k - k_0} (f_k e^{i\omega_0 t} a_k^\dagger - f_k^* e^{-i\omega_0 t} a_k) + \quad (B24)$$

$$i \int dk \frac{1}{k + k_0} (g_k e^{i\omega_0 t} b_k^\dagger - g_k^* e^{-i\omega_0 t} b_k) \quad (B25)$$

At this stage, we have a relatively simple interaction Hamiltonian describing the interaction of the light—specifically, photonic fluctuations around the laser drive—and the center-of-mass coordinate $x$ of the mirror. Physically, we are able to make measurements on the light field, and we want to use these measurements to infer the mechanical position $x$. To see how this works, we need to understand how the quantum state of the mechanics is encoded in the quantum state of the light.

Moving to the Heisenberg picture, the equations of motion for the bath modes are given by $\dot{a}_k = i[H, a_k]$, and so forth. The basic strategy will be to write an exact solution for the bath modes at time $t$ in terms of their initial conditions at some early time $t_0$ and in terms of the position operator $x(t)$ for the mechanics. It is easier to transform these operators to ones which are rotating with the laser frequency, $a_k \to a_k e^{-i\omega_0 t}$, $b_k \to b_k e^{-i\omega_0 t}$, which also shifts $\alpha_0 e^{i\omega_0 t} \to \alpha_0$, i.e., in this frame the laser is constant. In the frame co-rotating with the laser beam, the equations of motion of the light are

$$\dot{a}_k = -i\Delta_k a_k + if_k x - \frac{f_k}{k - k_0}$$
$$\dot{b}_k = -i\Delta_k b_k + ig_k x - \frac{g_k}{k + k_0}. \quad (B26)$$

in terms of the detuning of the mode from the laser $\Delta_k = k - k_0$. The solution of Eq. (B26) can be written as

$$a_k(t) = a_k^{\text{in}}(t)$$
$$+ i \int_{t_0}^t dt' f_k e^{-i\Delta_k(t-t')} \left( x(t') + i\frac{1}{k - k_0} \right), \quad (B27)$$

where $a_k^{\text{in}}(t) = e^{i\Delta_k(t-t_0)} a_k(t_0)$ is written in terms of an initial boundary condition at time $t_0$. Similarly, we can define $a_k^{\text{out}}(t) = e^{-i\Delta_k(t-t_1)} a_k(t_1)$ and write the solution in terms of late-time boundary conditions at $t_1$

$$a_k(t) = a_k^{\text{out}}(t)$$
$$- i \int_t^{t_f} dt' f_k e^{-i\Delta_k(t-t')} \left( x(t') + i\frac{1}{k - k_0} \right), \quad (B28)$$

The difference of these two equations gives

$$a_k^{\text{out}}(t) - a_k^{\text{in}}(t)$$
$$= i \int_{t_0}^{t_f} dt' f_k e^{-i\Delta_k(t-t')} \left( x(t') + i\frac{1}{k - k_0} \right). \quad (B29)$$

This is the basic in-out relation for the optical field monitoring the slab.

$$a_k(t) = e^{-i\Delta_k(t-t_0)} a_k(t_0) + if_k \int_{t_0}^t dt' e^{-i\Delta_k(t-t')} x(t'),$$

$$b_k(t) = e^{-i\Delta_k(t-t_0)} b_k(t_0) + ig_k \int_{t_0}^t dt' e^{-i\Delta_k(t-t')} x(t'). \quad (B30)$$

One can easily verify that these solve the equations of motion for the bath modes. These solutions show how the mode operators at time $t$ encode the entire past history of the position operator $x(t)$. However, these relations are not particularly useful as they are, because the equations of motion for $x(t)$ itself depend on the bath modes through the interaction (B2). The equations of motion for the slab are

$$\dot{x} = p/m \tag{B31}$$

$$\dot{p} = -m\omega_m^2 x - \int_0^\infty dk (f_k a_k^\dagger + f_k^* a_k)$$
$$\quad - \int_0^\infty dk (g_k b_k^\dagger + g_k^* b_k). \tag{B32}$$

Note that we take $f$ and $g$ to be real numbers, which is equivalent to choosing the phase of $\alpha_0$ so that it is imaginary. This choice of phase implies that the effect of changes in position are imprinted on the phase quadrature of the output light. Our task is then to figure out how to solve these coupled differential equations. This is possible since they are linear. We can convert to the phase and amplitude quadratures as

$$X_k(t) \equiv \frac{a_k + a_k^\dagger}{\sqrt{2}} \tag{B33}$$
$$= X_k^{\rm in} + \frac{1}{\sqrt{2}} \int_{t_0}^t dt' \frac{f_k}{k - k_0} (e^{-i\Delta_k(t-t')} + e^{i\Delta_k(t-t')}),$$

$$Y_k(t) \equiv \frac{a_k - a_k^\dagger}{\sqrt{2}i} \tag{B34}$$
$$= Y_k^{\rm in} + \frac{1}{\sqrt{2}} \int_{t_0}^t dt' f_k x(t')(e^{-i\Delta_k(t-t')} + e^{i\Delta_k(t-t')}).$$

We can get particularly simple answers by introducing our second key approximation, sometimes referred to as the Markov approximation. The idea is that when $|k - k_0| \gtrsim 1/\tau$, where $\tau$ is the interaction time, any phases that rapidly oscillate will average to zero. Since the phonons, or excitations of the harmonic potential, oscillate with a certain amplitude and frequency $\omega_m$, the phase will only pick up relevant modes around $k_0 + \omega_m$ and $k_0 - \omega_m$. When the mechanical frequency is much smaller than the laser frequency, we can make an approximation $f_k \equiv f = f_{k_0}$ and $g_k \equiv g = g_{k_0}$. Explicitly, by comparison with (B14), we get

$$g = \frac{|\alpha_0|\chi_{\rm e}\omega_0}{2\pi} \sqrt{A} \sin k\ell. \tag{B35}$$

Note that this is basically the same argument we used to make the RWA above. Then we can integrate over all modes and define $Y^{\rm in}(t) = \int_0^\infty dk Y_k^{\rm in} = \int_0^\infty dk \frac{a_k^{\rm in} - a_k^{\rm in\dagger}}{\sqrt{2}i}$, to get the input-output equations

$$Y_R^{\rm out} - Y_R^{\rm in} = \sqrt{2} f x, \tag{B36}$$
$$Y_L^{\rm out} - Y_L^{\rm in} = \sqrt{2} g x. \tag{B37}$$

These are essentially scattering equations (familiar from, for example, the LSZ reduction formula [162]). What they do is give us the Heisenberg-picture fields of the light at late time in terms of the fields of the light at early time plus the effect of the light scattering off the slab. As anticipated, the the center-of-mass position has been imprinted on the phase of the output light. The amplitude quadratures of the light will instead be shifted coherently, and obey different in-out relations

$$X_R^{\rm out} - X_R^{\rm in} = \sqrt{2\Gamma_R} \tag{B38}$$
$$X_L^{\rm out} - X_L^{\rm in} = \sqrt{2\Gamma_L}, \tag{B39}$$

for both left and right-moving modes. Here we defined $\sqrt{\Gamma_R} = \frac{f_k}{k - k_0}|_{k=k_0} = -i\frac{k_0\ell}{4}\alpha_0\chi_{\rm e}\sqrt{A}$ and $\sqrt{\Gamma_L} = \frac{g_{k_0}}{2k_0} = -i\frac{1}{4}\alpha_0\chi_{\rm e}\sqrt{A}\sin(k_0\ell)$, and used

$$\int_0^\infty dk e^{-i(k-k_0)(t-t')} \approx \int_{-\infty}^\infty dk e^{-i(k-k_0)(t-t')}$$
$$= 2\pi\delta(t-t'). \tag{B40}$$

It remains to solve the equations of motion for the slab. We can plug Eq. (B27) into Eq. (B26), to yield a second-order differential equation for the center-of-mass position

$$m\ddot{x} + m\omega^2 x = -\sqrt{2} f X_R^{\rm in} - \sqrt{2} g X_L^{\rm in} + F^{\rm in} +$$
$$\int_0^\infty dk \left( \frac{|f_k|^2}{k - k_0} + \frac{|g_k|^2}{k + k_0} \right) 4\pi\delta(k - k_0). \tag{B41}$$

Note that we used the fact that $\int_{t_0}^t dt'(e^{i\Delta_k(t-t')} + e^{-i\Delta_k(t-t')}) = 4\pi\delta(k - k_0)$, and defined the mechanical susceptibility $\chi_m^{-1}[\Omega] = m(-\Omega^2 + \omega_m^2)$. In the Fourier domain, these equations take the form

$$Y_L^{\rm out} = Y_L^{\rm in} + \sqrt{2} g \chi_m \big( -\sqrt{2} f X_R^{\rm in} - \sqrt{2} g X_L^{\rm in} +$$
$$F^{\rm in} + F_0 2\pi\delta(\Omega) \big) \tag{B42}$$
$$Y_R^{\rm out} = Y_R^{\rm in} + \sqrt{2} f \chi_m \big( -\sqrt{2} f X_R^{\rm in} - \sqrt{2} g X_L^{\rm in} +$$
$$F^{\rm in} + F_0 2\pi\delta(\Omega) \big). \tag{B43}$$

We note that $Y_R^{\rm out}$ describes the forward-scattered light, $Y_L^{\rm out}$ the back-scattered light, and the $F_0$ (which the last term in Eq. (B41)) can be interpreted as a constant radiation pressure on the slab (as it is constant term and quadratic in the amplitude $|\alpha_0|$).

In this 1D case, $f = 0$ under the Markovian approximation and, which implies that the SNR $= 0$, i.e., the forward scattered light contains no information about the force. We thus focus on the back-scattered light, for which the PSD is

$$S_{YY,L}^{\rm out} = S_{YY,L}^{\rm in} + 2g^2|\chi_m|^2 S_{FF} + 4f^2 g^2|\chi_m|^2 S_{XX,R}^{\rm in}$$
$$+ 4g^4|\chi_m|^2 S_{XX,L}^{\rm in} + 2\mathbf{Re}\left( 2g^2\chi_m S_{XY,L}^{\rm in} \right). \tag{B44}$$

We can give some intuition of the terms: $S_{YY,L}$ is the shot noise, while $S_{XX,L}$ is the back-action noise, proportional to the square of mechanical susceptibility. There

are also correlation terms that may be non-vanishing if one uses squeezed or non-Gaussian states of light. $F_0$ is the radiation pressure force on the slab, which would shift Y by shifting the slab position, but it has no fluctuations and will not contribute to the noise. From this PSD, we may construct an estimator for the force PSD as

$$S_{FF}(\nu) = \frac{S_{YY}^{\text{out}}(\nu)}{|\chi_{YF}(\nu)|^2} \tag{B45}$$

Therefore the noise in the signal can be expressed as such for a back-scattered readout

$$S_{FF,L}^{out} = \frac{S_{YY,L}^{in}}{2g^2|\chi_m|^2} + 2g^2 S_{XX,L} + 2f^2 S_{XX,R} + \frac{2}{|\chi_m|^2}\mathbf{Re}\left(\chi_m S_{XY,L}^{in}\right). \tag{B46}$$

When we set $f = 0$ in the Markov approximation, we obtain

$$\chi_{YX} = 1, \chi_{YY} = -2g^2\chi_m, \chi_{YF} = \sqrt{2}g\chi_m, \tag{B47}$$

for the susceptibilities.

## Appendix C: Homodyne detection

In many of the examples we consider, we wish to measure a particular quadrature of the photon mode of interest. The photon has frequency $\omega_0$ and is described by creation/annihilation operators $a$ and $a^\dagger$. Homodyne detection allows us to measure the *Schrödinger picture operator*

$$X_\phi(t) = \frac{1}{\sqrt{2}}\left(e^{i\omega_0 t - i\phi}a + e^{-i\omega_0 t + i\phi}a^\dagger\right) \tag{C1}$$

for an arbitrary, constant phase $\phi$. Note that under free time evolution, this operator is a constant of motion, since in the Heisenberg picture it satisfies

$$\frac{dX_\phi(t)}{dt} = i\omega_0[a^\dagger a, X_\phi] + \partial_t X_\phi = 0. \tag{C2}$$

One may measure this generalised quadrature as follows [163, 164]. Consider shining the light of mode $a$ on a 50-50 beam splitter, so that the beam output modes are related to the inputs by

$$\begin{pmatrix} c \\ d \end{pmatrix} = \frac{1}{\sqrt{2}}\begin{pmatrix} 1 & i \\ i & 1 \end{pmatrix}\begin{pmatrix} a \\ b \end{pmatrix}. \tag{C3}$$

The output observable we measure is the difference between the number of photons in each output arm

$$\begin{aligned} n_- &= c^\dagger c - d^\dagger d \\ &= i(a^\dagger b - b^\dagger a), \end{aligned} \tag{C4}$$

where in the second line we have expressed $n_-$ in terms of the input mode operators. If the $b$-mode into the beamsplitter is populated by a large amplitude coherent state of frequency $\omega_0$, i.e. $|b\rangle = |\beta e^{i\omega_0 t}\rangle$, then the expectation value of intensity difference is

$$\begin{aligned} \langle n_- \rangle &= \langle a|\langle b| n_- |a\rangle|b\rangle \\ &= \sqrt{2}|\beta|\langle X_\phi(t)\rangle, \end{aligned} \tag{C5}$$

where $\phi = -\arg\beta + \pi/2$, and we have assumed that the light in the two incoming beams is unentangled. This demonstrates that by adjusting the phase of the coherent state $\beta$, one may measure $X_\phi(t)$ for any value of $\phi$ in this way.

## Appendix D: Fluctuation-dissipation theorem

When a stochastic bath, such as a thermal gas or a noisy transmission line, is coupled to a system, it acts as a loss channel as well as a source of noise – the fluctuation-dissipation theorem is a relation between the noise and loss rate associated with this channel [165].

## Appendix E: Detailed optomechanical calculations

In this appendix we present a number of detailed calculations which were suppressed in the main text.

### 1. Driven optomechanical coupling

Here we give a detailed derivation of the optomechanical Hamiltonian under the linearization approximation used in Sec. IV, following [17]. We consider a Fabry-Pérot cavity in which the electromagnetic cavity mode couples to the mechanical modes through the cavity resonance frequency. To see this, note that length of the cavity changes by some amount $x$, the resonant wavelengths become

$$\lambda_n = \frac{2(L-x)}{n}, \tag{E1}$$

from which the harmonic frequencies are

$$\omega_n(x) = \frac{\pi n}{L-x} = \omega_n(0)\left[1 + \frac{x}{L} + \mathcal{O}\left(\left(\frac{x}{L}\right)^2\right)\right], \tag{E2}$$

where $\omega_n(0)$ is the bare resonance frequency of the cavity and we have assumed $x \ll L$. Thus, to first order in $x$, the mechanical motion serves to linearly shift the resonance frequency of our cavity. As in the main text, we now restrict our attention to a cavity with a mode at $\omega_c$ that dominates the opto-mechanical coupling, and mechanics with mass $m$ and resonance frequecy $\omega_m$. The

Hamiltonian, then becomes

$$H_{\text{bare}} = \omega_c(x)a^\dagger a + \omega_m d^\dagger d, \tag{E3}$$

$$= \omega_c(1 + \frac{x}{L})a^\dagger a + \omega_m d^\dagger d, \tag{E4}$$

$$H_{\text{bare}} = \omega_c a^\dagger a + \omega_m d^\dagger d + g_0 x_0(d + d^\dagger)a^\dagger a, \tag{E5}$$

where we have used $x = x_0(d+d^\dagger)$, with $x_0 := \sqrt{1/2m\Omega}$, and defined the bare (or vacuum) optomechanical coupling rate

$$g_0 := \frac{\omega_c}{L}. \tag{E6}$$

This vacuum coupling rate is typically much smaller than the optical and mechanical decoherence rates involved in the problem.

Moving to the frame co-rotating with the laser (at frequency $\omega_L$), we obtain

$$H_{\text{bare}} = \Delta a^\dagger a + \omega_m d^\dagger d + g_0 x_0 a^\dagger a(d^\dagger + d), \tag{E7}$$

where $\Delta \equiv \omega_c - \omega_L$. To enhance the coupling, one typically drives the cavity coherently with a strong laser. Including this driving in the Hamiltonian adds a term

$$H_{\text{driven}} = \Delta a^\dagger a + \omega_m d^\dagger d + g_0 x_0 a^\dagger a(d^\dagger + d) + \mathcal{E}(a + a^\dagger), \tag{E8}$$

where $\mathcal{E}$ is the drive strength. This drive displaces the steady state of both the intracavity field and the mechanical position. We can remove these displacements redefining

$$a \rightarrow \alpha + a, \tag{E9}$$

$$d \rightarrow \delta + d, \tag{E10}$$

where $\alpha \equiv \langle a \rangle$ and $\delta \equiv \langle d \rangle$. We note that $\delta$ is real and $\alpha$ is usually taken to be real, which then establishes the intracavity field as a phase reference for all other fields in the problem. Further, we set

$$\alpha = -\frac{\mathcal{E}}{\Delta + 2\delta g_0 x_0}, \tag{E11}$$

$$\delta = -\frac{g_0^2 x_0^2 \alpha^2}{\omega_m}, \tag{E12}$$

Making these transformations, and discarding any term not proportional to $a$ or $b$ (because they will not contribute to the dynamics), we obtain

$$H_{\text{driven}} = \left(\Delta - \frac{2g_0^2 x_0^2 \alpha^2}{\omega_m}\right)a^\dagger a + \omega_m d^\dagger d + g_0[\alpha(a + a^\dagger) + a^\dagger a](d + d^\dagger). \tag{E13}$$

The detuning is easily controlled experimentally, so we can simply further shift it

$$\Delta \rightarrow \Delta + \frac{2g_0^2 x_0^2 \alpha^2}{\omega_m}, \tag{E14}$$

to arrive at the simpler Hamiltonian

$$H_{\text{driven}} = \Delta a^\dagger a + \omega_m b^\dagger b + g_0 x_0[\alpha(a + a^\dagger) + a^\dagger a](d + d^\dagger). \tag{E15}$$

Now, observe that the second term in square brackets is not enhanced by the coherent amplitude of the laser, thus it is typically negligible. Dropping this term leads to the final, linearized Hamiltonian of the system

$$H_{\text{sys}} = \Delta a^\dagger a + \omega_m d^\dagger d + gx_0(a + a^\dagger)(d + d^\dagger),$$
$$= \frac{p}{2m} + \frac{1}{2}m\omega_m^2 x^2 + \frac{\Delta}{2}a^\dagger a + \sqrt{2}gxX \tag{E16}$$

where we have defined the enhanced opto-mechanical coupling rate as $g := |\alpha|g_0$. Thus, we see that in the linearized regime, our Hamiltonian becomes that of a pair of position-position coupled oscillators, as used in Eq. (112).

## 2. One vs two movable mirrors

In this appendix, we derive the Hamiltonian for a cavity where both ends are treated as movable, harmonic oscillators, and compare this to the case where only one end can move. The upshot is simple: the Hamiltonian for the distance between two movable mirrors is just that for a single mirror of mass $\mu = m/2$, i.e., the reduced mass.

If only one side of the cavity is free to move and the other side is fixed, then the relevant degree of freedom $x$ is the deviation of the mirror's position from its equilibrium. For a mirror of mass $m$, we model this as a simple harmonic oscillator with frequency $\omega$

$$H = \frac{1}{2}m\omega^2 x^2 + \frac{1}{2m}p^2. \tag{E17}$$

If both sides of the cavity are free to move, the relevant degree of freedom is the difference between the positions of the two movable mirrors: $x_1 - x_2 \equiv x_-$. If both of the mirrors have the same mass and same spring constant, then the Hamiltonian of the combined system is

$$H = \frac{m\omega^2}{2}(x_1^2 + x_2^2) + \frac{1}{2m}(p_1^2 + p_2^2) \tag{E18}$$

$$= \frac{1}{2}\mu\omega^2(x_+^2 + x_-^2) + \frac{p^2}{2\mu}(p_+^2 + p_-^2), \tag{E19}$$

where we have carried out a canonical transformation to new variables $x_\pm = (x_1 \pm x_2)$ and $p_\pm = \frac{1}{2}(p_1 \pm p_2)$ such that $[x_-, p_-] = [x_+, p_+] = i$, and we introduced the reduced mass $\mu = m/2$.

How do the bath modes couple to this two-sided cavity? We introduce two sets of bath modes, one for each movable end. Assuming pure position-position couplings (so that the bath merely damps the system), the Hamil-

tonian containing the bath modes is

$$
\begin{aligned}
H_{\text{bath}} &= \sum_p \omega_p B_p^\dagger B_p + \tilde{\omega}_p \tilde{B}_p^\dagger \tilde{B}_p + g_p x_1 X_p + \tilde{g}_P x_2 \tilde{X}_p \\
&= \sum_p \omega_p B_p^\dagger B_p + \tilde{\omega}_p \tilde{B}_p^\dagger \tilde{B}_p \\
&\quad + \frac{1}{2} x_+ (g_p X_p + \tilde{g}_P \tilde{X}_p) \\
&\quad + \frac{1}{2} x_- (g_p X_p - \tilde{g}_P \tilde{X}_p).
\end{aligned}
$$
(E20)

We now assume that the frequency of the bath modes are equal $\omega_p = \tilde{\omega}_p$, and that $g_p = \tilde{g}_p$ so that they couple with the same strength to each movable mirror[10]. The above Hamiltonian becomes

$$
H = \sum_p \omega_p (B_{p+}^\dagger B_{p+} + B_{p-}^\dagger B_{p-}) + \frac{1}{2} g_p x_+ X_{p+} + \frac{1}{2} g_p x_- X_{p-},
$$
(E21)

where $X_{p\pm} \equiv X_p \pm \tilde{X}_p$. Since the sum-mode $x_+$ and its associated bath modes are decoupled from the rest of the system (including the optomechanics), they are of no interest to us, and from now on we may drop them. This fact, along with equation (E19), implies that the Hamiltonian for the two-sided cavity is the same as the one-sided case with the substitutions $x \to x_-$ and $m \to \mu = m/2$.

### 3. Full optomechanical SQL

Here we will give the general expression for the inferred force noise power in an optomechanical device, as a function of the frequency $\omega_*$ at which one tunes the laser power to achieve the SQL. With uncorrelated input noise ($S_{YX}^{\text{in}} = 0$), the quantum noise contribution to the force PSD is [see Eq. (123)]

$$
S_{FF}^{\text{quantum}} := \frac{|\chi_{YY}|^2}{|\chi_{YF}|^2} S_{YY}^{\text{in}} + \frac{|\chi_{YX}|^2}{|\chi_{YF}|^2} S_{XX}^{\text{in}}.
$$
(E22)

Following the discussion in Sec. IV A, fix a reference frequency $\omega_*$, and tune the laser power such that $\partial S_{FF}(\omega_*)/\partial g^2 = 0$. This occurs when we tune the power so that the driven optomechanical coupling is given by

$$
g_*^2 = \frac{1}{2\kappa |\chi_c(\omega_*)|^2 |\chi_m(\omega_*)|},
$$
(E23)

where at this point for simplicity we assumed input vacuum noise $S_{XX}^{\text{in}} = S_{YY}^{\text{in}} = 1/2$. In the main text, we worked out the example where $\omega_* = \omega_m$, the mechanical

───────

[10] The sign of the interaction strength is arbitrary, as we may redefine $X_p \to -X_p$ and leave the free part of the Hamiltonian invariant.

resonance frequency and focused on the behavior of the noise PSD near the resonance. More generally, we can insert this value of $g$ and obtain, using Eqs. (117) and (118),

$$
\begin{aligned}
S_{FF}^{\text{quantum}}(\nu) &= \frac{|\chi_c(\omega_*)|^2 |\chi_m(\omega_*)|}{|\chi_c(\nu)|^2 |\chi_m(\nu)|^2} \\
&\quad + \frac{|\chi_c(\nu)|^2}{|\chi_c(\omega_*)|^2 |\chi_m(\omega_*)|}.
\end{aligned}
$$
(E24)

This expression gives the full frequency dependence of the PSD. As a check, note that if we focus on the frequency $\nu = \omega_*$, this reduces to $S_{FF} = 1/|\chi_m(\omega_*)|$, which in particular matches Eq. (125) if we choose to tune the laser on resonance $\omega_* = \omega_m$. However, Eq. (E24) is much more general. For example, many practical experiments use $\omega_* \gg \omega_m$ (e.g., LIGO uses $\omega_m \sim 1$ Hz and $\omega_* \sim 100$ Hz). This is relevant in the "free particle limit", where observations are done in a regime where the harmonic mechanical motion can be approximated as just free motion. The same limit is useful in impulse sensing, since the signal is broadband and generally supported on much higher frequencies than the mechanical resonance.

More generally, we can work out the analogous result without assuming uncorrelated or vacuum input noise—in particular, this allows us to calculate the spectrum with an input squeezed state. In Eq. (170), we gave the general expression for the quantum contribution to the force noise PSD, and in Eq. (171) we gave the optimal coupling $g_*$ as a function of $\omega_*$. Plugging this back and again using the explicit transfer functions, we obtain the full noise PSD:

$$
\begin{aligned}
S_{FF}^{\text{quantum}}(\nu) &= \sqrt{\frac{S_{XX}^{\text{in}}(\omega_*)}{S_{YY}^{\text{in}}(\omega_*)}} \frac{|\chi_c(\omega_*)|^2 |\chi_m(\omega_*)|}{|\chi_c(\nu)|^2 |\chi_m(\nu)|^2} S_{YY}^{\text{in}}(\nu) \\
&\quad + \sqrt{\frac{S_{YY}^{\text{in}}(\omega_*)}{S_{XX}^{\text{in}}(\omega_*)}} \frac{|\chi_c(\nu)|^2}{|\chi_c(\omega_*)|^2 |\chi_m(\omega_*)|} S_{XX}^{\text{in}}(\nu) \\
&\quad + 2m(\omega_m^2 - \nu^2) S_{YX}^{\text{in}}(\nu).
\end{aligned}
$$
(E25)

Again, one can check that this reduces to Eq. (172) at the frequency where the noise is optimized, $\nu = \omega_*$. It also reduces to (E24) in the limit of input vacuum noise.

### 4. Strain PSD with frequency-dependent squeezing

Here we show how to go from the strain PSD (175) to the simplified expression (177). Writing out the explicit

transfer functions [see Eq. (117)], we have

$$S_{hh}^{SMS}(\omega) = \frac{h_{\text{SQL}}^2}{2} \left( |\chi_{YX}| + \frac{1}{|\chi_{YX}|} \right) \cosh 2r$$
$$+ \frac{h_{\text{SQL}}^2}{2} (-|\chi_{YX}| + \frac{1}{|\chi_{YX}|}) \cos\theta \sinh 2r$$
$$- \frac{h_{\text{SQL}}^2}{2} \left( \frac{\chi_{YY}^*}{\chi_{YX}^*} + \frac{\chi_{YY}}{\chi_{YX}} \right) |\chi_{YX}| \sin\theta \sinh 2r,$$

(E26)

where we have defined the standard quantum limit for the square root of the double-sided power spectral density

$$h_{\text{SQL}} = \sqrt{\frac{2}{m^2 L^2 \omega^4} \frac{|\chi_{YX}(\omega)|}{|\chi_{YF}(\omega)|^2}}.$$

(E27)

We now make the standard approximation of treating the suspended mirrors as free masses. Recall that this approximation manifests itself in the mechanical susceptibility as $\chi_m \to -1/m\omega^2$. Further, recalling the convenient definition of $\mathcal{K} = |\chi_{YX}|$, the strain sensitivity simplifies considerably to

$$S_{hh}^{SMS}(\omega) = \frac{h_{\text{SQL}}^2}{2} \left( \mathcal{K} + \frac{1}{\mathcal{K}} \right) \cosh 2r$$
$$+ \frac{h_{\text{SQL}}^2}{2} (-\mathcal{K} + \frac{1}{\mathcal{K}}) \cos\theta \sinh 2r$$
$$+ \frac{h_{\text{SQL}}^2}{2} \left( \frac{2}{\mathcal{K}} \right) \mathcal{K} \sin\theta \sinh 2r.$$

(E28)

Next, to make a direct comparison to the un-squeezed case, we essentially factor out the un-squeezed PSD from the full expression to obtain

$$S_{hh}^{SMS}(\omega) = \frac{h_{\text{SQL}}^2}{2} \left( \mathcal{K} + \frac{1}{\mathcal{K}} \right) [\cosh 2r$$
$$+ \left( \frac{1-\mathcal{K}^2}{1+\mathcal{K}^2} \cos\theta + \frac{2\mathcal{K}}{\mathcal{K}^2+1} \sin\theta \right) \sinh 2r].$$

(E29)

The final, fairly non-obvious, step is to realize that further simplification is possible if we identify a rotation angle $\Phi \equiv 2\arctan 1/\mathcal{K}$. Doing so, the expression becomes

$$S_{hh}^{SMS}(\omega) = \frac{h_{\text{SQL}}^2}{2} \left( \mathcal{K} + \frac{1}{\mathcal{K}} \right) [\cosh 2r$$
$$- (\cos\Phi\cos\theta - \sin\Phi\sin\theta) \sinh 2r].$$

(E30)

Then, using a standard trig identity, we arrive at our final expression from the main text.

$$S_{hh}^{SMS}(\omega)$$
$$= \frac{h_{\text{SQL}}^2}{2} \left( \mathcal{K} + \frac{1}{\mathcal{K}} \right) [\cosh 2r - \cos(\theta + \Phi) \sinh 2r].$$

(E31)

This matches the expression in Eq. (46) of Ref. [107], a standard reference in the field.

## Appendix F: Dark matter distribution

The basic model for terrestrial dark matter searches is that the galaxy's mass is dominated by a spherical "halo" of dark matter, with the visible matter forming a small planar region inside. See Fig. 19. The dark matter's mass distribution is not known in detail, however, it is generally expected to have a non-trivial radial profile $\rho(r)$, such that the average mass density in our local part of the galaxy is of order

$$\rho_{\text{DM}} \approx 0.3 \frac{\text{GeV}}{\text{cm}^3}.$$

(F1)

The uncertainty on this is low in the sense that the total DM mass in the galaxy is well measured by a variety of astrophysical observations. However, it should be emphasized that these only probe the distribution on scales of order parsecs ($\sim$ lightyears), so the uncertainty about our local density is quite high, e.g., we could easily be living in a large over- or under-density compared to Eq. (F1). In practice, we will follow standard convention and show projections and exclusion plots based on assumption that Eq. (F1) is correct in the vicinity of Earth.

Importantly, the dark matter must be "cold" in order to form the halo. Here cold means that the kinetic energy of the DM, in the galactic frame, is sub-dominant to its rest mass. It also means that in the rest frame of the average dark matter ("galaxy rest frame"), the DM is distributed according to a Boltzmann distribution,

$$f_{\text{DM,rest}}(v) \sim e^{-v/v_0}, \quad v_0 \approx 10^{-3},$$

(F2)

or about $v_0 \approx 220$ km/s in SI units. This is the "virial velocity". As Earth-bound observers we are not in this rest frame, but rather moving through the background DM distribution with a velocity $v_{\text{obs}}$, which also happens to be around the same as the galactic virial velocity. The distribution expected on Earth is given by

$$f(v) = \frac{v}{\sqrt{\pi}v_0 v_{\text{obs}}} e^{-(v+v_{\text{obs}})^2/v_0^2} (e^{4vv_{\text{obs}}/v_0^2} - 1),$$

(F3)

with $v_{\text{obs}} \approx 232$ km/s. Note that the Earth's velocity through the galaxy also picks out a particular direction, so we do not expect to observe an isotropic DM signal but rather one which is directional and modulated in time according to both the period of the Earth's self-rotation and its rotation around the Sun.

Beyond these two basic facts, to say more about the expected DM distribution requires specifying the mass of a particular DM candidate of interest. This is usually broken into two categories: particle and wave-like DM. The borderline is blurry, but the basic idea is simple. Consider a dark matter candidate with mass $m_\chi$. The phase space for single quanta is divided up into de Broglie cells of order $\lambda_\chi = 2\pi/m_\chi v_0$. We can then ask what fraction of these de Broglie cells are occupied by the DM halo. Let $n_{\text{DM}} = \rho_{\text{DM}}/m_\chi$ be the number density (assuming

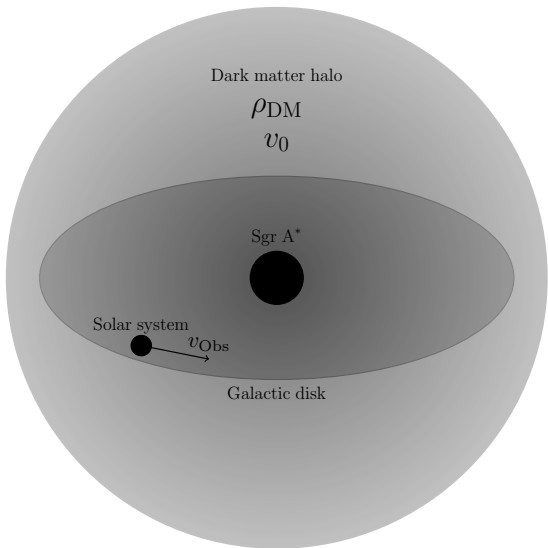

FIG. 19. **The dark matter halo:** Here we show how the DM distribution in our galaxy is typically modeled. The local dark matter has density $\rho_{\rm DM}$ and a velocity dispersion $v_0$ in the galactic rest frame. The Solar system moves with respect to this frame due to its orbit around Sagittarius A$^*$ with a velocity $v_{\rm Obs}$ in the direction of the constellation Cygnus. The orbit of the Earth around the Sun induces subdominant velocity modulations of $\mathcal{O}(10\%)$ on top of this, which we do not draw.

all of the dark matter consists of $\chi$ excitations), then we have

$$\lambda_\chi^3 n_{\rm DM} \approx 60 \times \left(\frac{10 \text{ eV}}{m_\chi}\right)^3. \tag{F4}$$

We see that for reasonably heavy candidates with $m \gtrsim 10$ eV (i.e., about 100 times lighter than an electron), the field is distributed as a more or less dilute gas of particles. Conversely, for "ultra-light" quanta with $m_\chi \lesssim 10$ eV, the field is condensed, forming more of a wave-like background rather than a gas. Note also that the typical wavelength of such a condensate is of order $\lambda_\chi \gtrsim 10^{-4}$ m $\times (10 \text{ eV}/m_\chi)$ and thus tends to be coherent over macroscopic scales.

In the case of ultra-light dark matter with $m_\chi \lesssim 10$ eV, the basic picture is that our Earth-bound laboratory is filled with a highly occupied, long-wavelength "conden-

sate" of ambient dark matter excitations. Since the energy of a non-relativistic field is approximately it's rest mass, the wavelength of the dark matter field is approximately $\lambda_\chi \sim 1/(\omega_\chi v_0) \gg 1/\omega_\chi$.

A more accurate description for most candidates is to imagine randomly superposing a large number of plane waves, each with a random phase and with directions distributed according to the Boltzmann distribution as viewed in our moving frame. In its own rest frame, the field oscillates at a frequency approximately equal to its mass $\omega_\chi \approx 2\pi m_\chi$. Superposing many such waves yields a signal in our frame with some frequency distribution, with width $\gamma_\chi$ set by the dark matter's velocity dispersion $v_0$, specifically $\gamma_\chi \approx \omega_\chi v_0^2 \approx 10^{-6}\omega_\chi$. In other words, the dark matter signal itself has an effective quality factor $Q_\chi = \omega_\chi/\gamma_\chi \approx 10^6$.

In more detail, one can derive an expected power spectral density for these fields [166]. This is given by

$$S_{\chi\chi}(\nu) = \frac{2\pi\rho_{\rm DM}}{m_\chi^3} \frac{f(v)}{v}\Big|_{|v|=\sqrt{\frac{2\nu}{m_\chi}-2}}. \tag{F5}$$

Here $\rho_{\rm DM}$ is given in Eq. (F1), and the velocity distribution is the one we observe on Earth, given in Eq. (F3). In our units, $\chi$ is a canonical bosonic field and thus has dimensions of a mass, so $S_{\chi\chi}$ has units of mass$^2$/Hz.

The integrated signal power is

$$\int \frac{d\nu}{2\pi} S_{\chi\chi}(\nu) = \int_{-\infty}^{\infty} \frac{dv}{2\pi} m_\chi v S_{\chi\chi}(v) \tag{F6}$$

$$= \frac{2\rho_{\rm DM}}{m_\chi^2}. \tag{F7}$$

We occasionally approximate $S_{\chi\chi}(\nu)$ as being flat across an angular frequency range $\Delta\omega_\chi \approx v_0^2 m_\chi$ with the constant value

$$\bar{S}_{\chi\chi} \equiv \frac{2\rho_{\rm DM}}{m_\chi^2 \Delta\omega_\chi}, \tag{F8}$$

such that

$$\Delta\omega_\chi \bar{S}_{\chi\chi} = \int \frac{d\nu}{2\pi} S_{\chi\chi}(\nu). \tag{F9}$$

.