# Peer review of "Quantum measurements in fundamental physics: a user's manual"

_SciPost Physics Reviews_

## Round 1 · Referee Report · Anthony Brady (Referee 2) · 2025-8-25

Strengths

  • Gives a detailed, pedagogical exposition of quantum sensing for high-energy physics applications — e.g., searching for dark matter or new particles — with linear bosonic sensing devices, such as microwave cavities and mechanical oscillators.

  • The great benefit is the interdisciplinary nature and the introductory feel, which should be accessible to quantum measurement experts and high-energy theorists/experimentalists/phenomenologists alike.

Weaknesses

  • One downside of the manuscript is the strict focus on linear detectors. This isn't ideal for 2 reasons: (1) Much of, e.g., dark matter searches are moving to photon-counting based readouts — a non-linear, non-Gaussian detection scheme — since such may bypass vacuum fluctuations of linear detection [see suggested refs. below]. This fact was pointed out early by Lehnart+2013 with regard to axion hunting. (2) In the current paper, there is actually a hidden subsection on QND measurements with a qubit, which is really a photon-number detector in disguise!! (3) In the absence of radiation-pressure back-action, excitation measurements are generically optimal for problems of interest in this paper!

-- I thus suggest a few dedicated paragraphs to be placed somewhere about photon-counting (or more generally, excitation measurement) technology/techniques. I think this along with some relevant references will greatly strengthen the scope and usefulness.

  • The initial heuristic discussion about the SQL is very nice, especially for those that are not accustomed with quantum measurement science. However, there is an abrupt transition from II.A => II.B which undermines the pedagogical exposition. [I myself was confused at times in II.B]. I don't know the exact fix, but softening this transition would be helpful for the reader.

Report

This is overall a great, useful reference for quantum measurement scientists and the hep community alike. After minor revisions, the work should certainly be published.

Requested changes

  • Below eq. (118), the authors state "One can get a similar expression for the amplitude quadrature X^out, but we will mostly be considering the detectors which measure the output phase." A comment justifying this would be useful for the reader. [I am personally aware that the output amplitude quadrature doesn't hold any information about the signal, but the general reader may be befuddled.]

  • Below eq. (137), should it read "Fig. 11" instead of "Fig. 10"?

  • I am reading a black-and-white copy. So curves in figs. 9, 11, 16, 17 are very hard to discern! [Figs. 14 and 15 are good though.] Using a different line scheme (dots, dot-dashed etc.) or more contrast would be beneficial.

  • Finally a list of suggested (but non-exhaustive) references:

-- Photon counting: Dark-matter-candidate searches (10.1103/PhysRevX.15.021031) and (10.1103/PhysRevLett.132.140801) with microwave cavities; ideas for photon counting with LIGO-type receivers (10.1103/PhysRevX.15.011034)

-- Quantum sensing: original squeezing receiver proposal for haloscopes + comparison to photon counters (https://arxiv.org/abs/1607.02529); distributed quantum sensing review (10.1088/2058-9565/abd4c3); direct application to gB-L dark matter search with optomechanical sensor network (10.1038/s42005-023-01357-z)

Recommendation

Publish (easily meets expectations and criteria for this Journal; among top 50%)

---

## Round 1 · Referee Report · Anonymous (Referee 3) · 2025-10-22

Report

The paper is a very useful and timely resource for quantum measurements in fundamental physics. It aims to be a pedagogical introduction to the standard quantum limits and ways to evade them in many contexts, emphasizing the important use cases of axion dark matter, dark photon dark matter and gravitational wave detection. The paper is written well, covers an important topic and deserves publication.

My comments and questions are more from the point of view of a potential user of the manual rather than the expert, so I leave it up to the authors' judgement on how much of my suggestions to implement.  

Overall comments:The organization of sections and subsections may be a bit better, separating pedagogical examples, general formalism and applications. There is a proliferation of variables and definitions used, with/without subscripts. It may be important to include them for consistency with literature, but reducing the number of different variables representing the same physics will help pedagogy a lot. I include a couple of examples In the discussion above equation (110), x_0 and p_0 are introduced but never used again. (There is probably a typo defining x_m) In equation 117 phi_c is only implicitly defined. There are many equations, and sometimes they can feel like moving variables around. When the end result (SNR / scan rate / sensitivity curves) is plotted, it's sometimes far from clear where the functional form is coming from. Sometimes it is not explicitly provided, other times it requires backtracking variables and definitions for many pages. The figures are very useful and important, and putting in an explicit functional form for the sensitivity curves from and a contained quick explanation for the behaviour will be very useful (e.g. figure 7)

  • Appendix D does not seem to add anything, and can be left as a comment? In figure 3, where does the functional dependence come from? What is the X-axis? In figure 5, was the integration time chosen to be 15 min? Might be good to clarify. What was the integration time used by Haystack? In equation (28), is S_FF the PSD for F_E? Or O_out? It was not clear to me why the pendulum example was relevant. What about the pendulum is used? It may help the reader to clarify where equations (58) and (59) come from. In equation (88) will it not be useful to include how the SNR scales if the integration time is greater or less than the coherence time? Below equation (105) how do we compare the dimensionful coupling rate to the dimensionless strain? Below (156): I am not sure if I have understood the setup. If we are talking about a scattering \delta p such that one electron kick is resolved, can we still model this as coupling to the zero mode CoM predominantly, and not to higher oscillator modes as well? Some minor typographical comments that I noticed: fig 10 - > fig 11 below equation (137) equation (148) \slashed{A}should go between n's* in paragraph below equation(150), \chi is not introduced, and m_chi -> m_\chi

Recommendation

Publish (surpasses expectations and criteria for this Journal; among top 10%)

---

## Editorial Decision

awaiting_resubmission